# Restoring nuclear entry of Sirtuin 2 in oligodendrocyte progenitor cells promotes remyelination during ageing

Xiao-Ru Ma [1,16], Xudong Zhu[2,16], Yujie Xiao[3], Hui-Min Gu[1], Shuang-Shuang Zheng[1], Liang Li[3], Fan Wang[1], Zhao-Jun Dong[1], Di-Xian Wang[1], Yang Wu [1], Chenyu Yang[4], Wenhong Jiang[5], Ke Yao [6], Yue Yin[7], Yang Zhang[8], Chao Peng [7], Lixia Gao[9], Zhuoxian Meng [10], Zeping Hu [6,11], Chong Liu [5], Li Li[12], Hou-Zao Chen [13✉], Yousheng Shu[14✉], Zhenyu Ju [15✉] & Jing-Wei Zhao [1,4✉]

The age-dependent decline in remyelination potential of the central nervous system during ageing is associated with a declined differentiation capacity of oligodendrocyte progenitor cells (OPCs). The molecular players that can enhance OPC differentiation or rejuvenate OPCs are unclear. Here we show that, in mouse OPCs, nuclear entry of SIRT2 is impaired and $NAD^+$ levels are reduced during ageing. When we supplement β-nicotinamide mononucleotide (β-NMN), an $NAD^+$ precursor, nuclear entry of SIRT2 in OPCs, OPC differentiation, and remyelination were rescued in aged animals. We show that the effects on myelination are mediated via the $NAD^+$-SIRT2-H3K18Ac-ID4 axis, and SIRT2 is required for rejuvenating OPCs. Our results show that SIRT2 and $NAD^+$ levels rescue the aged OPC differentiation potential to levels comparable to young age, providing potential targets to enhance remyelination during ageing.

[1] Department of Pathology of Sir Run Run Shaw Hospital and Department of Human Anatomy, Histology and Embryology, System Medicine Research Center, Center for Neuroscience, NHC and CAMS Key Laboratory of Medical Neurobiology, Zhejiang University School of Medicine, Hangzhou 310058 Zhejiang, China. [2] Key Laboratory of Aging and Cancer Biology of Zhejiang Province, Department of Pathology and Pathophysiology, School of Basic Medical Sciences, Hangzhou Normal University, Hangzhou 311121 Zhejiang, China. [3] Department of Neurology, Huashan Hospital, State Key Laboratory of Medical Neurobiology, Institute for Translational Brain Research, MOE Frontiers Center for Brain Science, Fudan University, 200032 Shanghai, China. [4] Center of Cryo-Electron Microscopy, Zhejiang University, Hangzhou 310058 Zhejiang, China. [5] Zhejiang University School of Brain Science and Brain Medicine, and Department of Neurosurgery of the Second Affiliated Hospital, Zhejiang University School of Medicine, Hangzhou 310058 Zhejiang, China. [6] School of Pharmaceutical Sciences, Tsinghua University, 100084 Beijing, China. [7] National Facility for Protein Science in Shanghai, Zhangjiang Lab, Shanghai Advanced Research Institute, Chinese Academy of Sciences, 201210 Shanghai, China. [8] Department of cardiology, Zhongshan Hospital of Fudan University, 200035 Shanghai, China. [9] Department of Neurology of the Second Affiliated Hospital, Interdisciplinary Institute of Neuroscience and Technology, Zhejiang University School of Medicine, 310020 Hangzhou, China. [10] Department of Pathology and Pathophysiology and Zhejiang Provincial Key Laboratory of Pancreatic Disease of the First Affiliated Hospital, Key Laboratory of Disease Proteomics of Zhejiang Province, Zhejiang University School of Medicine, 310058 Hangzhou, China. [11] Tsinghua-Peking Joint Center for Life Sciences and Beijing Frontier Research Center for Biological Structure, Tsinghua University, 100084 Beijing, China. [12] Beijing Key Laboratory of Drug Targets Identification and Drug Screening, Institute of Materia Medica, Chinese Academy of Medical Sciences & Peking Union Medical College, 100050 Beijing, China. [13] State Key Laboratory of Medical Molecular Biology, Department of Biochemistry and Molecular Biology, Institute of Basic Medical Sciences, Chinese Academy of Medical Sciences & Peking Union Medical College, 100005 Beijing, China. [14] State Key Laboratory of Cognitive Neuroscience and Learning & IDG/McGovern Institute for Brain Research, Beijing Normal University, 100875 Beijing, China. [15] Key Laboratory of Regenerative Medicine of Ministry of Education, Guangzhou Regenerative Medicine and Health Guangdong Laboratory, Institute of Ageing and Regenerative Medicine, Jinan University, Guangzhou 510632 Guangdong, China. [16]These authors contributed equally: Xiao-Ru Ma, Xudong Zhu. ✉email: chenhouzao@ibms.cams.cn; yousheng@fudan.edu.cn; zhenyuju@163.com; jingweizhao@zju.edu.cn

The increasingly ageing global population has ignited great interest in identifying rejuvenating molecules that may delay ageing and enhance the regeneration of the aged central nervous system (CNS). The ensheathment of axons by myelin gives structure to saltatory conduction[1,2], and provides metabolic support to maintain both axonal functional integrity and long-term survival[3,4]. Therefore, proper myelination is not only a prerequisite for, but also a consequence of, normal CNS activity[5,6]. As the human brain ages, white matter volume shrinks more prominently than grey matter[7] and some myelin sheaths exhibit myelin ageing[8–10], which in turn drives CNS ageing[11]. The reasons for myelin ageing remain unclear. However, the decreased remyelination capacity of oligodendrocyte progenitor cell (OPC) is one of the popularly proposed mechanisms[11].

Demyelination is the key pathological feature of the auto-immune inflammatory diseases of CNS such as multiple sclerosis (MS)[11,12] and an early pathological hallmark of neurodegenerative diseases[13,14]. Consequently, demyelination may cause devastating irreversible axonal degeneration[15]. Remyelination by oligodendrocytes differentiated from OPCs occurs throughout life[12,16]. Unfortunately, with ageing, the efficiency of remyelination declines mainly due to the reduced capacity of OPC differentiation[17–21]. Such decline leads to disability in MS[18], whose course usually spans several decades and progresses with ageing[22]. It is essentially untreatable when MS develops into a progressive phase[22,23]. Identifying new molecular targets in OPC to rejuvenate the aged OPC therefore holds much promise for this unmet medical need.

As the central player for myelination and remyelination, OPCs are distributed throughout the CNS[12,24]. The ageing of OPCs is orchestrated by epigenetic mechanisms[21,25] likely involving sirtuins, the nicotinamide adenine dinucleotide (NAD$^+$)-dependent histone deacetylases. Among the seven members of sirtuins, it remains elusive which members of the sirtuin family are expressed in the OPC. Sirtuin 2 (SIRT2) is the only member which is predominantly localised in the cytoplasm, and its role in the CNS remains largely unclear[26–28]. So far, although mature oligodendrocytes express SIRT2[29–31], it is not clear whether OPCs in the CNS express SIRT2, or what role SIRT2 plays in remyelination in the aged CNS.

Enhancing remyelination in the aged CNS of animal models has proven to be an effective strategy to rejuvenate aged OPCs, and recent studies have demonstrated that a youthful systemic environment[20], fasting, or metformin[18] can restore remyelination efficiency in the aged CNS, indicating that the aged OPC can be rejuvenated. Emerging evidence suggests that the metabolomic profile defines an OPC's cell fate and shapes its ability to differentiate[32]. However, so far it is not clear how the endogenous metabolic fingerprints of OPCs are affected by ageing.

Herein, we report that the depletion of SIRT2 and its nuclear localisation together with declined NAD$^+$ are features of aged OPCs. Importantly, supplementation of NAD$^+$ by β-nicotinamide mononucleotide (β-NMN) induces re-expression and restores nuclear entry of SIRT2 in OPCs, and rejuvenates aged OPCs by promoting them differentiating into mature oligodendrocytes and eventually enhances new myelin generation in the aged CNS. Mechanistically, we reveal that SIRT2 is necessary for the effect of NAD$^+$ on OPCs. This study identifies SIRT2 as a molecular target for OPCs, and by restoring the nuclear entry of SIRT2 in OPCs, β-NMN delays myelin ageing in the normal CNS and enhances remyelination in the demyelinated aged CNS.

## Results

**Nuclear entry of SIRT2 in OPCs during remyelination is impaired in the aged mice in vivo.** For myelination and remyelination, the OPC is the central player[12]. However, it remains unclear which members of the seven sirtuins are expressed in OPCs. Using primary cultured rat OPCs (Fig. 1a), the mRNA levels of *sirt1* to *sirt7* were compared by qRT-PCR. Surprisingly, we found that the abundance of *sirt2* mRNA was dominant over any other member of sirtuin family for over 50 folds in the OPC (Fig. 1b), and similar pattern was seen in the mature oligodendrocyte (Fig. 1b). The mRNA level of sirtuins in OPCs and mature oligodendrocytes were almost identical between rat and mouse (Supplementary Fig. 1a). Among different cell types in the rat brain, *sirt2* mRNA expression was the highest in mature oligodendrocyte, and the second highest in OPC, whereas very little in neuron, microglia, and astrocyte (Fig. 1c).

Next, we investigated whether OPCs express the SIRT2 protein during myelin development in vivo. The SIRT2 expression profile in both white matter such as corpus callosum and optic nerve and the grey matter were detected in the postnatal mouse brain. SIRT2 was indeed expressed in OPCs in the brain from postnatal day 0 (P0) to P21 (Fig. 1d), the crucial time window for postnatal development of CNS myelin[33], and the expression declined from P0 in a time-dependent manner. At P0, when OPCs start differentiating, SIRT2 was expressed in a significant number of OPCs, consistent with our in vitro data (Fig. 1b). While at P21 when the differentiation of OPCs slows down, the number of OPCs expressing SIRT2 reduced dramatically. Interestingly, about 2/3 of the SIRT2$^+$ OPCs had SIRT2 in their nuclei (Fig. 1d–e). At P90, when myelin development has been completed, SIRT2 was depleted in OPCs and was exclusively expressed in the cytoplasm of mature oligodendrocytes (Fig. 1d–f, see images with broader view in Supplementary Fig. 1b, c and Supplementary Fig. 2a). To investigate whether OPCs in the primate brain express SIRT2, with the brain of the marmoset, we showed that the nuclear expression of SIRT2 in OPCs was indeed observed in the cortex at P3 (Fig. 1g and Supplementary Fig. 1d–e), whereas in the adult (aged 8 years) SIRT2 was not observed in OPCs but exclusively in the cytoplasm of mature oligodendrocytes (Fig. 1h). However, in cell types other than OPCs and mature oligodendrocytes, such as the astrocyte, neuron, microglia, and endothelial cell, the expression of SIRT2 was not detectable in the brain of either mouse or marmoset (Supplementary Fig. 2b, c). In the adult human cerebral cortex, SIRT2 was also expressed in the cytoplasm of mature oligodendrocyte-like cells and on myelin-like structures (Fig. 1i). Our results indicate that SIRT2 is expressed in postnatal OPCs, and its expression pattern switches from nucleus-containing in the majority of OPCs during myelin development to exclusively cytoplasmic in mature oligodendrocytes. This is an evolutionary reserved phenomenon spanning from mouse, monkey to human.

Further, we explored how the SIRT2 protein level in the brain changes with progressive ageing, and found that SIRT2 declined in old mice aged 18 months (M) when compared to young mice aged 6 M (Fig. 1j, k). It is currently unclear how SIRT2 expression is regulated by remyelination during ageing. To this end, we induced a focal demyelination by injecting lysolecithin (LPC) into the corpus callosum of the mouse brain, and remyelination occurs[12]. Interestingly, at 5 days post-lesion (dpl), when OPC proliferation reaches its peak, SIRT2 re-expression occurred in 70% OPCs in the lesion (Fig. 1l–m) in young mice, and nuclear localisation of SIRT2 was seen in about half of SIRT2$^+$ OPCs (Fig. 1n–o, Supplementary Fig. 2d). In contrast, in the old mice aged 21 M, demyelination upregulated the level of SIRT2 in OPCs to a much lower level than that of the young mice (Fig. 1l, Supplementary Fig. 2d), equivalent to 1/3 in the proportion (Fig. 1m) and about half in the density of SIRT2$^+$ OPCs of the young mice (Fig. 1n). Among SIRT2$^+$ OPCs, nuclear SIRT2$^+$

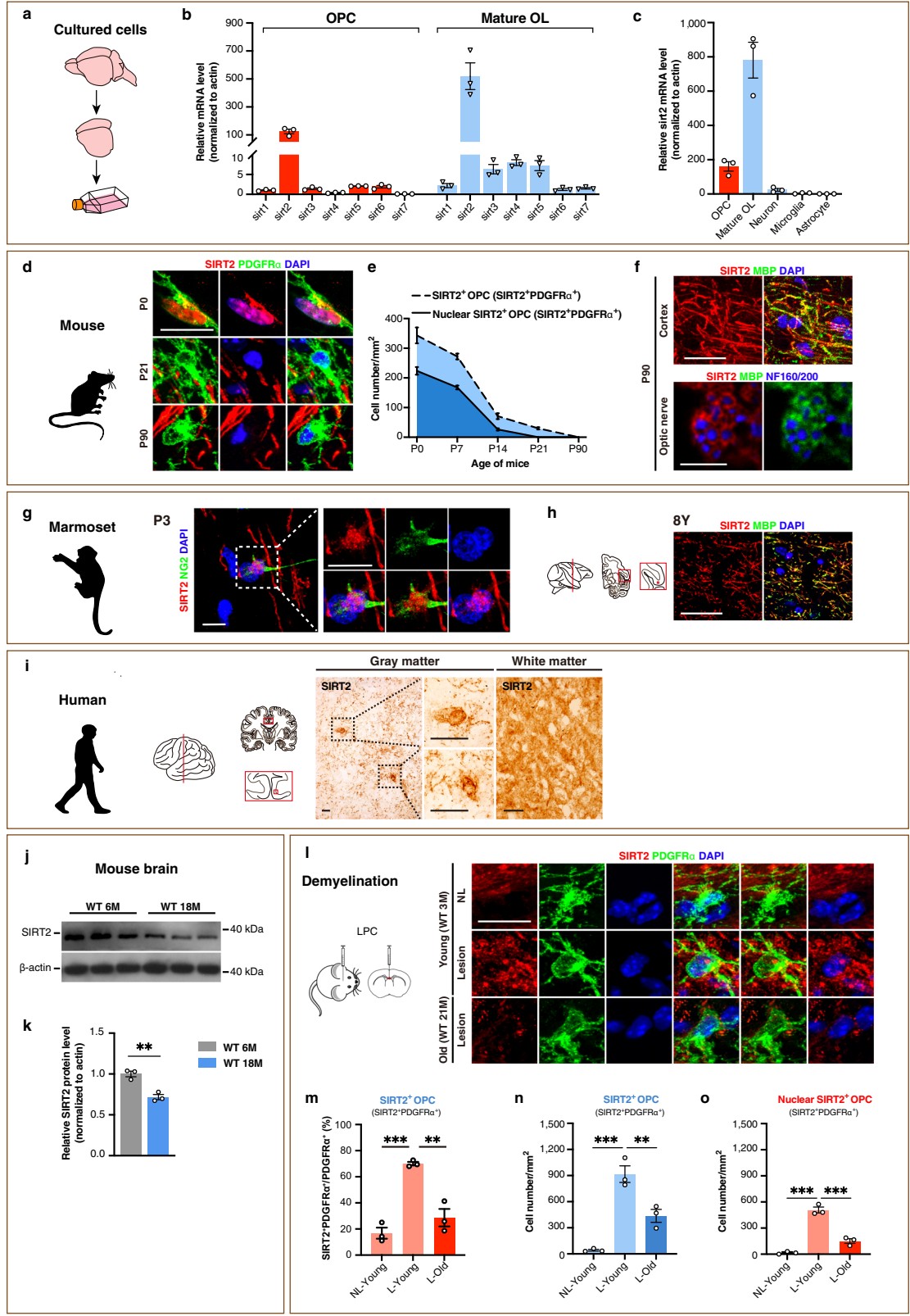

OPCs in the aged was only 1/3 of the young (Fig. 1l–o, Supplementary Fig. 2d). Our results reveal that the nuclear localisation of SIRT2 in OPCs during remyelination declines during ageing. Interestingly, our results show that the depleted nuclear entry of SIRT2 in aged OPCs within the demyelination lesion correlates well with the decline of remyelination capacity during ageing[12,18,20].

**SIRT2 is critical for remyelination of the mice in vivo**. So far, the function of SIRT2 in OPCs remains unclear. To explore this, we used SIRT2−/− mice whose knockout efficiency was confirmed by Western blot (Fig. 2a). In the primary OPC culture, SIRT2 knockout impaired both the proliferation (Fig. 2b, c) and the differentiation (Fig. 2d, e) of OPCs. We further induced demyelination in the mice by injecting LPC. Following

**Fig. 1 Nuclear entry of SIRT2 in OPCs during remyelination is impaired in the aged mice in vivo. a** Primary culture of various types of cells from P0 rat cortex. **b** qRT-PCR of seven members of sirtuins in primary cultured OPCs and mature oligodendrocytes from the cortex of P0 rat ($n = 3$). **c** qRT-PCR of *sirt2* in primary cultured various types of cells from the cortex of P0 rat ($n = 3$). **d–f** Immunofluorescence and quantification of SIRT2+ cells in the cortex of mice at different ages ($n = 3$). Scale bar, 10 μm (**d**), 50 μm (upper panel images of **f**), 5 μm (lower panel images of **f**). **g–h** Immunofluorescence of SIRT2 in the cortex of marmosets at postnatal day 3 (P3, **g**) or age of 8 years (**h**). Scale bar, 50 μm. **i** Immunohistochemistry of SIRT2 in the cortex of human at age of 53 years. Scale bar, 20 μm. **j, k** Relative SIRT2 protein level in brains of WT young (6 M) and old (18 M) mice ($n = 3$). **l–o** Immunofluorescence and quantification of SIRT2+ OPCs and nuclear SIRT2+ OPCs in corpus callosum of WT young and old mice ($n = 3$). NL, non-lesion, L, demyelination lesion induced by LPC at 5 dpl. Scale bar, 10 μm. All data are presented as mean ± SEM. *$p < 0.05$, **$p < 0.01$, ***$p < 0.001$ by two-tailed *t*-test (**k**) or one-way ANOVA followed by Tukey's post hoc test (**m–o**). In all instances ***$p < 0.001$. *n.s.* no significance. In (**k**), **$p = 0.004$; in (**m**), **$p = 0.002$ (L-Young vs. L-Old); in (**n**), **$p = 0.007$ (L-Young vs. L-Old).

demyelination, compared with the WT mice, SIRT2−/− mice showed significantly decreased numbers of both proliferating OPCs (Ki67+Olig2+ cells) (Fig. 2f–j) and differentiated oligodendrocytes (CC1+Olig2+ cells), and the differentiation of OPCs is more severely impaired (Fig. 2k–o).

To accurately evaluate the remyelination efficiency, we used the transmission electron microscopy (TEM) technique to distinguish unambiguously the normal myelinated, demyelinated (no myelin) or remyelinated (newly formed myelin, thinner) axons within the lesion (Fig. 2p, q). Taking the published studies[8–10] into account, together with our own careful observation, in this work we established a set of graded evaluation criteria from grade 0 (normal) to grade 4 on the ultrastructure of myelin in the corpus callosum (Supplementary Data 1). Since myelin can become loose via demyelination in the old, the G-Ratio cannot entirely represent the myelin thickness. Therefore, combining G-Ratio with the distance between the dense line (DL) becomes a more reliable way to evaluate myelin thickness and compaction. According to this set of criteria, at 21 dpl, when remyelination is completed, we showed that the young SIRT2−/− mice (aged 6 M) showed more impaired remyelination efficiency not only than WT young but also than WT old (aged 18 M) (Fig. 2r, s). Graded myelin analysis further revealed this was mainly due to SIRT2−/− mice having more grade 2 myelin pathology than WT old (Fig. 2t), indicating that the general myelin structure in SIRT2−/− mice is worse than that of WT old mice. Moreover, SIRT2−/− mice exhibited thinner and looser newly-formed myelin than WT young mice: a result no different to the WT old mice (Fig. 2u–x). Collectively, we show that the SIRT2−/− mice exhibit impaired remyelination with thinner and looser new myelin, more severe than even the WT old mice three times of its biological age. Our in vitro and in vivo data indicate that SIRT2 is critical for remyelination.

Given that myelin not only ensheathes axons but also provides energy to axons[3,4], using samples from corpus callosum, we next examined the structure of axons in SIRT2−/− mice under TEM. Our data show that in comparison with WT young (6 M) mice, the proportion of normal-looking axons decreased in age-matched SIRT2−/− mice, similar to WT old (18 M) mice (Supplementary Fig. 3), suggesting that there is axonal pathology in SIRT2−/− mice and WT old mice.

To test whether loss of SIRT2-caused impaired remyelination is a transient effect, we induced demyelination in SIRT2−/− mice and tested the remyelination at 30, 60, and 90 dpl. The results reveal that even when SIRT2−/− mice took over fourfolds of time (90 dpl) than WT young mice (21 dpl), the remyelination still did not fully reach the level of age-matched WT mice (Supplementary Fig. 4), suggesting that the impaired remyelination in SIRT2−/− mice is NOT a transient effect.

To clearly investigate the role of SIRT2 specifically in OPCs in vivo, we obtained NG2-CreERT; Sirt2flox/flox mice by crossing Sirt2flox/flox mice with NG2-CreERT mice (Supplementary Fig. 5a). The mice were intraperitoneally (i.p.) injected with tamoxifen for

5 days[34–36] to exclusively delete SIRT2 in OPCs. Four days later[35,36], we induced demyelination through focal injections of LPC into the corpus callosum and detected remyelination at 21 dpl under TEM (Supplementary Fig. 5b). The genetic deletion of SIRT2 in OPCs was confirmed by immunofluorescence (Supplementary Fig. 5c). Our TEM results showed that conditional knockout of SIRT2 in OPCs reduced the frequency of remyelinated axons (Supplementary Fig. 5d, e) and decreased the thickness of newly-formed myelin (Supplementary Fig. 5f). The pathological analysis of myelin revealed that the frequency of normal looking newly-formed myelin was reduced while the grade 1 and grade 2 myelin increased (Supplementary Fig. 5g). These results provide direct evidence that SIRT2 in OPCs plays a critical role in remyelination.

**Elevating NAD+ by β-NMN enhances SIRT2 nuclear entry in OPCs of the aged mice within demyelination lesion in vivo.** Age-related changes and defects in the myelin structure are key features of CNS ageing[9–11,37]. Progressive telomere attrition is a primary cause of organismal ageing[38]. Critical telomere shortening caused by telomerase deficiency in the third generation of telomerase RNA component knockout mice (G3 Terc−/− mice) leads to premature ageing[38]. After confirming that G3 Terc−/− mice had shorter telomere than the age-matched WT mice (Supplementary Fig. 6a, b), using TEM technique and samples from corpus callosum, we confirmed that G3 Terc−/− mice exhibited a premature ageing phenotype of myelin ultrastructure in the brain (Supplementary Fig. 6e–k). Using primary cultured OPCs and LPC-induced in vivo demyelination, we further revealed that G3 Terc−/− mice exhibited compromised capacity of proliferation (Supplementary Fig. 7a, b; e–i), differentiation (Supplementary Fig. 7c, d; j–n) and remyelination (Supplementary Fig. 7o–v) in comparison with the age-matched WT mice, similar to the old mice three times of its biological age, confirming the premature ageing phenotype of OPC in G3 Terc−/− mice. In parallel, we found that SIRT2 protein level in the brain declined in G3 Terc−/− mice, equivalent to that of the naturally-ageing old mice three times of its biological age (Supplementary Fig. 6c, d).

Endogenous metabolites play a vital role in stem cell ageing[25]. However, it is not clear how the global metabolome was changed in OPCs during ageing. To address this question, we used premature ageing OPC from G3 Terc−/− mice to avoid the technical difficulties of culturing the aged OPC and compared the global metabolome of OPCs from G3 Terc−/− mice with that of age-matched WT mice. To capture the native metabolism status, OPCs were obtained by immunopanning, which took 4 hours (h) for isolation (Fig. 3a). Importantly, the targeted metabolic profiling results showed that NAD+ was one of the top metabolites that significantly decreased in G3 Terc−/− OPCs (Fig. 3b and Supplementary Data 2). The decreased NAD+ level was further confirmed in the brain of young G3 Terc−/− mice

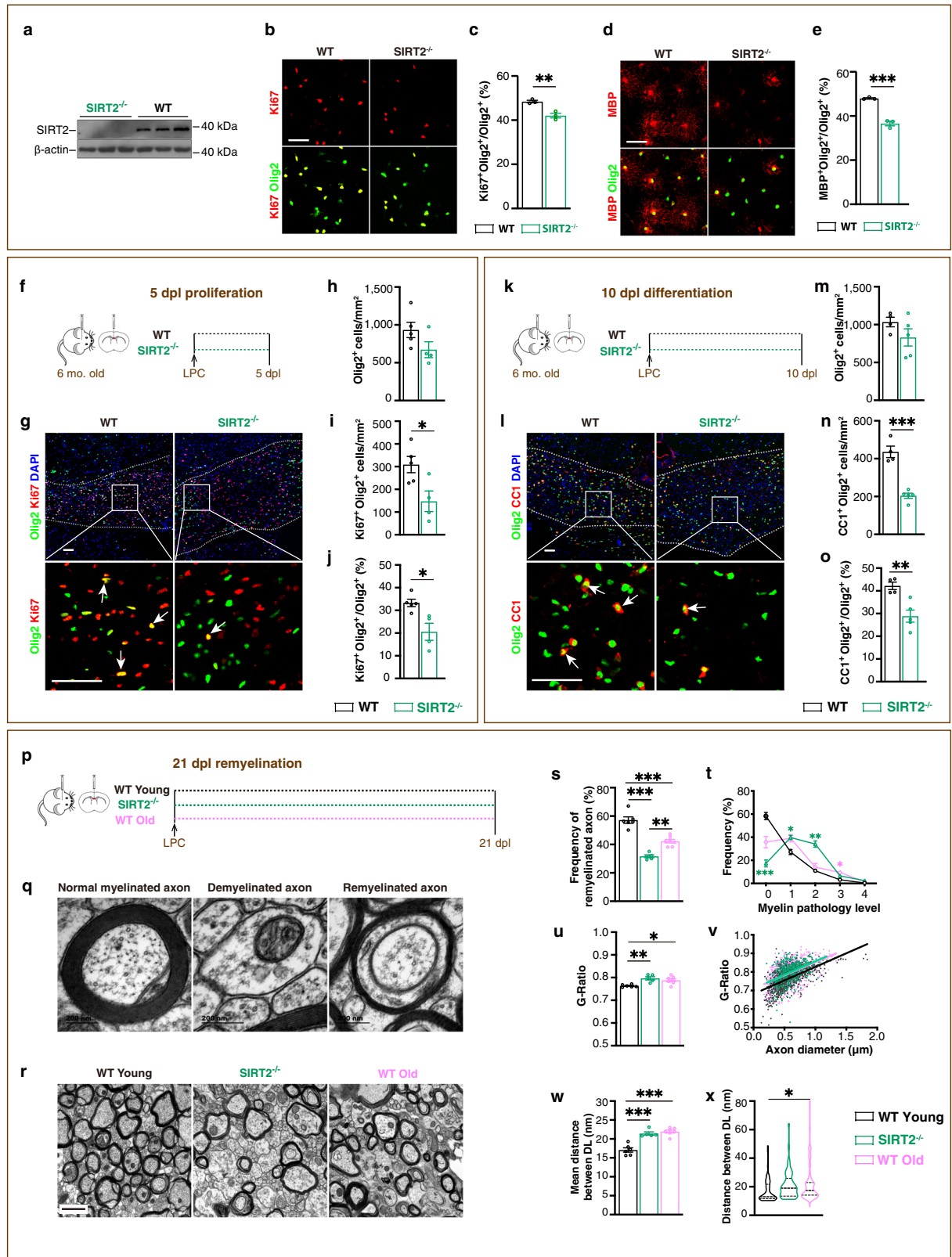

aged 6 M, which had decreased to 1/3 of that in age-matched WT mice, and equivalent to that of old WT mice aged 15 M (Fig. 3c).

We demonstrated here that depleted SIRT2 in the aged OPC correlated with the decline of NAD[+], and that these two processes correlated with declined remyelination during ageing as previously shown[12,18,20]. We further ask whether NAD[+] supplementation could restore the age-related decline of SIRT2

in aged OPCs. β-NMN, an immediate biosynthesis precursor of NAD[+], was used to supplement NAD[+]. Using liquid chromatography-mass spectrometry/mass spectrometry (LC-MS/MS) (Fig. 3d) we investigated the changes of proteome differentially regulated by NAD[+] or DMSO in cultured rat OPCs. The volcano plot shows markedly elevated expression of 399 genes, most of which are involved in metabolism. The metabolic

**Fig. 2 SIRT2 is critical for remyelination of the mice in vivo. a** SIRT2 protein level in brains of SIRT2$^{-/-}$ and WT mice ($n = 3$). **b–e** Images and quantification of proliferating OPCs (**b, c**, Olig2$^+$Ki67$^+$, $n = 3$) cultured for 36 h and differentiated oligodendrocytes (**d, e**, Olig2$^+$MBP$^+$, $n = 3$) cultured for 48 h. Scale bars, 50 µm. **f** Schematic diagram of the experiment for testing OPC proliferation in vivo at 5 dpl. **g–j** Images and quantification of oligodendrocyte lineage cells (Olig2$^+$) and proliferating OPCs (arrows, Ki67$^+$Olig2$^+$) within the demyelination lesions (dotted line) at 5 dpl ($n = 5$ for WT group, $n = 4$ for the SIRT2$^{-/-}$ group). Scale bar, 50 µm. **k** Schematic diagram of the experiment for testing OPC differentiation in vivo at 10 dpl. **l–o** Images and quantification of oligodendrocyte lineage cells (Olig2$^+$) and differentiated oligodendrocytes (arrows, CC1$^+$Olig2$^+$) within the demyelination lesions (dotted line) at 10 dpl ($n = 4$ for WT group, $n = 5$ for the SIRT2$^{-/-}$ group). Scale bar, 50 µm. **p** Experiment design for testing remyelination efficiency in vivo at 21 dpl. **q** TEM micrographs of normal myelinated, demyelinated and remyelinated axons. Scale bar, 200 nm. **r** TEM micrographs within the lesions at 21 dpl. Scale bar, 1 µm. **s–x** Quantification of the proportion of remyelinated axons (**s**), myelin pathology level (**t**), G-Ratio average (**u**), individual G-Ratio distribution (**v**, linear regression) and distance between DL (**w** and **x**) within the lesions at 21 dpl ($n = 6$ for the WT young group, $n = 5$ for the SIRT2$^{-/-}$ group, $n = 7$ for the WT old group). All data are presented as mean ± SEM. The center, upper and lower line represent the median, upper and lower quartiles, respectively (**x**). *$p < 0.05$, **$p < 0.01$, ***$p < 0.001$ by two-tailed $t$-test (**c, e, h–j, m–o**) or one-way ANOVA followed by Tukey's post hoc test (**s, u, w, x**) or two-way repeated ANOVA followed by Sidak's post hoc test (**t**). In all instances ***$p < 0.001$. n.s. no significance. In (**c**), **$p = 0.006$; in (**i**), *$p = 0.03$; in (**j**), *$p = 0.01$; in (**o**), **$p = 0.005$; in (**s**), **$p = 0.001$ (SIRT2$^{-/-}$ vs. WT Old); in (**u**), **$p = 0.004$ (WT Young vs. SIRT2$^{-/-}$), *$p = 0.02$ (SIRT2$^{-/-}$ vs. WT Old); in (**x**), **$p = 0.03$ (WT Young vs. WT Old).

pathways consistently ranked as the top, revealed by biological processes analysis (Fig. 3e, f). Among all seven family members of sirtuins, we screened out that SIRT2 was the only one upregulated by NAD$^+$ in cultured rat OPCs at both protein level by LC-MS/MS (Fig. 3e) and at transcriptional level by qRT-PCR (Fig. 3g and Supplementary Data 3). This set of data are consistent with our results of qRT-PCR and immunostaining of SIRT2 on OPC (Fig. 1). Meanwhile, β-NMN significantly downregulated the mRNA level of several cellular senescence-related genes[18,39] in OPCs from G3 Terc$^{-/-}$ mice (Supplementary Fig. 8a), suggesting multiple mechanisms could be involved in the effect of β-NMN on OPCs.

To test whether β-NMN administration restores NAD$^+$ and SIRT2 level in the brain in vivo, we i.p. injected β-NMN or PBS for 3 months (Fig. 3 h), and found in the brain tissue of G3 Terc$^{-/-}$ mice treated with β-NMN, the level of both NAD$^+$ and SIRT2 protein was doubled than that of PBS-treated control mice (Fig. 3i–k). We further tested whether NAD$^+$ supplementation could restore age-related decline of nuclear localisation of SIRT2 in OPCs in vivo. In terms of SIRT2 nuclear localisation in OPCs, almost no nuclear SIRT2$^+$ OPC was observed in normal adult brain, while NAD$^+$ repletion significantly increased nuclear SIRT2$^+$ OPC density in non-lesion condition (Fig. 3l–n and Supplementary Fig. 8b). Importantly, LPC-induced demyelination further enhanced NAD$^+$ induced SIRT2 nuclear localisation in the aged OPCs over twofolds (Fig. 3l–n and Supplementary Fig. 8b), recapitulating SIRT2 nuclear localisation in OPCs in vivo, a phenomenon that occurred during myelin development (Fig. 1d, g). This set of data indicate that elevating NAD$^+$ level by β-NMN enhances SIRT2 nuclear entry in premature ageing OPC.

**Elevating NAD$^+$ by β-NMN delays myelin ageing in the aged mice in vivo.** To explore whether NAD$^+$ supplementation affects the proliferation and the differentiation of OPCs, we supplied β-NMN to in vitro cultured OPCs and found that β-NMN improved the proportion of proliferating OPCs from G3 Terc$^{-/-}$ mice, but had no effect on OPCs from WT mice (Fig. 4a, b). Interestingly, supplementing β-NMN to primary cultured OPCs improved the proportion of differentiated oligodendrocytes in both G3 Terc$^{-/-}$ mice, WT mice and WT rats (Fig. 4c, d and Supplementary Fig. 9), and upregulated myelin basic protein (MBP) at the protein level in WT rats (Fig. 4e, f). This data indicates that NAD$^+$ supplementation enhances both proliferation and differentiation of aged OPCs while only promoting differentiation of young OPCs in vitro.

To test whether the long-term NAD$^+$ supplementation (daily injection for 3 M) affects the proliferation and/or differentiation

of OPCs in vivo in non-lesion condition, we found in both untreated G3 Terc$^{-/-}$ (6 M) and WT old (21 M) mice, the density of OPCs, oligodendrocytes, the oligodendrocyte lineage cells, and the proportion of the oligodendrocyte lineage cells among all cells declined in comparison with untreated WT young (6 M) mice (Supplementary Fig. 10a–l). We further explored whether β-NMN affected these impaired indexes in G3 Terc$^{-/-}$ and WT old mice. Our results showed that β-NMN increased the density of OPCs, the differentiated oligodendrocytes, the oligodendrocyte lineage cells, and the proportion of the differentiated oligodendrocytes among all cells in both G3 Terc$^{-/-}$ mice and WT old mice (Supplementary Fig. 10a–l). Importantly, β-NMN fully restored all the impaired indexes described above in both G3 Terc$^{-/-}$ mice and WT old mice to the levels of WT young mice, confirming that β-NMN rejuvenates OPCs in both G3 Terc$^{-/-}$ mice and WT old mice.

To explore whether elevating NAD$^+$ delays myelin ageing in vivo, we i.p. injected G3 Terc$^{-/-}$ mice with β-NMN or PBS for 3 months (Fig. 4g). This long-term NAD$^+$ supplementation exerted a similar effect on myelin ultrastructure in both G3 Terc$^{-/-}$ mice (Fig. 4g–n) and WT old mice aged 21 M (Fig. 4o–v). NAD$^+$ supplementation significantly increased the frequency of normal myelin (grade 0), generally decreased the myelin pathological level (grade 1–4, Fig. 4i, q) and decreased the distance between DL (Fig. 4j, k, r, s). These results indicate that long-term NAD$^+$ supplementation delays myelin ageing in vivo by making myelin more compact in both G3 Terc$^{-/-}$ premature ageing and normal aged mice.

**Elevating NAD$^+$ by β-NMN enhances remyelination in the aged mice in vivo.** The encouraging effect of NAD$^+$ on restoring SIRT2 nuclear entry in the aged OPCs and delaying myelin ageing prompted us to test whether NAD$^+$ supplementation affects the proliferation or differentiation of OPCs following demyelination in the aged CNS. To this end, we i.p. injected β-NMN or PBS into G3 Terc$^{-/-}$ mice (3 M) daily for 3 months and then induced demyelination (Fig. 5a). We found that, at 5 dpl (Fig. 5b), NAD$^+$ did not affect the density of oligodendrocyte lineage cells (Fig. 5c), the density and the proportion of OPCs (Supplementary Fig. 11d, e), and the proliferation of OPCs (Fig. 5d, e). Interestingly, at 10 dpl (Fig. 5f–j), NAD$^+$ increased the proportion of differentiated oligodendrocytes by 68% (Fig. 5j) but decreased the density and the proportion of OPCs (Supplementary Fig. 11i–j). These results indicated that NAD$^+$ preferentially promotes the differentiation of OPCs in demyelinated G3 Terc$^{-/-}$ mice in vivo.

To further test whether NAD$^+$ repletion could enhance remyelination in G3 Terc$^{-/-}$ mice, we examined the

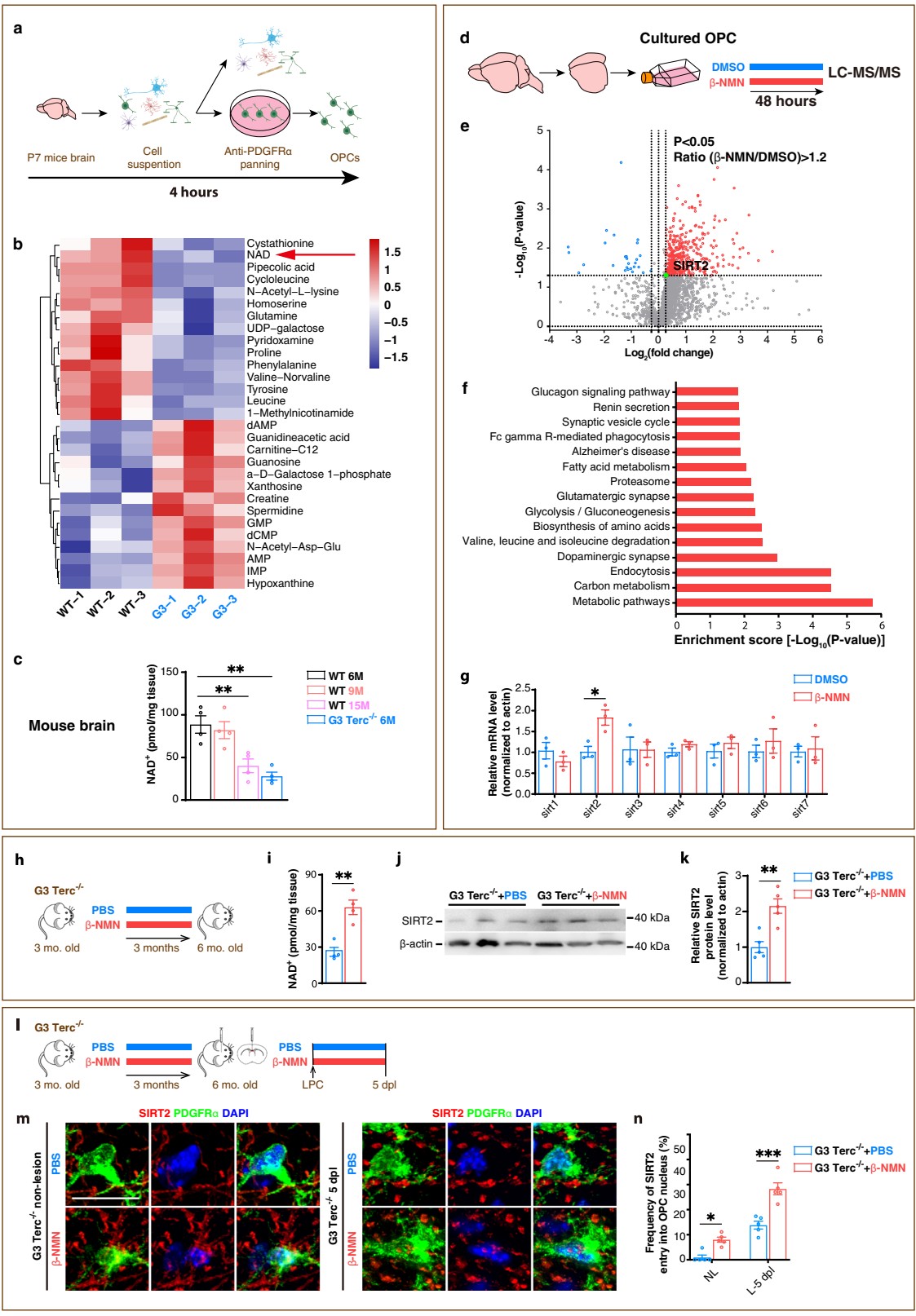

demyelinated region of the corpus callosum under TEM (Fig. 5k, l). We showed that at 21 dpl, NAD$^+$ repletion by β-NMN injection in G3 Terc$^{-/-}$ mice doubled the frequency of remyelinated axons (Fig. 5m), increased the frequency of normal myelin (grade 0) and generally decreased the myelin pathology level in comparison with G3 Terc$^{-/-}$ control mice (grade 1–4,

Fig. 5n). These results confirmed that NAD$^+$ repletion enhances remyelination efficiency in G3 Terc$^{-/-}$ mice. Furthermore, NAD$^+$ repletion also decreased both the G-Ratio (Fig. 5o–p) and the distance between DL (Fig. 5q, r), indicating that NAD$^+$ not only increases the thickness of newly-formed myelin but also makes it more compact.

**Fig. 3 Elevating NAD$^+$ by β-NMN enhances SIRT2 nuclear entry in OPCs of the aged mice within demyelination lesion in vivo. a** Schematic of OPC immunopanning protocol. **b** Heatmap of top 29 metabolites significantly different between the OPCs of WT and G3 Terc$^{-/-}$ mice ($n = 3$). **c** NAD$^+$ levels in brain of WT and G3 Terc$^{-/-}$ mice ($n = 4$). **d, e** Volcano plot of 2585 proteins that are differentially expressed between rat OPCs treated with β-NMN or DMSO for 48 h ($n = 3$). β-NMN, 1 mM. **f** KEGG enrichment analysis of proteins upregulated by β-NMN in OPCs. **g** Relative mRNA levels of sirtuins in OPCs treated with β-NMN or DMSO for 48 h ($n = 3$). **h** β-NMN was once daily i.p. injected to G3 Terc$^{-/-}$ mice for 3 months. **i** NAD$^+$ levels in brain of G3 Terc$^{-/-}$ mice ($n = 4$). **j, k** Relative SIRT2 protein level in brains of G3 Terc$^{-/-}$ mice ($n = 5$). **l–n** Immunofluorescence and quantification of SIRT2$^+$ OPCs in corpus callosum of G3 Terc$^{-/-}$ mice with or without demyelination ($n = 5$). NL, non-lesion, L, demyelination lesion induced by LPC at 5 dpl. Scale bar, 10 μm. All data are presented as mean ± SEM. *$p < 0.05$, **$p < 0.01$, ***$p < 0.001$ by two-tailed $t$-test (**g, i, k**), one-way ANOVA followed by Tukey's post hoc test (**c**) or two-way repeated ANOVA followed by Sidak's post hoc test (**n**). In all instances ***$p < 0.001$. *n.s.* no significance. In (**c**), **$p = 0.008$ (WT 6 M vs. WT 15 M), **$p = 0.001$ (WT 6 M vs. G3 Terc$^{-/-}$); in (**g**), *$p = 0.03$; in (**i**), **$p = 0.0037$; in (**k**), **$p = 0.002$; in (**n**), *$p = 0.04$ (NL).

To evaluate whether the improved myelin ultrastructure could translate into the recovered signal conduction of the relevant axons in G3 Terc$^{-/-}$ mice, we next tested the effect of NAD$^+$ repletion on nerve signal conduction. Using acute brain slices, we stimulated one side of the corpus callosum and recorded at the other side the compound action potential (CAP) with two clearly separated peaks, corresponding to the activation of myelinated and unmyelinated axons, respectively (Fig. 5s–u). We found that NAD$^+$ supplementation increased the proportion of effective conduction of G3 Terc$^{-/-}$ mice from 40% to 92%, to the level of the WT young mice (Fig. 5v). The CAP amplitude ratio of myelinated axons (Amp 1) to unmyelinated axons (Amp 2) represents the dominant level of myelinated axons[40]. These results showed that the ratio of Amp 1/Amp 2 of G3 Terc$^{-/-}$ mice was lower than that of WT young mice. Importantly, NAD$^+$ supplementation increased the Amp 1/Amp 2 ratio of G3 Terc$^{-/-}$ mice to the level of WT young mice (Fig. 5w), whereas NAD$^+$ repletion did not change either fast or slow conduction velocity in all groups tested (Fig. 5x–y). These results validated that NAD$^+$ repletion indeed functionally rejuvenates remyelination in G3 Terc$^{-/-}$ mice.

Having proven that long-term pretreatment with NAD$^+$ enhances remyelination efficiency in demyelinated G3 Terc$^{-/-}$ mice, we wondered whether immediate NAD$^+$ supplementation can also enhance proliferation or differentiation of OPCs in G3 Terc$^{-/-}$ mice in vivo. To test this, we started a regime of daily i.p. injection of β-NMN and a focal induction of demyelination simultaneously (referred to as immediate NAD$^+$ supplementation). At 5 dpl (Supplementary Fig. 12a, b), the immediate NAD$^+$ supplementation did not affect the density of the oligodendrocyte lineage cells (Supplementary Fig. 12c), or the proliferating OPCs (Supplementary Fig. 12d) and the proportion of proliferating OPCs (Supplementary Fig. 12e). At 10 dpl (Supplementary Fig. 12f, g), immediate NAD$^+$ supplementation did not change the density of the oligodendrocyte lineage cells (Supplementary Fig. 12h), but increased the density and proportion of differentiated oligodendrocytes by 140% (Supplementary Fig. 12i) and 80% (Supplementary Fig. 12j) respectively.

To test whether immediate or delayed NAD$^+$ supplementation enhances remyelination efficiency in demyelinated G3 Terc$^{-/-}$ mice in vivo, immediate NAD$^+$ administration and lesion started simultaneously while delayed NAD$^+$ administration started at 3 dpl when demyelination had been fully established (Supplementary Fig. 12k). Both immediate and delayed NAD$^+$ supplementation exerted almost identical effects in all indexes tested. NAD$^+$ supplementation significantly increased both the proportion of remyelinated axons (Supplementary Fig. 12l, m) and the frequency of normal myelin (Supplementary Fig. 12n), indicating that NAD$^+$ enhances remyelination and alleviates myelin pathology. Meanwhile, NAD$^+$ increased the thickness of myelin (Supplementary Fig. 12o, p) while simultaneously decreased the distance between DL (Supplementary Fig. 12q, r), indicating that NAD$^+$ makes the newly formed myelin thicker and more compact and therefore higher quality. Together, these

results show that NAD$^+$ supplementation by long-term pretreatment, immediate or delayed administration, significantly enhances remyelination efficiency in demyelinated G3 Terc$^{-/-}$ mice in vivo.

**The enhancing effect of β-NMN on remyelination of the mice requires SIRT2.** As re-expression of SIRT2 and restoring nuclear entry of SIRT2 in the aged OPCs induced by β-NMN is sufficient to rejuvenate the aged OPCs, we further queried whether SIRT2 is required for NAD$^+$ to take effect on OPCs. Using primary cultured mouse OPCs, we found that the effect of NAD$^+$ on OPC differentiation was diminished by a SIRT2 inhibitor, thiomyristoyl (TM), in a dose-dependent manner, and was fully blocked at doses over 5 μM. (Fig. 6a). In primary OPC cultured from SIRT2$^{-/-}$ mice, SIRT2 knockout completely abolished the effect of NAD$^+$ on both the proliferation (Fig. 6b, c) and the differentiation (Fig. 6d, e) of OPCs. The in vitro pharmacological inhibition and genetic deletion data indicated that for NAD$^+$ to enhance the differentiation of OPCs requires SIRT2.

To further validate the role of SIRT2 in the enhancing effect of NAD$^+$ on remyelination in vivo, we applied long-term NAD$^+$ supplementation in SIRT2$^{-/-}$ mice and WT young mice with the same age, both of which received LPC injection-induced demyelination. We found long-term NAD$^+$ supplementation did not affect the proliferation of OPCs in both SIRT2$^{-/-}$ mice and age-matched WT young (6 M) mice (Fig. 6f–j). Interestingly, in WT young mice, long-term NAD$^+$ supplementation promoted the differentiation of OPCs (Fig. 6k–o) and improved the myelin quality (Fig. 6p–w). By contrast, in SIRT2$^{-/-}$ mice, long-term NAD$^+$ supplementation did not affect the impaired capacity of OPCs on proliferation (Fig. 6f–j), differentiation (Fig. 6k–o) or remyelination efficiency (Fig. 6p–w). Loss of SIRT2 in mice completely abolished any changes in all indexes we observed in comparison with age-matched young mice. Collectively, our in vitro and in vivo data confirmed that the effect of NAD$^+$ on the differentiation of OPCs and remyelination requires SIRT2.

**β-NMN induces the nuclear entry of SIRT2 and suppresses transcription of ID4, thus promoting OPC differentiation in vitro.** To further unveil the molecular mechanism underlying the effect of NAD$^+$ on OPCs, we showed that β-NMN treatment increased the nuclear proportion of SIRT2 in cultured mouse OPCs (Fig. 7a–c). The fact that β-NMN promoted nuclear entry of SIRT2 was validated in OLN93 cells by cellular fractionation and Western blot (Fig. 7d). These in vitro results recapitulate the in vivo phenotype that NAD$^+$ induces SIRT2 nuclear localisation in OPCs. SIRT2 exerts its function through deacetylating several histone sites such as H3K18 and H3K56, which had been identified previously[41,42]. We detected the acetylation level of these histone acetylation sites and found that a SIRT2 knockdown increased the acetylation levels of both H3K18 and tubulin, consistent with a previous finding that SIRT2-dependent histone H3K18 deacetylation occurred in bacterial infections[42].

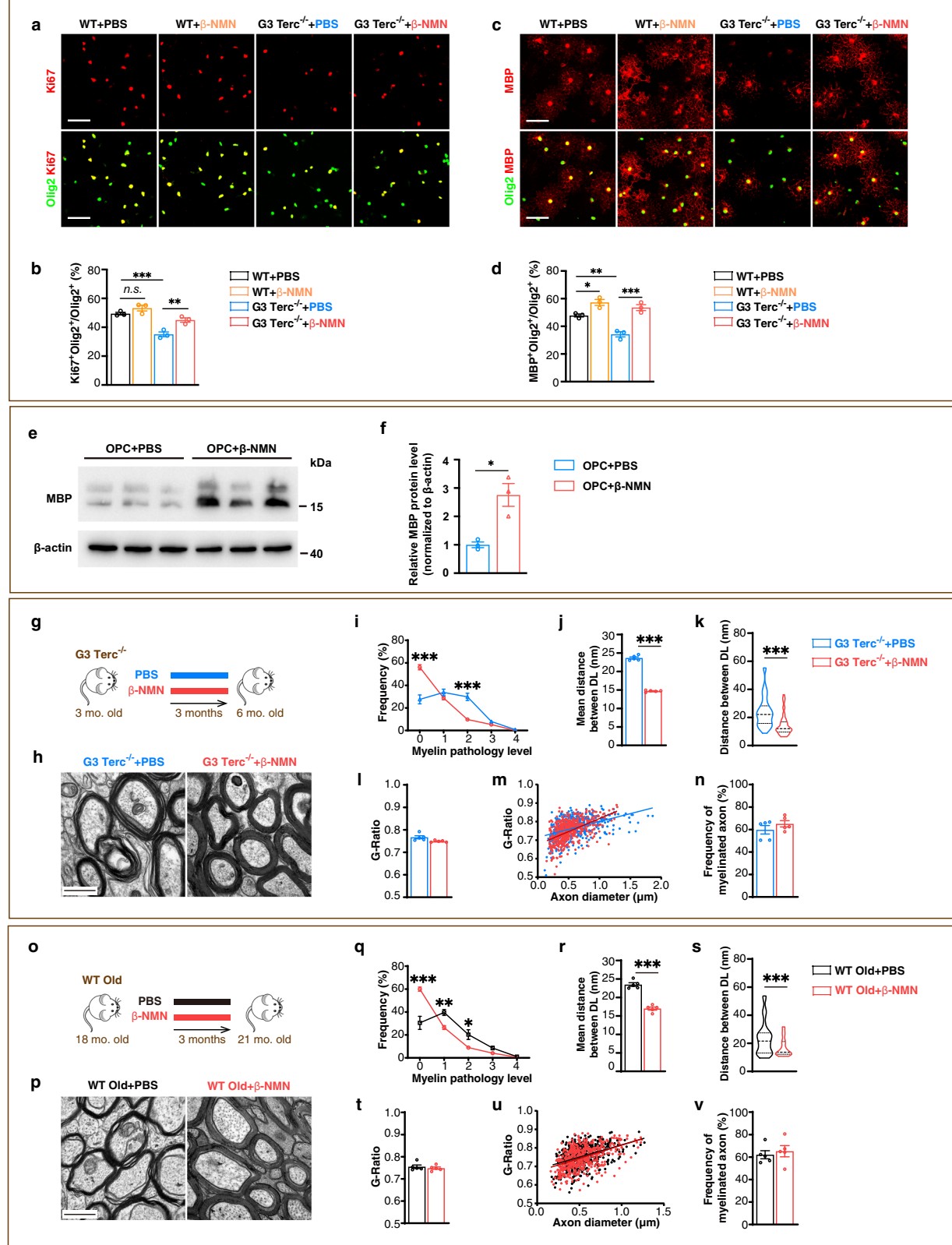

In contrast, the acetylation level of H3K56 remained unchanged (Fig. 7e).

ID4 has been identified as a transcription factor which inhibits the differentiation of OPCs by repressing the transcription of MBP[43,44]. The mRNA level of ID4 was elevated to 6 folds in OPCs of G3 Terc$^{-/-}$ mice compared to OPCs of WT mice. Importantly, in the OPCs of G3 Terc$^{-/-}$ mice, β-NMN treatment

decreased the elevated mRNA level of ID4 to less than half (Fig. 7 f) but did not change the mRNA level of ID2 (Supplementary Fig. 13a). In OPCs of WT mice, β-NMN also downregulated the protein level of ID4 (Fig. 7g–h), which was inversely correlated with β-NMN induced upregulation of MBP (Fig. 4e–f). Since we have established that impaired differentiation of OPCs is the key pathology for SIRT2$^{-/-}$ mice (Fig. 2),

**Fig. 4 Elevating NAD$^+$ by β-NMN delays myelin ageing in the aged mice in vivo. a–d** Images and quantification of proliferating OPCs (Olig2$^+$Ki67$^+$, $n = 3$) cultured for 36 h and differentiated oligodendrocytes (Olig2$^+$MBP$^+$, $n = 3$) cultured for 48 h. β-NMN, 1 mM. Scale bar, 50 μm. **e, f** Western blots of MBP in primary cultured rat OPCs treated with PBS or β-NMN for 48 h ($n = 3$). **g** β-NMN was once daily i.p. injected to G3 Terc$^{-/-}$ mice for 3 months. **h** TEM micrographs of myelin in corpus callosum of the brains of G3 Terc$^{-/-}$ mice. Scale bar: 500 nm. **i–n** Quantification of myelin pathology level (**i**), distance between DL (**j, k**), G-Ratio average (**l**), individual G-Ratio distribution (**m**, linear regression) and myelinated axons (**n**) in the corpus callosum of G3 Terc$^{-/-}$ mice brain ($n = 5$). **o** β-NMN was once daily i.p. injected to WT old mice for 3 months. **p** TEM micrographs of myelin in corpus callosum of the brains of WT old mice. Scale bar, 500 nm. **q–v** Quantification of myelin pathology level (**q**), distance between DL (**r, s**), G-Ratio average (**t**), individual G-Ratio distribution (**u**, linear regression) and myelinated axons (**v**), in the corpus callosum of WT old mice brain ($n = 5$). All data are presented as mean ± SEM. The center, upper and lower line represent the median, upper and lower quartiles, respectively (**k, s**). *$p < 0.05$, **$p < 0.01$, ***$p < 0.001$ by two-tailed $t$-test (**f, j–l, n, r–t, v**) or two-way repeated ANOVA followed by Sidak's post hoc test (**i, q**). In all instances ***$p < 0.001$. *n.s.* no significance. In (**b**), **$p = 0.009$ (G3 Terc$^{-/-}$+PBS vs. G3 Terc$^{-/-}$+β-NMN); In (**d**), *$p = 0.03$ (WT + PBS vs. WT + β-NMN), **$p = 0.004$ (WT + PBS vs. G3 Terc$^{-/-}$+PBS); In (**f**), *$p = 0.01$; In (**q**), **$p = 0.005$ (grade 1), *$p = 0.02$ (grade 2).

we asked whether ID4 is elevated in SIRT2$^{-/-}$ mice. Indeed, the mRNA level of ID4 increased sharply in the primary cultured OPCs of SIRT2$^{-/-}$ mice (Fig. 7i). Consistently, the mRNA level of ID4 in the brain of SIRT2$^{-/-}$ mice was higher than that of WT mice (Fig. 7j). Using chromatin immunoprecipitation-quantitative PCR (ChIP-qPCR) analysis (Fig. 7k), we found that β-NMN increased SIRT2 binding to the promoter of ID4 (Fig. 7 l), but not to ID2 (Supplementary Fig. 13b), whereas β-NMN decreased H3K18Ac binding to the promoter of ID4 (Fig. 7m). These data confirmed that SIRT2 inhibits the transcription of ID4 by deacetylating H3K18.

Based on our results, we propose a working model of NAD$^+$ effecting on OPCs: aged OPCs exhibit depleted SIRT2 nuclear localisation and have lower NAD$^+$. Supplementation of β-NMN elevates the NAD$^+$ level, restores nuclear entry of SIRT2 in OPCs and deacetylates H3K18, thus suppressing the transcription of ID4 and further promotes OPC differentiation. NAD$^+$ repletion restores SIRT2 nuclear entry in OPCs, promotes OPCs differentiating into mature oligodendrocytes, and eventually delays myelin ageing in the normal CNS and enhances myelin repair in the demyelinated aged CNS (Fig. 8).

## Discussion

The present study identifies SIRT2 as a molecular target for OPC. Werner et al. and Li et al. reported that SIRT2 is expressed throughout the oligodendrocyte lineage cells, and exclusively in mature oligodendrocytes in the adult CNS[29,30]. The transcriptome database provided by Cahoy et al. and Zhang et al. also revealed that the mRNA of SIRT2 is enriched in oligodendrocytes lineage cells[45,46]. In this work, we showed that the *sirt2* mRNA in cultured OPCs was overwhelmingly dominant over any of the other six members within the sirtuin family, and ranked the second highest among various cell types in the brain, just next to mature oligodendrocytes. Consistent with previous results[29–31], we demonstrated that SIRT2 was indeed exclusively localised to the cytoplasm of mature oligodendrocytes. However, surprisingly, we further revealed that the SIRT2 protein was transiently expressed in the nucleus of OPCs during myelin development while depleted in OPCs of the adult CNS, indicating that SIRT2 is a specific target for oligodendrocyte lineage cells. The switch of SIRT2 localisation from OPCs during development to mature oligodendrocytes in the adult CNS is reserved to mammals and, in particular, primates. Interestingly, in the adult CNS, demyelination induced the re-expression of SIRT2 in the nuclei of OPCs, recapitulating a developmental program. However, such recapitulation was impaired in the aged CNS, corelating with impaired remyelination in the aged CNS[47].

Given that nuclear SIRT2 expression occurred during myelin postnatal development, and this program was recapitulated within the demyelination lesion in the adult CNS, we next explored the function of SIRT2 in OPCs. To clearly investigate the role of SIRT2 specifically in OPCs in vivo, we used NG2-Cre$^{ERT}$; *Sirt2*$^{flox/flox}$ mice, and i.p. injected tamoxifen to exclusively delete SIRT2 in OPCs in a temporally controllable manner. Our results showed that the conditional knockout of SIRT2 in OPCs reduced the frequency of remyelinated axons, decreased the thickness of newly formed myelin, and reduced the frequency of normal-looking newly formed myelin. These results provide direct evidence that SIRT2 in OPCs indeed plays a critical role in remyelination, and have established a causality between the loss of SIRT2 in OPCs and impaired remyelination. Intriguingly, the autoantibody to SIRT2 is prevalent in the cerebrospinal fluid of secondary progressive MS patients[48], whose pathological feature is failed remyelination. In addition, SIRT2 protein levels diminished within the lesion in both the EAE model and in MS patients[49].

Next, we demonstrate that elevating NAD$^+$ with β-NMN delays myelin ageing and enhances myelin repair in the aged CNS. Using G3 Terc$^{-/-}$ mice, a premature ageing OPC model established in this work, we revealed that NAD$^+$ was one of the top depleted metabolites in the aged OPCs. The present study provides a comparison of the metabolic fingerprints of OPCs between young and old, laying a foundation for future studies concerning the metabolism of aged OPCs. Since NAD$^+$ is a regulator of SIRT2 activity[27,50], we supplemented β-NMN, the immediate precursor of NAD$^+$[51], in a dose comparable to the clinical trial in which the long-term clinical safety of β-NMN has been proven[52], and found that NAD$^+$ repletion restored the nuclear entry of SIRT2 in the aged OPCs and delayed myelin ageing. Furthermore, we demonstrated that NAD$^+$ supplementation enhances remyelination efficiency both ultrastructurally and functionally in the aged CNS in both a preventive and therapeutic way; these results highlight the potential for this work to go towards clinical translational studies, bringing hope to potential treatments for the progressive phase of MS, a so far unmet therapeutic challenge[22].

To elucidate the molecular mechanism underlying the effect of NAD$^+$ on OPCs, by using multiple techniques, we unveiled that nuclear SIRT2 deacetylates H3K18, thus making chromatin condensed and repressed. This leads to the decreased transcription of ID4 and further promoting OPC differentiation by disinhibiting the transcription of myelin genes such as MBP. Importantly, SIRT2 is necessary for the effect of NAD$^+$ on OPCs. Interestingly, the localisation switch of SIRT2 is similar to that of ID4, which also switches from the nuclei of OPCs to the cytoplasm of mature oligodendrocytes shown in the previous study[53]. The current work reveals that the effects of β-NMN are mediated through NAD$^+$-SIRT2-H3K18Ac-ID4 sequentially, and that SIRT2 is necessary for NAD$^+$ to rejuvenate the aged OPCs. Nevertheless, the process by which NAD$^+$ causes nuclear entry of SIRT2 in OPC awaits further study. For translational purposes,

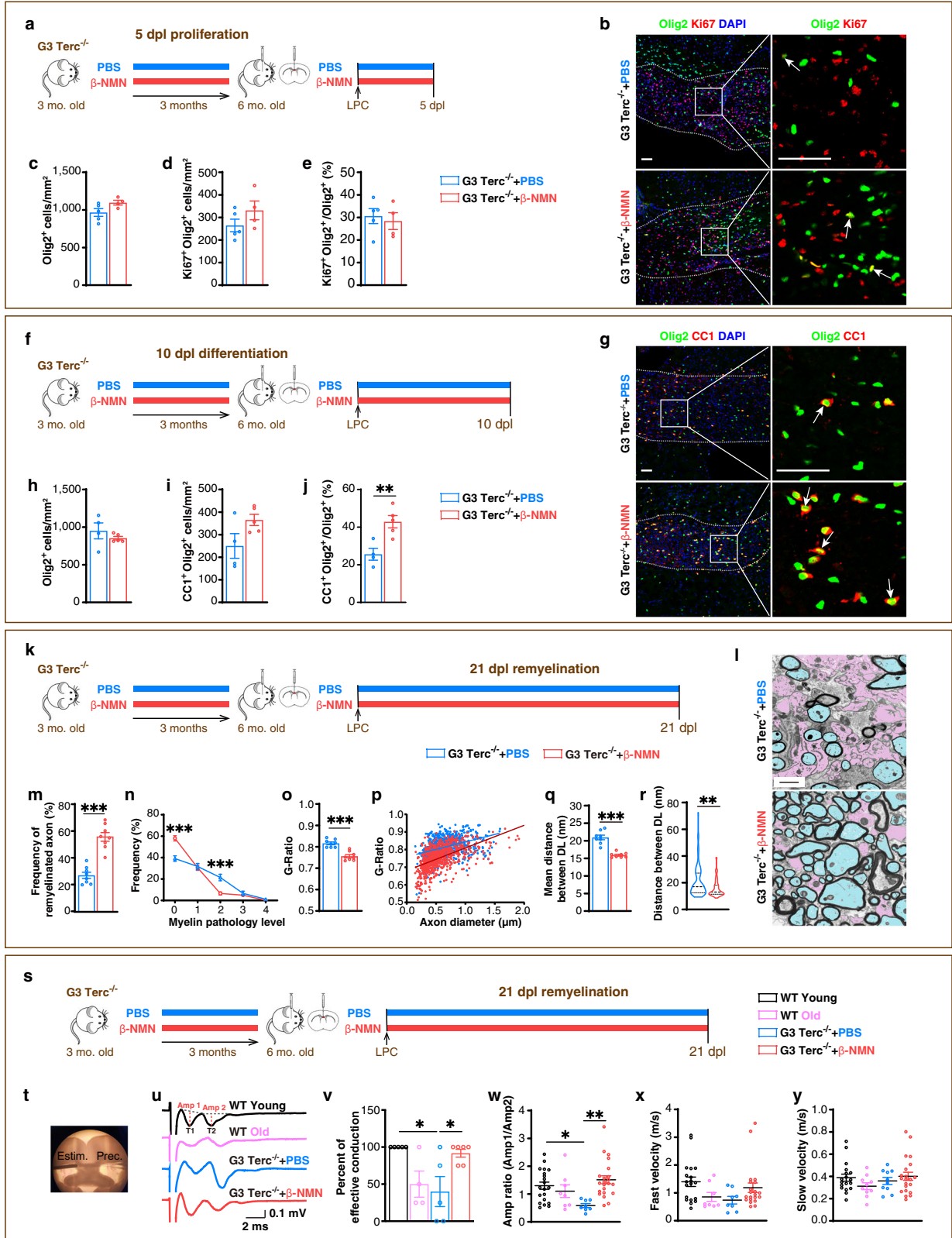

oral administration is desirable in view of the effects of oral administration of β-NMN on OPCs, other neural functions, and other systems need to be systematically examined.

Recent studies have shown that in young mice, supplementing unstable NAD$^+$ decreases the EAE score, correlating with beneficial Th1/Th17 immune responses[54] or helpful CD4$^+$ T cell differentiation[55], and supplying niacin, an indirect NAD$^+$

precursor, modulates macrophages[19], suggesting that NAD$^+$ has versatile targets. Our work clarifies that β-NMN rejuvenates aged OPCs, the central player for myelination and remyelination[12], by restoring the nuclear entry of SIRT2.

Together, we report that the nuclear entry of SIRT2 is depleted and NAD$^+$ is substantially reduced in the aged OPCs. Supplementing β-NMN restores the nuclear entry of SIRT2 and

**Fig. 5 Elevating NAD⁺ by β-NMN enhances remyelination in the aged mice in vivo. a–e** Images and quantification of oligodendrocyte lineage cells and proliferating OPCs within the demyelination lesions at 5 dpl ($n = 5$ for the G3 Terc$^{-/-}$+PBS group, $n = 4$ for the G3 Terc$^{-/-}$+β-NMN group). Scale bar, 50 μm. **f–j** Images and quantification of oligodendrocyte lineage cells and differentiated oligodendrocytes within the demyelination lesions at 10 dpl ($n = 4$ for the G3 Terc$^{-/-}$+PBS group, $n = 5$ for the G3 Terc$^{-/-}$+β-NMN group). Scale bar, 50 μm. **k, l** TEM micrographs within the lesions at 21 dpl. Axons are coloured in pink for demyelinated ones whereas in blue for remyelinated ones. Scale bar, 1 μm. **m–r** Quantification of the frequency of remyelinated axons (**m**), myelin pathology level (**n**), G-Ratio average (**o**), individual G-Ratio distribution (**p**, linear regression) and distance between DL (**q, r**) ($n = 8$). **s** Experiment design for functional remyelination efficiency. **t** 250 μm thick coronal brain slices contain the focal demyelination in corpus callosum. Estim., tungsten bipolar stimulation electrode; Prec., glass recording micropipette (electrode). **u** Representative CAPs recorded in corpus callosum. Fast velocity = δDistance/δT1; Slow velocity = δDistance/δT2; Amp 1, the fast phase amplitude; Amp 2, the slow phase amplitude. **v–y** Quantification of the proportion of effective conduction (**v**), the ratio of Amp 1/Amp 2 (**w**, ratio of myelinated axons to unmyelinated axons), fast velocity (**x**, conduction velocity of myelinated axons) and slow velocity (**y**, conduction velocity of unmyelinated axons) of WT young ($n = 5$), WT old ($n = 4$), G3 Terc$^{-/-}$+PBS ($n = 5$) and G3 Terc$^{-/-}$+β-NMN ($n = 6$) mice. All data are presented as mean ± SEM. The center, upper and lower line represent the median, upper and lower quartiles, respectively (**r**). *$p < 0.05$, **$p < 0.01$, ***$p < 0.001$ by two-tailed $t$-test (**c–e, h–j, m, o, q, r**), one-way ANOVA followed by Tukey's post hoc test (**v–y**) or two-way repeated ANOVA followed by Sidak's post hoc test (**n**). In all instances ***$p < 0.001$. *n.s.* no significance. In (**j**), **$p = 0.0074$; In (**r**), **$p = 0.001$; In (**v**), *$p = 0.02$ (WT Young vs. G3 Terc$^{-/-}$+PBS), *$p = 0.04$ (G3 Terc$^{-/-}$+β-NMN vs. G3 Terc$^{-/-}$+PBS); In (**w**), *$p = 0.03$ (WT Young vs. G3 Terc$^{-/-}$+PBS), **$p = 0.002$ (G3 Terc$^{-/-}$+β-NMN vs. G3 Terc$^{-/-}$+PBS).

rejuvenates aged OPCs by promoting their differentiation into mature oligodendrocytes and eventually enhances new myelin generation in the aged CNS. Mechanistically, the effects of β-NMN are mediated through NAD⁺-SIRT2-H3K18Ac-ID4 sequentially, and SIRT2 is necessary. The present study identifies SIRT2 as a molecular target for rejuvenating the aged OPCs, and by restoring nuclear entry of SIRT2 in the aged OPCs, β-NMN delays myelin ageing in normal and enhances remyelination in demyelinated aged CNS, paving the way for future clinical translation study.

## Methods

**Human brain tissue.** Human brain tissue was provided by National Health and Disease Human Brain Tissue Resource Center (http://zjubrainbank.zju.edu.cn) and studies were approved by Medical Ethics Review Committee of Zhejiang University School of Medicine (ETHICS number: 2018-009). The sample was anonymously coded in accordance with local ethical guidelines, and written informed consent was obtained. Patient was male aged 53 years.

**Marmoset brain tissue.** The marmoset brain tissue was from Lixia Gao's lab. One was collected from a P3 healthy male Marmoset and another was collected from an 8 years old healthy male Marmoset. All studies with marmosets were approved by the Experimental Animal Welfare Ethics Review Committee of Zhejiang University (ETHICS number: ZJU20190079).

**Mice and diet.** C57BL/6, G3 Terc$^{-/-}$ mice[56], SIRT2$^{-/-}$ mice[57], Sirt2$^{flox/flox}$ mice and NG2-Cre$^{ERT34}$ mice were used in this study. They are male mice of different ages (3 M 6 M 18 M 21 M) on a C57BL/6 background. The C57BL/6 mice were purchased from the Beijing Vital River Laboratory Animal Technology Co., Ltd. (Beijing China). The G3 Terc$^{-/-}$ mice were from the Zhenyu Ju laboratory and genotyped using PCR reactions with three primers to amplify the WT and Knockout alleles (1, 5′-TTCTGACCCACCACCAACTTCAAT-3′, 2, 5′-GGGGCTG CTAAAGCGCAT-3′, 3, 5′-CTAAGCCGGCACTCCTTACAAG-3′). The sizes of the PCR products of WT and Knockout alleles are 220 bp and 180 bp, respectively. The SIRT2$^{-/-}$ mice were from the Houzao Chen laboratory and genotyped using PCR reactions with three primers to amplify the WT and Knockout alleles (1, 5′-GACTGGAAGTGATCAAAGCTC-3′, 2, 5′-CAGGGTCTCACGAGTCTCATG-3′, 3, 5′-TCAAATCTGGCCAGAACTTCAG-3′). The sizes of the PCR products of WT and Knockout alleles are 538 bp and 700 bp, respectively. The Sirt2$^{flox/flox}$ mice were purchased from Shanghai Model Organisms (#NM-CKO-190038, Shanghai, China) and genotyped using PCR reactions with two primers to amplify the WT and Sirt2$^{flox}$ alleles (1, 5′-TGCTACCAAACCTAATGACCTGAG-3′, 2, 5′-CCATC CATCTATCTACCCACCATC-3′). The sizes of the PCR products of WT and Sirt2$^{flox}$ alleles are 214 bp and 364 bp, respectively. The NG2-Cre$^{ERT}$ mice were from the Chong Liu laboratory and genotyped using PCR reactions with three primers to amplify the WT and Knockout alleles (1, 5′-GATGTGAATAAAAG GCGACATTC-3′, 2, 5′-TGTATTATTTTTCCATACTAGATGTCCA-3′, 3, 5′-CTG AACGGGCAGATCAACAT-3′). The sizes of the PCR products of WT and Knockout alleles are 261 bp and 220 bp, respectively. NG2-Cre$^{ERT}$; Sirt2$^{flox/flox}$ mice were obtained by crossing the Sirt2$^{flox/flox}$ mice with the NG2-Cre$^{ERT}$ mice.

All mice were housed in an environment of suitable temperature and humidity with ad libitum access to water and food (25 °C, suitable humidity (typically 50%), 12 h dark/light cycle) at the Zhejiang University Animal Experimental Center. All mice appeared healthy, received regular monitoring from animal care staff and were not involved in prior procedures or testing. For β-NMN treated mice, β-NMN (#BTO5, Bontac) were dissolved in PBS and intraperitoneally injected daily (10 mg/kg bodyweight). The NG2-Cre$^{ERT}$; Sirt2$^{flox/flox}$ mice were intraperitoneally injected with tamoxifen (Sigma, #T5648, 100 mg/kg) for 5 days[34–36] so that the Cre recombinase-mediated excision creates a nonfunctional allele exclusively in OPCs in a temporally-controlled manner, and the tamoxifen was dissolved in a 9:1 ratio of corn oil to ethanol. Four days post the last administration of tamoxifen[35,36], demyelination was induced by focal injections of LPC into the corpus callosum and detected remyelination at 21 dpl with TEM. All mice were randomly allocated to experimental groups. All studies with mice were approved by the Experimental Animal Welfare Ethics Review Committee of Zhejiang University (ETHICS number: 14660).

**Primary culture of OPCs, microglia and astrocytes.** For the rat OPC culture, two newborn SD rat cerebral cortices were dissected out in ice-cold HBSS (#14025092, Gibco), then the tissues were minced and digested for 12 min (min) at 37 °C in 0.25% trypsin (#25200056, Gibco). The digestion was terminated with DMEM/F12 (#11320082, Gibco) containing a 10% foetal bovine serum (FBS, #10099141, Gibco). Then the tissue was blown into a single-cell suspension with a pipette and the cells were collected by centrifuging at 500 g for 5 min at 4 °C. The cells were then plated in a 75 cm² tissue culture flask which was pre-coated with poly-D-lysine (PDL, #P0899, Sigma, 0.1 mg/mL). Cells were maintained in DMEM/F12 containing 10% FBS and incubated at 37 °C with 5% CO₂. The medium was then completely changed at day 2, day 5, day 8. At day 8, the flask was shaken on a shaker at 200 rpm for 2 h at 37 °C and the medium was centrifuged at 300 g for 5 min to collect microglia. Then fresh medium was added into the flask. For OPC purification, the flask was shaken at 250 rpm for 16–18 h and the medium was centrifuged at 200 g for 5 min after which the OPCs were collected. The firmly-attached cells at the bottom of the flask are astrocytes. The astrocytes were collected by 2 min digestion in 0.25% trypsin at 37 °C with a 5 min centrifugation at 300 g. After purification, microglia and astrocytes were seeded onto dishes and cultured in DMEM/F12 containing 10% FBS. OPCs were seeded onto coverslips or dishes coated with PDL and cultured in DMEM/F12 containing 10% FBS. After 3–4 h culture, the medium of OPCs was changed to Neurobasal™ (#21103049, Gibco) medium containing B27™ Supplement (#17504044, Gibco) and N2 Supplement (#17502001, Gibco). All studies with rats were approved by the Experimental Animal Welfare Ethics Review Committee of Zhejiang University (ETHICS number: 14660).

For mouse OPC culture, four newborn mice cerebral cortices were dissected out in ice-cold HBSS. Then the tissues were minced into pieces and blown into cell suspension with a 1 mL pipette. The cell suspension was then transferred into a 75 cm² tissue culture flask which was pre-coated with PDL and maintained in DMEM/F12 containing 20% FBS and incubated at 37 °C, 5% CO₂. The medium was completely changed at day 3, day 5, day 7. The next steps are the same as those for the rat OPC cultures. For β-NMN treatment, OPCs were treated with 1 mM β-NMN, and β-NMN was dissolved in PBS.

**Primary culture of neuron.** For the rat neuron culture, the hippocamps from embryonic day 18 rats were dissected out in ice-cold HBSS and digested for 12 min at 37 °C in 0.125% trypsin. The digestion was terminated with DMEM (#11965092, Gibco) containing 10% F12 (#11765062, Gibco) and 10% FBS (#10099141, Gibco). Then the tissue was gently pipetted into a single-cell suspension with a pipette (1 mL) and left still for 2 min. The single-cell suspension was then transferred to a new 1.5 mL tube and centrifuged at 200 g for 5 min at 4 °C. The cells were resuspended in the Neurobasal™ medium containing 2% B27™ Supplement and 1% Glutamax (#35050061, Gibco) and seeded onto dishes that were pre-coated with PDL. 1.25 μM cytarabine was added to inhibit the growth of glial cells.

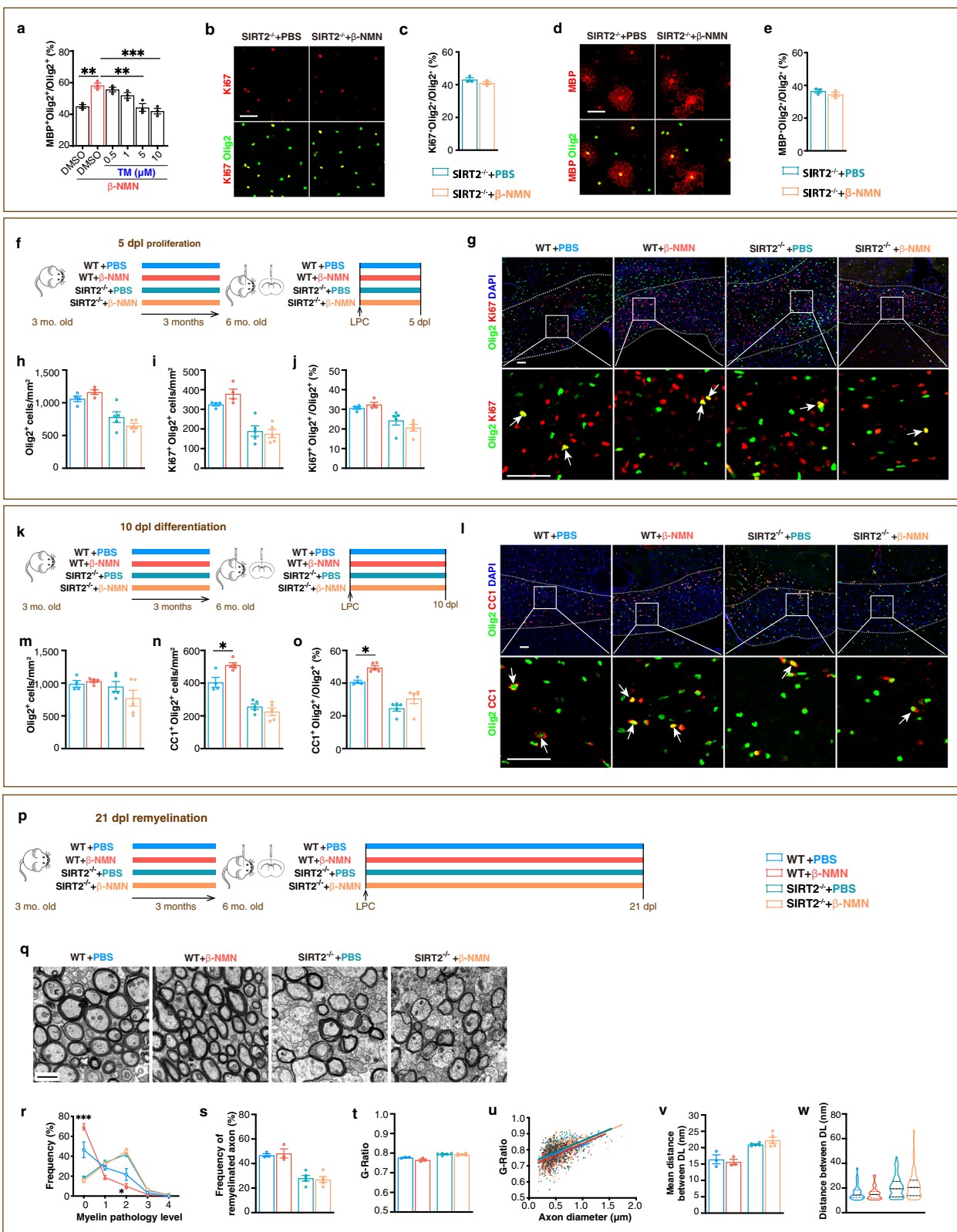

**Cell lines and cell culture**. HEK293T (Human embryonic kidney) and OLN93 (Rat oligodendrocyte) cell lines were used in this study. Both of these two cell lines were cultured in DMEM (#11965092, Gibco) with 10% FBS (#04-001-1ACS, Biological Industries) and 1% Penicillin/Streptomycin (#15140163, Gibco) at 37 °C with 5% $CO_2$. HEK293T cells were purchased from the American Type Culture Collection (ATCC). The OLN93 cells were gifted by the Hui Fu laboratory (Wuhan University).

**Immunofluorescence for cells**. Cultured cells were rinsed with PBS three times and fixed with 4% paraformaldehyde (PFA) for 10 min at room temperature (RT). Then the cells were washed three times with PBS and blocked with 5% normal donkey serum (NDS, #D9663, Sigma), 0.3% Triton X-100 (#T8787, Sigma) in PBS for 2 h at 4 °C. The cells were then incubated with primary antibodies (diluted with 2.5% NDS, 0.3% Triton X-100 in PBS) for 4 h at RT and then washed with PBS for three times. Primary antibodies used in this study include: rat anti-Ki67 (#14-5698-82, Thermo

**Fig. 6 The enhancing effect of β-NMN on remyelination of the mice requires SIRT2. a** Quantification of differentiated oligodendrocytes from P0 WT mice cultured for 48 h ($n = 3$). TM, Thiomyristoyl. **b–e** Images and quantification of proliferating OPCs (**b, c**) cultured for 36 h and differentiated oligodendrocytes (**d, e**) cultured for 48 h from P0 SIRT2$^{-/-}$ mice ($n = 3$). Scale bars, 50 μm. **f** Experiment design for testing the impact of β-NMN on OPC proliferation in vivo. **g–j** Images and quantification of oligodendrocyte lineage cells and proliferating OPCs within the demyelination lesions at 5 dpl ($n = 5$). Scale bar, 50 μm. **k** Experiment design for testing the impact of β-NMN on OPC differentiation in vivo. **l–o** Images and quantification of oligodendrocyte lineage cells and differentiated oligodendrocytes within the demyelination lesions at 10 dpl ($n = 5$). Scale bar, 50 μm. **p** Experiment design for testing the impact of β-NMN on remyelination efficiency in vivo at 21 dpl. **q** TEM micrographs within the lesions at 21 dpl. Scale bar, 1 μm. **r–w** Quantification of the proportion of myelin pathology level (**r**), remyelinated axons (**s**), G-Ratio average (**t**), individual G-Ratio distribution (**u**, linear regression) and distance between DL (**v, w**) within the lesions at 21 dpl ($n = 5$). All data are presented as mean ± SEM. The center, upper and lower line represent the median, upper and lower quartiles, respectively (**w**). $*p < 0.05$, $**p < 0.01$, $***p < 0.001$ by two-tailed $t$-test (**c, e**) or one-way ANOVA followed by Tukey's post hoc test (**a, h–j, m–o, s, t, v, w**) or two-way repeated ANOVA followed by Sidak's post hoc test (**r**). In all instances $***p < 0.001$. $n.s.$ no significance. In (**a**), $**p = 0.004$ (DMSO vs. DMSO + β-NMN), $**p = 0.002$ (DMSO + β-NMN vs. β-NMN + 5 μM TM); In (**n**), $*p = 0.02$ (WT + PBS vs. WT + β-NMN); In (**o**), $*p = 0.05$ (WT + PBS vs. WT + β-NMN); In (**r**), $*p = 0.02$ (grade 2, WT + PBS vs. WT + β-NMN).

Fisher Scientific, 1:500), rat anti-MBP (#MCA409S, Bio-Rad Laboratories, 1:500), rabbit anti-Olig2 (#AB9610, Millipore, 1:400). Then the cells were incubated with secondary antibodies (diluted with 2.5% NDS, 0.3% Triton X-100 in PBS) for 2 h at 4 °C and then rinsed with PBS for three times. Secondary antibodies used in this study include: Cy$^{TM}$3 affinipure donkey anti-rat IgG (H + L) (#712-165-153, Jackson ImmunoResearch, 1:400), Alexa Fluor® 488 affinipure donkey anti-rabbit IgG (H + L) (#711-545-152, Jackson ImmunoResearch, 1:400). For nuclei staining, the cells were incubated with DAPI (#D9542, Sigma, 0.1 μg/mL) for 10 min at RT followed by washing with PBS for three times. Then the coverslips were mounted onto glass slide in a drop of Flurosave$^{TM}$ Reagent (#345789, Millipore). Image acquisition was performed using Olympus FV1200 microscope and further image processing and analysis was performed using the ImageJ software.

**Immunofluorescence for tissue sections**. Mice were anesthetised with sodium pentobarbital (50 mg/kg) by intraperitoneal injection and transcardially perfused with 4% PFA in 0.1 M PB (pH 7.4). The brain was removed and post-fixed with 4% PFA for 2 h at 4 °C. Then the brain was quickly harvested and immersed into up-graded sucrose: 10% and 20% for 2 h, 30% in 0.1 M PB until it sank to the bottom. The brain was then embedded in a tissue freezing medium (#03813266, Leica) and frozen at −80 °C. 12 μm coronal slices were cut using a cryostat (Leica CM 1950). Then the slides were put into slides box and stored at −80 °C after dried at RT for 1 h.

For immunofluorescence, the slides were dried at RT for 10 min and washed with PBS (5 min, three times) to wash away the OCT. For antigen-retrieval the slides were placed in a pre-heated citrate buffer (pH 6.0) in a water bath for 15 min (almost boiling). Then the slides were washed with PBS (5 min, three times) and blocked with 5% NDS, 0.3% Triton X-100 in PBS for 2 h at 4 °C. The slides were then incubated with primary antibodies (diluted with 2.5% NDS, 0.3% Triton X-100 in PBS) for 48–72 h at 4 °C. Primary antibodies used in this study include: rat anti-Ki67 (#14-5698-82, Thermo Fisher Scientific, 1:500), rabbit anti-Olig2 (#AB9610, Millipore, 1:400), mouse anti-CC1 (or mouse anti-APC, #ab16794, Abcam, 1:200), Goat anti-PDGFRα (#AF1062, R&D Systems, 1:100), Rat anti-NG2 (a gift from Hao Huang, Hangzhou Normal University, 1:100), Rabbit anti-SIRT2 (#S8447, Sigma, 1:500), Mouse anti-NeuN (#MAB377, Millipore, 1:400), Guinea pig anti-NeuN (#ABN90, Millipore, 1:500), Chicken anti-GFAP (#AB5541, Millipore, 1:500), Isolectin B$_4$ (#L2140, Sigma, 1:500), Mouse anti-iNOS (#610329, BD Biosciences, 1:100), Goat anti-arginase I (#sc-18354, Santa Cruz, 1:100), Rat anti-CD11b (#MCA711, Bio-Rad, 1:200), Rabbit anti-collagen antibody, type IV (#AB756P, Millipore, 1:400), Mouse anti-NF160/200 (#N2912, Sigma, 1:200). The slides were washed with PBS (5 min, three times) and incubated with secondary antibodies (diluted with 2.5% NDS, 0.3% Triton X-100 in PBS) for 2 h at 4 °C. The secondary antibodies used in this study which were purchased from Jackson ImmunoResearch include: Cy$^{TM}$3 affinipure donkey anti-rat IgG (H + L) (#712-165-153, 1:400), Cy$^{TM}$3 affinipure donkey anti-mouse IgG (H + L) (#715-165-151, 1:400), Cy$^{TM}$3 affinipure donkey anti-rabbit IgG (H + L) (#711-165-152, 1:400), Cy$^{TM}$3 affinipure donkey anti-goat IgG (H + L) (#705-165-147, 1:400), Alexa Fluor® 488 affinipure donkey anti-rabbit IgG (H + L) (#711-545-152, 1:400), Alexa Fluor® 488 affinipure donkey anti-rat IgG (H + L) (#712-545-153, 1:400), Alexa Fluor® 488 affinipure donkey anti-mouse IgG (H + L) (#715-545-151, 1:400), Alexa Fluor® 488 affinipure donkey anti-chicken IgG (H + L) (#703-545-155, 1:400), Alexa Fluor® 488 affinipure donkey anti-guinea pig IgG (H + L) (#706-545-148, 1:400), Alexa Fluor® 488 affinipure donkey anti-goat IgG (H + L) (#705-545-147, 1:400), Alexa Fluor® 647 affinipure donkey anti-mouse IgG (H + L) (#715-605-151, 1:400), Alexa Fluor® 488 streptavidin (#016-540-084, 1:400). The slides were washed with PBS (5 min, three times) and incubated with DAPI for 10 min at RT to stain the nuclei and finally washed with PBS (5 min, three times). Then the slides were mounted with coverslips using Fluorsave. Image acquisition was performed using the Olympus FV1200 microscope and further image processing and analysis was performed using the ImageJ software.

**Immunohistochemistry for tissue section**. The slides were dried at RT for 10 min and washed with TBS (Tris-base 12.1 g, NaCl 9 g, ddH$_2$O 1 L, adjust pH to 7.4) for 5 min (three times) to wash away the OCT. Then the slides were quenched with quench solution (10% Methanol, 10% concentrated Hydrogen peroxides in distilled water) for 5 min at RT to destroy the endogenous enzyme activity and washed with TBS (5 min, three times). The slides were next blocked with block solution (5% NDS, 1% BSA, 0.3% Triton X-100 in TBS) at 4 °C for 2 h. The slides were then incubated with primary antibodies (diluted with 2.5% NDS, 0.5% BSA, 0.3% Triton X-100 in TBS) for 12 h at 4 °C. Primary antibodies used in this study include: rabbit anti-SIRT2 (#S8447, Sigma, 1:500). The slides were washed with TBS (10 min, three times) and incubated with secondary antibodies (diluted with 2.5% NDS, 0.5%BSA, 0.3% Triton X-100 in TBS) for 4 h at 4 °C. Secondary antibodies used in this study include: Biotin-SP affinipure donkey anti-rabbit IgG (H + L) (#711-065-152, Jackson ImmunoResearch, 1:400). Next, the slides were washed with TBS (10 min, three times). ABC solution (#PK-6100, Vector Laboratories, 1:200 diluted with 0.3% Triton X-100 in TBS) was prepared during washing and then added to slides for 2 h at 4 °C. The slides were washed with TBS for 10 min (one times) and TNS (Tris-bsae 6.06 g, ddH2O 1 L, adjust pH to 7.4 using concentrated HCl) for 10 min (two times). Then 200 μl DAB solution (#D4293, Sigma) was added to slides and terminated the reaction after appropriate time. The slides were washed with TNS (10 min, three times) and dried in heating oven for several hours. Finally they were dehydrated in up-graded ethanol and cleared in xylene and finally coversliped with DPX. Image acquisition was performed using the Olympus BX53 microscope and further image processing and analysis was performed using the ImageJ software.

**White matter demyelination induction**. Mice were anesthetised with sodium pentobarbital (50 mg/kg) by intraperitoneal injection and demyelination was induced by the focal injection of 2 μL 1% lysolecithin (LPC, #L4129, Sigma) into the middle of the corpus callosum (bregma 0.5 mm or −1 mm for CAP recording). LPC was delivered at a rate of 0.2 μL/min, and the needle remained in position for 10 min after LPC delivery and then was pulled out the slowly. The wound was sutured and the mice were placed on a heating pad until woken up.

**Transmission electron microscopy**. Mice were anesthetised with sodium pentobarbital (50 mg/kg) by intraperitoneal injection and transcardially perfused with 4% glutaraldehyde in 0.1 M PB (pH 7.4). The brain was harvested and immersed in 4% glutaraldehyde at 4 °C for at least 1 week. Then the lesion region of corpus callosum was dissected (1 mm³) and immersed into 4% glutaraldehyde at 4 °C overnight. The samples were first washed with 0.1 M Cacodylic Acid Sodium (CAS) buffer for 10 min at 4 °C for three times and post-fixed in 2% osmium-tetroxide (OsO$_4$) containing 3% potassium hexacyanoferrate trihydrate (K$_3$Fe (CN)$_6$) for 1 h on ice. Then the samples were washed with deionised water for 5 min at 4 °C for four times and incubated in 4% aqueous uranyl acetate for 1 h on ice, and then washed with deionised water for 5 min at RT for four times. Next, the samples were dehydrated in 50%, 70%, 90% and 95% acetone for 15 min and 100% acetone for 30 min (three times), respectively. Then the samples were infiltrated in grades of acetate: Epon embedding mixture (1:3, 1:1, 3:1, each for 2 h) and then embedded in a pure Epon mixture overnight. For polymerisation, the samples were embedded in pure Epon at 45 °C for 12 h and then transferred to 65 °C for 48–72 h. 60 nm ultrathin sections were cut, and double stained with utanyl acetate and lead citrate. Image acquisition was performed with a Tecnai G2 Spirit 120 KV transmission electron microscope.

**Assessment of remyelination**. G-Ratio was measured on transmission electron micrographs using ImageJ software[58]. 70–100 normal myelinated (grade 0) axons were analysed per mouse. The myelin sheath was regarded as a torus, and the areas of the inner and outer circles of the myelin sheath were measured with a freehand tool on ImageJ by tracing the outer surface of each structure, then converted the areas into hypothetical radius (Supplementary Fig. 14). The G-Ratio was calculated

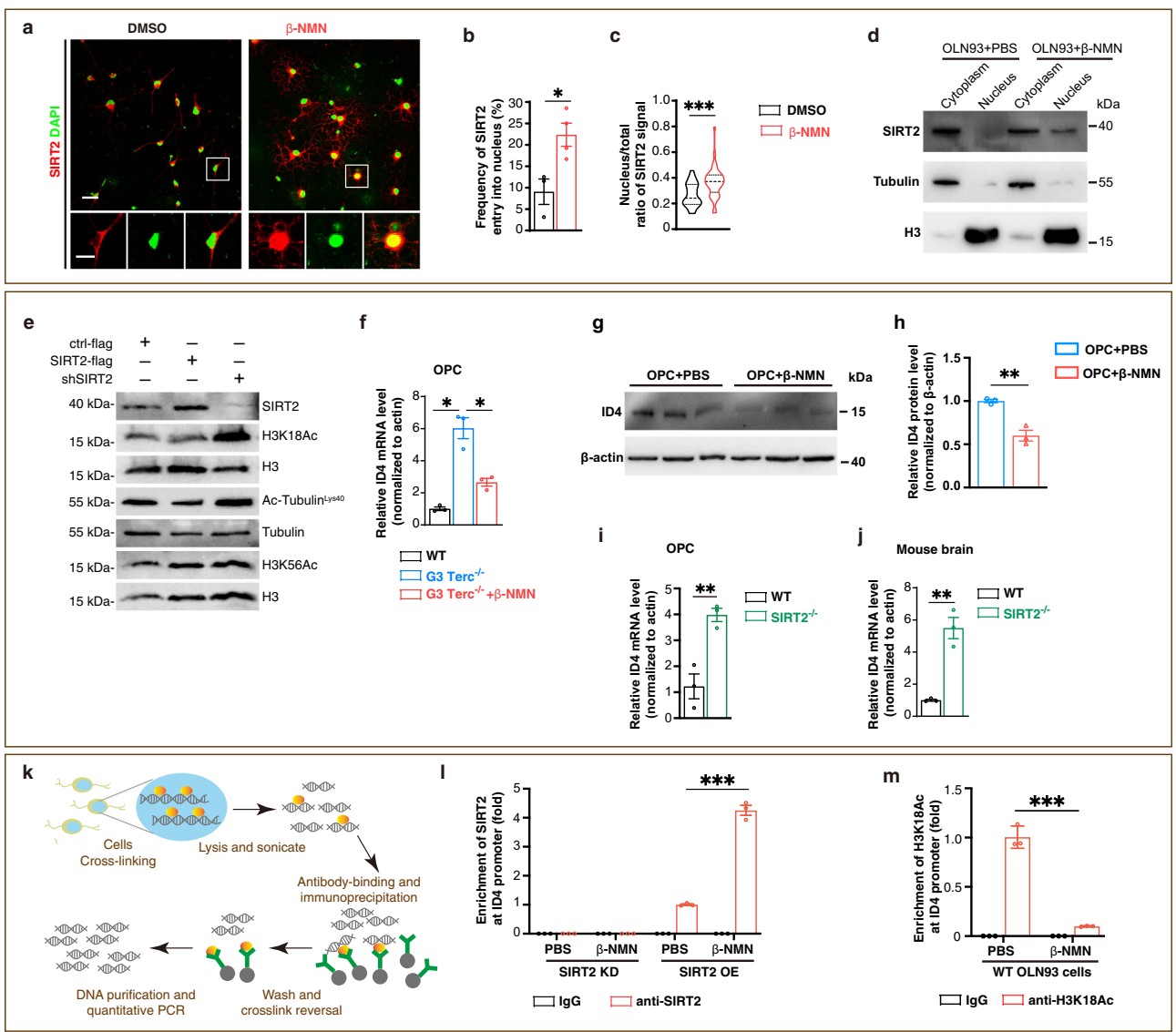

**Fig. 7 β-NMN induces the nuclear entry of SIRT2 and suppresses transcription of ID4, thus promoting OPC differentiation in vitro. a** Images of SIRT2 entry into the nucleus in primary cultured mouse OPCs. SIRT2 (red), DAPI (green). Scale bar, 20 μm for the upper panel, 10 μm for the lower panel. **b** Quantification of the proportion of SIRT2 entry into the nuclei ($n = 3$ for the DMSO group, $n = 4$ for the β-NMN group). **c** Quantification of the nucleus/total ratio of SIRT2 signal ($n = 37$ cells for the DMSO group, $n = 36$ cells for the β-NMN group). **d** Western blot of SIRT2 using nuclear and cytoplasmic fractionation of OLN93 cells treated with β-NMN or PBS. **e** Immunoblotting of SIRT2 and its possible downstream molecular targets (H3K18Ac, Ac-Tubulin$^{Lys40}$ and H3K56Ac, H3 and Tubulin serve as internal reference) regulated by overexpression or knockdown of SIRT2 in OLN93 cell line. ctrl-flag: pCDH-EF1α-MCS-flag-P2A-copGFP, SIRT2-flag: pCDH-EF1α-SIRT2-flag-P2A-copGFP, shSIRT2: pGreenPuro-shSIRT2. **f** Relative mRNA level of ID4 in primary cultured OPCs of WT and G3 Terc$^{-/-}$ mice ($n = 3$). **g, h** Western blots of ID4 in primary cultures rat OPCs treated with DMSO or β-NMN for 48 h ($n = 3$). **i** Relative mRNA level of ID4 in primary cultured OPCs of WT and SIRT2$^{-/-}$ mice ($n = 3$). **j** Relative mRNA level of ID4 in the brains of WT and SIRT2$^{-/-}$ mice ($n = 3$). **k** Schematic of the protocol for ChIP-qPCR. **l** ChIP-qPCR assessment of the enrichment of SIRT2 at the promoter region of ID4 in OLN93 cell line knocking down or overexpressing SIRT2 ($n = 3$). **m** ChIP-qPCR assessment of the enrichment of H3K18Ac at the promoter region of ID4 in WT OLN93 cell line ($n = 3$). All data are presented as mean ± SEM. The center, upper and lower line represent the median, upper and lower quartiles, respectively (**c**). *$p < 0.05$, **$p < 0.01$, ***$p < 0.001$ by two-tailed $t$-test (**b**, **h–j**), one-way ANOVA followed by Tukey's post hoc test (**f**) or two-way repeated ANOVA followed by Sidak's post hoc test (**l–m**). In all instances ***$p < 0.001$. *n.s.* no significance. In (**b**), *$p = 0.0219$, In (**f**), *$p = 0.04$ (WT vs. G3 Terc$^{-/-}$), *$p = 0.03$ (G3 Terc$^{-/-}$ vs. G3 Terc$^{-/-}$+β-NMN); In (**h**), **$p = 0.03$ (); In (**i**), **$p = 0.007$, In (**j**), **$p = 0.0025$.

as the ratio of the diameter of inner circle over that of the outer circle on the same myelin, which is inversely correlated with myelin thickness. The G-Ratio of unmyelinated and demyelinated axons is 1, which is the maximum value of the G-Ratio. The diameter of axons was calculated by area, which was also measured by tracking the outer surface of axons. Axon diameters were used in our scatter plot of axon diameter and G-Ratio, which represent the linear relationship between the G-Ratio and the axon diameter. The distance between DL was calculated by dividing the distance from the outermost DL to the innermost DL by the number of DL. In the mouse corpus callosum, the mean diameter of unmyelinated axons is more or less constant, with an overall mean diameter of $0.25 \pm 0.01$ μm[59].

Therefore, only axons with a diameter >0.25 μm were counted as unmyelinated axons. These axons are more likely to be axons without remyelination, rather than axons that were initially unmyelinated.

**Evaluation of axonal pathology.** "Defective axons" was evaluated by: shrunk axons, mitochondria accumulation or swelling, myelin sphere inclusion in axon or in the space between myelin and axon, and axon break. Only the axons with diameters >0.25 μm and were wrapped with myelin were analysed.

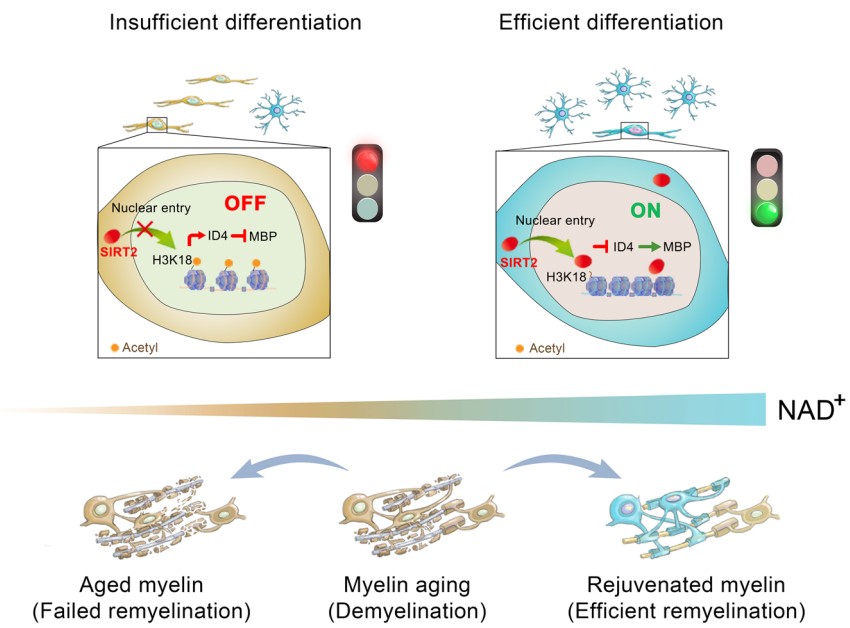

## NAD⁺ enhances SIRT2 nuclear entry and rejuvenates aged OPC

**Fig. 8 A working model for NAD⁺ on rejuvenating the aged OPC.** NAD⁺ targets OPCs and restores SIRT2 nuclear localisation in aged OPCs and delays myelin ageing and enhances myelin repair following demyelination in the aged CNS.

**Immunopanning of OPCs**. As previously described[60], 50 μL BSL-1 (#L-1100, Vector Laboratories) was diluted in 22 mL DPBS, and four plates were coated with BSL-1 solution at RT for at least 4 h, then replaced by DPBS containing 0.2% BSA. Two 100 mm Petri dishes were incubated with 15 mL Tris buffer containing 72 μg goat anti-rat IgG (H + L) (#112-005-167, Jackson ImmunoResearch) at 4 °C overnight. Then, one of the secondary antibody-coated plates was washed three times with DPBS, and incubated with the primary antibody solution, which contained 10 μL rat anti-PDGFRα antibody (#558774, BD Biosciences-Pharmingen, 1:1000), 10 mL DPBS and 0.2% BSA, at RT for at least 2 h.

For tissue dissociation, 7-day-old WT and G3 Terc⁻/⁻ mice brain tissues were dissected and incubated in the papain solution at 37 °C for 45–60 min. A trypsin inhibitor buffer was gently added to stop digestion. The dissociated cell solution was carefully layered over a 4 mL standard trypsin inhibitor buffer in a new 14 mL polystyrene tube. After being centrifuged at 200 g for 15 min, the cell pellet was resuspended with DPBS.

To deplete microglia, the cell suspension was incubated in a BSL-1 coated plate for 15 min at RT. Then, the supernatant was successively incubated in new BSL-1 coated plates for three times. The supernatant was transferred to a new tube, centrifuged at 200 g for 10 min and resuspended with a 6 mL panning buffer (DPBS, 0.02% BSA). To deplete the non-specific binding, cells were transferred to the secondary antibody-coated plate and incubated at RT for 8 min. Then the supernatant was transferred to the primary antibody-coated plate and incubated at RT for 15 min to collect OPCs. The floating cells were washed away with DPBS for 8–10 times. More washing should be performed if there is still significant number of floating cells. The remained cells were collected and stored at –80 °C until the next experiment.

**Corpus callosum compound action potential (CAP) recording**. As described previously, mice were first anesthetised with intraperitoneal injection of sodium pentobarbital (50 mg/kg) and decapitated. Then the brains were dissected out and transferred to an ice-cold slicing solution (2.5 mM KCl, 1.25 mM NaH₂PO₄, 26 mM NaHCO₃, 10 mM Dextrose, 213 mM Sucrose, 2 mM MgSO₄, 2 mM CaCl₂), which was bubbled with mixed gas (95% O₂, 5% CO₂). Coronal slices were cut (250 μm thick, four slices before and after bregma −1.0 mm) using a vibratome (Leica VT1200S). The slices were then transferred to an incubation chamber filled with artificial cerebrospinal fluid (ACSF) (126 mM NaCl, 2.5 mM KCl, 1.25 mM NaH₂PO₄, 26 mM NaHCO₃, 25 mM Dextrose, 2 mM MgSO₄, 2 mM CaCl₂; 315–325 mOsm, pH = 7.2–7.3) and maintained at 34.5 °C for 45 min. After incubation, slices were kept in the same solution at RT and allowed to equilibrate at least 30 min prior to recording.

For electrophysiological recordings, Slices were transferred to a recording chamber perfused at 2 mL/min rate with aerated ACSF at 20.5–22.7 °C. Referring to a previous report[40], a tungsten bipolar electrode was used for stimulation in the

corpus callosum of one hemisphere and a glass electrode (impedance of 1–3 MΩ) filled with ACSF was placed in the contralateral hemisphere for recording. The stimulating pulses of 0.1 ms duration, 1 mA current and 10 s interval were applied via an isolator (ISO-Flex, A.M.P.I). Evoked CAPs were recorded by a MultiClamp 700B amplifier (Molecular Devices) and sampled by a Micro3 1401 (CED, Cambridge Electronic Design) at 25 kHz using Spike2 software (Bessel filter set to 10 kHz) for offline analysis. For each slice, we recorded two trials with the same stimulation site of two and different recording sites in the current clamp mode.

For data analysis and statistics, 100 repeated responses of each trial were averaged for waveform analysis. The time courses of conduction (T1 and T2) were calculated from the onset of stimulus artifact to negative peaks. Conduction velocity was estimated by the difference of two different distances between the stimulating and recording electrodes (δDistance) divided by the difference of corresponding peak latency (δT) in the same slice, that is, Fast velocity = δDistance/δT1; Slow velocity = δDistance/δT2. CAP amplitude was measured as the vertical distance from the local negative peak of two depolarising phases of the CAPs (Amp.1 and Amp.2) to a tangent joining preceding and following positivities. Due to the serious damage of myelin in some brain slices, the effective conduction was measured to reveal the percentage of slices which lack the fast component even with higher intensity of stimulus.

**Targeted metabolic profiling**. 500 μL 80% Methanol (#67-56-1, Millipore) was added to OPCs (10⁵ cells per sample) purified by immunopanning and vortexed fully to extract metabolites. 480 μL supernatant was collected after 15 min centrifugation (13000 g, 4 °C) and dried at SpeedVac Concentrator (ISS110-230, ThermoFisher) for 4 h. The dried metabolites were then reconstituted in 30 μL of 0.03% formic acid in analytical water. After being vortexed and centrifuged, the supernatant was transferred to an HPLC vial for metabolomics analysis. The targeted metabolite profiling was performed using an LC–MS/MS approach. Separation was achieved on an ACQUITY UPLC HSS T3 column (2.1 × 150 mm, 1.8 μm) using an Ultra High Performance Liquid Chromatograph (UHPLC) system (Nexera×2 LC-30A, Shimadzu) with the following gradient: 0–3 min, 1% mobile phase B; 3–15 min, 1–99% B; 15–17 min, 99% B; 17–17.1 min, 99–1% B; 17.1–20 min, 1% B. The mobile phases employed were 0.03% formic acid in water (A) and 0.03% formic acid in acetonitrile (B). The column was maintained at 35 °C and the samples were kept in the autosampler at 4 °C. The flow rate was 0.25 mL/min, and the injection volume was 20 μL. The mass spectrometer was a triple quadrupole mass spectrometer (QTRAP 6500 +, Sciex) with an electrospray ionisation (ESI) source. The sample analysis was performed in multiple reaction monitoring mode. A total of 256 metabolites were monitored with 160 ion transitions in positive mode and 96 ion transitions in negative mode. A chromatogram review and peak area integration were performed using the MultiQuant software v.3.0 (Sciex). The peak area for each detected metabolite was normalised against the

total ion count of that sample, and thus is a fraction of the total detected metabolite content of that sample. Normalised peak areas were used as variables for multivariate and univariate statistical data analyses.

**NAD$^+$ quantification**. NAD$^+$ quantification was performed using NAD$^+$/NADH Quantification Colorimetric Kit (#K337, BioVision) according to the instructions. Briefly, 20 mg brain tissues were washed with cold PBS and homogenised in 400 μL of NADH/NAD Extraction buffer in a micro-centrifuge tube, then centrifuged at 18000 g for 5 min and the supernatant transferred into a new tube and diluted 70 times with NADH/NAD Extraction buffer. For total NAD detection, 50 μL of diluted samples were transferred into a labeled 96-well plate. For NADH detection, the samples were incubated in a 60 °C water bath for 30 min to decompose the NAD$^+$. For a standard curve preparation, 10 μL of 1 nmol/ul NADH standard was diluted with 990 μL NADH/NAD Extraction buffer to generate 10 pmol/μL standard NADH. Then 0, 2, 4, 6, 8, 10 μL of NADH standard were added into 96-well plate to generate a 0, 20, 40, 60, 80, 100 pmol/well standard. 100 μL of reaction mix (98 μL of NAD Cycling Buffer and 2 μL of NAD Cycling Enzyme Mix) was added into each well and incubated at RT for 5 min. Then 10 μL of NADH developer was added into each well and incubated at RT for 1–4 hours. The reactions were stopped by adding 10 μL of stop solution and the plate was read at OD 450 nm. The content of NAD$^+$ was calculated by subtracting the content of NADH from total NAD.

**Western blot analysis**. For tissue lysis, the tissues had been ground in liquid nitrogen and lysed in lysis buffer [50 mM Tris-HCl-pH 7.5, 5 mM EDTA, 1% SDS, 1% NP-40, 0.5% sodium deoxycholate, 3.69% CHAPS, 1% Triton X-100, 50 mM NaCl, 1 mM DTT (#3483-12-3, Biosharp), protease inhibitor (#18518900, Roche), phosphatase inhibitor A (#B15001-A, Biotool) and B (#B15001-B, Biotool), MG132] for 30 min on ice. Then, the mixture was centrifuged at 1000 g for 5 min at 4 °C and the supernatant was stored at −80 °C. For cell lysis, the cells were collected by centrifuged at 300 g for 5 min and lysed in lysis buffer at 4 °C for 30 min. For cellular fractionation, the cytoplasm and the nucleus of OLN93 cells were separated by using the "PARIS™ kit" (ThermoFisher, #AM1921). Protein quantification was implemented using the BCA protein assay kit measured with a microplate reader. Equal amounts of protein (20–30 μg) were loaded mixed with 5X loading buffer (#FD006, Fdbio science) after boiling to 95 °C for 5 min. Proteins were loaded onto a 12% Blot Bis-Tris plus SDS gels (#FD346, Fdbio science) and run at 200 V in an SDS running buffer for 60 min for separation. Proteins were then transferred onto 0.45 μm PVDF membrane (#ISEQ00010, Millipore) in Blot transfer buffer at 300 mA for 75 min on ice. Blots were blocked at RT for 2 h in 5% skim milk blocking buffer in TBS-Tween (TBST). Primary antibodies were incubated overnight at 4 °C on a rocker in 0.1% TBST. Primary antibodies used in this study include: rabbit anti-SIRT2 (#S8447, Sigma, 1:1000), mouse anti-β-actin (#A5441, Sigma, 1:1000), mouse anti-Tubulin (#M1305-2, HUABIO, 1:5000), mouse anti-Ac-Tubulin$^{Lys40}$ (#T7451, Sigma, 1:1000), rabbit anti-H3 (#ab1791, Abcam,1:10000), rabbit anti-H3K18Ac (#39694, Active motif, 1:1000), rabbit anti-H3K56Ac (#39282, Active motif, 1:1000), rat anti-MBP (#MCA409S, Bio-Rad Laboratories, 1:500), rabbit anti-ID4 (#LS-C384049, LSbio, 1:1000). Blots were washed with TBST (10 min, three times) and incubated at RT for 2 h with HRP secondary antibodies in TBST. Secondary antibodies used in this study include: Peroxidase affinipure donkey anti-rabbit IgG (H + L) (#711-035-152, Jackson ImmunoResearch, 1:10000), Peroxidase affinipure donkey anti-mouse IgG (H + L) (#715-035-151, Jackson ImmunoResearch, 1:10000). Blots were then washed with TBST (10 min, three times). HRP blots were developed with Pierce™ ECL Western Blotting Substrate (#32106, Thermo Fisher Scientific) for 2 min and imaged with X-Ray films or ChemiDoc Touch Imaging Syetem (Bio-Rad). For the antibodies incubated in the same blots, after imaging, the blots were stripped with stripping buffer (25 mM Glycine and 1% SDS in ddH2O, pH 2.0) for 20 min at RT to remove antibodies and washed in TBST for 10 min three times. The blots were blocked at RT for 2 h in 5% skim milk blocking buffer in TBST and then incubated with another primary antibody. For protein quantification, densitometry was performed with ImageJ and normalised to β-actin.

**Relative telomere length detection**. Genomic DNA was extracted from 100 μL of venous blood which was drawn from the tail vein of mice using the TIANamp Genomic DNA Kit (#DP304, TIANGEN). Then DNA, with either telomere primers or 36B4 gene primers, and ChamQ Universal SYBR qPCR Master Mix (#Q711, Vazyme) were mixed as instructed by the manufacturer. Fold changes in gene expression were calculated using the delta Ct method in Microsoft Excel. Primer sequences: Tel-F, 5′-CGGTTTGTTTGGGTTTGGGTTTGGGTTTGGG TTTGGGTT-3′, Tel-R, 5′-GGCTTGCCTTACCCTTACCCTTACCCTTACCCTT ACCCT-3′, 36B4-F, 5′-CAGCAAGTGGGAAGGTGTAATCC-3′, 36B4-R: 5′-CC CATTCTATC ATCAACGGGTACAA-3′.

**RNA isolation and qRT-PCR**. RNA was isolated using the FastPure Cell/Tissue Total RNA Isolation Kit (#RC101, Vazyme). All RNA samples were stored at −80 °C until the next experiment. cDNA was generated using the HiScript III 1st Strand cDNA Synthesis Kit (#R312, Vazyme). For qRT-PCR, cDNA, primers and ChamQ Universal SYBR qPCR Master Mix (#Q711, Vazyme) were mixed as

instructed by the manufacturer. Data were exported from the CFX Manager 3.0 and fold changes in gene expression were calculated using the delta-delta Ct method in Microsoft Excel. Primer sequences are listed in Supplementary Data 4.

**Protein mass spectrometry**. OPCs were lysed in RIPA (#P0013B, Beyotime) for 30 min on ice, then centrifuged at 1000 g for 5 min at 4 °C, and the supernatant was stored at −80 °C. The protein was quantified using a BCA protein assay kit (#P0011, Beyotime) measured with microplate reader (SynergyMx M5, Molecular Devices).

For protein precipitation and digestion, proteins were precipitated with acetone and the protein pellet was dried by using a Speedvac for 1−2 min. The pellet was subsequently dissolved in 8 M urea, 100 mM Tris-HCl, pH 8.5. TCEP (5 mM, Thermo Scientific) and iodoacetamide (10 mM, Sigma) for reduction and alkylation were added to the solution and incubated at RT for 30 min, respectively. The protein mixture was diluted four times and digested overnight with Trypsin at 1:50 (w/w) (Promega). The tryptic-digested peptide solution was desalted by using a MonoSpinTM C18 column (GL Science, Tokyo, Japan) and dried with a SpeedVac.

For LC/ tandem MS (MS/MS) analysis of peptide, the peptide mixture was injected twice as technical repeats and analysed by a home-made 35 cm-long pulled-tip analytical column (75 μm ID packed with ReproSil-Pur C18-AQ 1.9 μm resin, Dr. Maisch GmbH). The column was placed in-line with an Easy-nLC 1200 nano HPLC (Thermo Scientific, San Jose, CA) for mass spectrometry analysis. The analytical column temperature was set at 55 °C during the experiments. The mobile phase and elution gradient used for peptide separation were as follows: 0.1% formic acid in water as buffer A and 0.1% formic acid in 80% acetonitrile as buffer B, 0–1 min, 3–8% B; 1–191 min, 8–25% B; 191–219 min, 25–50% B, 219–220 min, 50–100% B, 220–240 min, 100% B. The flow rate was set as 300 nL/min.

For Mass Spectrometry, data-dependent tandem mass spectrometry (MS/MS) analysis was performed with a Q Exactive Orbitrap mass spectrometer (Thermo Scientific, San Jose, CA). Peptides eluted from the LC column were directly electrosprayed into the mass spectrometer with the application of a distal 2 kV spray voltage. A cycle of one full-scan MS spectrum (m/z 300–1800) was acquired followed by top 20 MS/MS events, sequentially generated on the first to the twentieth most intense ions selected from the full MS spectrum at a 28% normalised collision energy. The full-scan resolution was set to 70,000 with automated gain control (AGC) target of 3e6. The MS/MS scan resolution was set to 17,500 with an isolation window of 1.8 m/z and AGC target of 1e5. The number of microscans was one for both MS and MS/MS scans and the time of maximum ion injection was 50 and 100 ms, respectively. The dynamic exclusion settings used were as follows: charge exclusion, 1 and >8; exclude isotopes, on; and exclusion duration, 30 s. MS scan functions and LC solvent gradients were controlled by the Xcalibur data system (Thermo Scientific).

For data analysis, the acquired MS/MS data were analysed against the UniProt Knowledgebase (https://www.uniprot.org, Swiss-Prot Rattus norvegicus database released on Apr. 1, 2019) using PEAKS Studio 8.5 (Bioinformatics Solutions, Waterloo, Ontario, Canada). The database search parameters were set as the followings: MS and MS/MS tolerance of 20 ppm and 0.1 Da, respectively, FDR was set as 1% and the protein identification threshold was set as $(−10 \log P) \geqq 20$. A protein was identified when at least one unique peptide was matched. Protein quantification was based on label-free quantitative analysis.

**Lentiviral constructs and production**. We initially developed the lentiviral transfer plasmid. For SIRT2 over expression, we inserted the *sirt2* gene into the pCDH-CMV-EF1α-flag-P2A-copGFP plasmid. For SIRT2 knockdown, we designed shRNA sequences using the BLOCK-iT™ RNAi Designer (Thermo Fisher) and inserted the sequences into the pGreenPuro plasmid (#SI505A-1, System Biosciences). Then the lentivirus producer cell line HEK239T was transiently co-transfected with transfer (pCDH-CMV-EF1α-SIRT2-flag-P2A-copGFP or pGreenPuro-shSIRT2 plasmid), envelope (pMD2.G, #12259, Addgene) and packaging (psPAX2, #12260, Addgene) plasmids to generate lentiviral particles. HEK293T cells were transfected using Polyethylenimine (PEI, #23966, Polysciences Inc., 1 mg/mL). After 8 h of transfection, the medium was changed to a fresh medium, and the lentivirus was collected after 48 h of transfection by centrifuged at 800 g at 4 °C for 10 min to remove cell debris. The SIRT2 shRNA sequences are: shRNA-F, 5′-gatccGGATGAAAGAGAAGATCTTCTtcaagagaAGAAGATCTT CTCTTTCATCCTTTTTg-3′, shRNA-R, 5′-aattcAAAAAGGATGAAAGAGAAG ATCTTCTtctcttgaaAGAAGATCTTCTCTTTCATCCg-3′.

**Construction of stable cell lines**. $5 \times 10^5$ OLN93 cells per well were seeded into a six-well plate and the culture medium was replaced with the collected lentiviral supernatant after 12 h. Fluorescence was observed after 48 h of lentivirus infection. The cells were gradually diluted and seeded into a 96-well plate and fluorescent cell clones were selected for passage after 2 weeks of culture.

**ChIP-qPCR**. For the cross-linking of protein and DNA, 270 μL of 37% formaldehyde was added into a 10 cm dish when the cell confluency reaches 90%, and the dish was shaken slowly for 10 min at RT. Then 500 μL of 2.5 M glycine was added into the medium and the dish was shaken for 5 min to quench cross-linking. The cells were washed three times with cold PBS and digested in 0.25% trypsin for

3 min, then collected by centrifugation (800 $g$ for 5 min at 4 °C) and washed twice with cold PBS. Cells were lysed in 500 μL ChIP-lysis buffer (50 mM Tris-HCl-pH 8.0, 0.5% SDS, 5 mM EDTA, protease inhibitor) for 5 min at 4 °C and centrifuged briefly. The samples were sonicated to shear the chromatin into about 300 bp fragments (typically, 0.5 s pulses followed by 0.5 s rest periods for 15 min at output 10). Then 5–10 μg of antibody or IgG was added into samples and incubated overnight at 4 °C. Antibodies used in this study include: Rabbit anti-SIRT2 (#S8447, Sigma, 1:100), mouse anti-flag (#M185-3L, MBL, 1:100), rabbit anti-H3K18Ac (#39694, Active motif), normal mouse IgG (#A7028, Beyotime), and normal rabbit IgG (#A7016, Beyotime). A 1% volume of the sample was pipetted out as input and the rest sample was centrifugated and the supernatant was transferred to a fresh micro-centrifuge tube. 50 μL Protein A/G PLUS-Agarose (#sc-2003, SANTA CRUZ) was blocked in the block buffer (10 mM Tris-HCl-pH 8.0, 1 mM EDTA-pH 8.0, 1% BSA) twice for 5 min and centrifugated for 2 min (800 $g$, 4 °C). The samples were incubated with agarose for 2 h at 4 °C and centrifugated for 2 min (800 $g$, 4 °C) and then the samples were washed with wash buffer I (20 mM Tris-HCl-pH 8.0, 150 mM NaCl, 2 mM EDTA, 1% TritonX-100, 0.1% SDS), wash buffer II (20 mM Tris-HCl-pH 8.0, 500 mM NaCl, 2 mM EDTA, 1% TritonX-100, 0.1% SDS), wash buffer III (10 mM Tris-HCl-pH 8.0, 0.25 M LiCl, 1 mM EDTA, 1% NP-40), TE buffer (10 mM Tris-HCl-pH 8.0, 1 mM EDTA-pH 8.0), TE buffer for 10 min at 4 °C followed by 2 min of centrifugation (800 $g$, 4 °C), respectively. The agarose was resuspended with 100 μL elution buffer (0.1 M NaHCO₃, 1% SDS, 1 mg/mL protease K) twice by using a rotary shaker for 10 min at RT and the supernatant was collected after centrifugation (800 $g$, 2 min, RT). The sample tubes were placed in a 65 °C water bath overnight to release cross-linking. The DNA was purified from samples using a Universal DNA Purification Kit (#DP214, TIANGEN) as instructed by the manufacturer.

For qPCR, DNA, primers and ChamQ Universal SYBR qPCR Master Mix (#Q711, Vazyme) were mixed as instructed by the manufacturer. Fold changes were calculated using the Delta-Delta Ct method in Microsoft Excel. The primer sequences are as follows: ChIP-ID4-F, 5′-TGGCACTGTCCTCCTGATTG-3′, ChIP-ID4-R, 5′-CCCTCAAAGTAACGACTTCCAA-3′.

**Statistics and reproducibility**. All data analyses were performed using GraphPad Prism 8.0 or R. The data are shown as mean ± SEM. All the images, immunoblots and statistical graphs are representatives of at least three experiments, unless otherwise stated. The "n" numbers for each experiment are specified in the figure legends. For in vivo experiments, each data point represents an individual animal. For in vitro experiments, each data point represents a biological replicate. For the comparison between two groups, statistical significance was determined using the unpaired two-tailed Student's $t$ test. For multiple comparisons, a one-way ANOVA test, followed by Tukey's post hoc test, or two-way ANOVA, followed by Sidak's multiple comparisons, was performed. Differences were considered statistically significant if $p < 0.05$ (*), $p < 0.01$ (**) or $p < 0.001$ (***). The heatmap was generated with the ggplot2 library.

**Reporting summary**. Further information on research design is available in the Nature Research Reporting Summary linked to this article.

## Data availability

The mass spectrometry proteomics data was analysed against the UniProt Knowledgebase (https://www.uniprot.org, Swiss-Prot Rattus norvegicus database released on Apr. 1 st 2019). The mass spectrometry proteomics raw data generated during this study is available at the ProteomeXchange Consortium via the iProX partner repository, with the dataset identifier PXD022046 and IPX0002529000. All other relevant data supporting the key findings of this study is available within the article and its Supplementary Information files. The source data generated in this study is provided in the Source Data files. Source data are provided with this paper.

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

## Acknowledgements

We thank the Center of Cryo-Electron Microscopy, Zhejiang University (ZJU), and the Core Facilities of ZJU School of Medicine. Particular thanks to Drs. Xing Zhang, Mengsheng Qiu, Hao Huang, Hui Fu and Mrs. Sanling Wu. Thanks to Ruo-Tong Zhao for providing language editing. This work was supported by the National Key R&D Program of China (grants 2017YFA0104900, 2021ZD0201700 and BZZ19J005 to J.-W.Z., and grant 2021ZD0202501 to Y.S.), and the National Natural Science Foundation of China (grants 81971144 and 81571170 to J.-W.Z., and grants 92049304 and 82030039 to Z.J.), and the Chinese Academy of Medical Sciences Innovation Fund for Medical Sciences (grant 2021-I2M-1-050) to H.-Z.C.

## Author contributions

J.-W.Z. and Z.J. conceived the project; J.-W.Z., Z.J., X.-R.M., and X.Z. designed the experiment; X.-R.M., X.Z., Y.X., H.-M.G., S.-S.Z., Liang Li, C.Y., W.J., K.Y., Y.Y., and C.P. performed most of the experiments and acquired data; X.-R.M., Y.X., H.-M.G., S.-S.Z., Liang Li, W.J., C.L., K.Y., Z.H., Z.M., and Y.Y. analysed and interpreted the data; F.W., Z.-J.D., D.-X.W., Y.W., Y.Z. and L.G. assisted with the experiments and helped to analyse the data. J.-W.Z. and X.-R.M. wrote the paper; Y.S., H.-Z.C., X.Z. and Li Li edited the paper; revised the paper. J.-W.Z. supervised all phases of the project.

## Competing interests

The authors declare no competing interests.
