## [Peer Review File · Nature Communications]

Restoring nuclear entry of Sirtuin 2 in oligodendrocyte progenitor cells promotes remyelination during ageingReviewers' Comments:

Reviewer #1:

Remarks to the Author:

Restoring nuclear entry of SIRT2 rejuvenates the aged oligodendrocyte progenitor cell and enhances remyelination (Ma et al.)

Ma et al. investigated the function of the NAD⁺ dependent deacetylase SIRT2 in oligodendrocyte precursor cells (OPCs). This is timely and interesting but focusses only on one function of SIRT2 in the oligodendrocyte lineage. While most studies in the past have analyzed SIRT2 in microglia and neurons, Ma provide further evidence that oligodendrocytes and OPCs are the main SIRT2 expressing cells in the brain. This is in accordance with the original papers by Werner et al. 2007, Cahoy et al. 2008 and Zhang et al. 2014.

However, there are some major concern and minor points that should be addressed:

(1) There is transient SIRT2 expression in OPCs during myelination and sustained SIRT2 expression in oligodendrocytes, yet the authors concentrate only on the function of SIRT2 in OPCs. However, with the genetic tools used, they cannot distinguish between SIRT2 function in OPCs and oligodendrocytes. Moreover, the effects of SIRT2 deficiency were most visible for myelin, a compartment made by mature oligodendrocytes. Thus, it may be necessary to compare conditional SIRT2 knockout mice that can separate the different functions of SIRT2 in OPCs (nuclear localization) and oligodendrocytes (cytoplasmic localization).

(2) Similarly, reexpression of SIRT2 specifically in OPCs (e.g. using viral vectors) would strengthen the idea that it is the nuclear entry in OPCs (but not in oligodendrocytes) that is required for efficient remyelination.

(3) Ma et al. use an impressive broad range of techniques, including genetically modified mice, cells acutely isolated cells (by immunopanning), primary cell culture and cell lines, metabolomics, proteomics, qPCR and immunoblot. However, this broad technical spectrum, which is a strength of the paper, also yields some inconclusive data (see below).

(4) The discussion of data is very brief. Important references strengthening the observations by Ma et al. are missing (see above). In contrast, the authors draw too strong conclusions on causality from SIRT2 loss of function and impaired remyelination and relate this to inflammatory demyelinating events in MS and EAE. As SIRT2 is an abundant myelin protein, SIRT2 reduction could also be a consequence of impaired remyelination, i.e. a measure of myelin abundance.

Minor points:

(1) The methods section is missing important details. The culture conditions described do not support OPC differentiation. Culture conditions used for neurons and microglia are missing.

Immunoblots appear to be stripped for detection of modified and total protein species. That procedure is missing and it is critical that primary antibodies are removed (note that by conventional stripping at low pH and elevated temperatures only secondary antibodies are stripped). Alternatively, primary antibodies obtained from different species should be used.

(2) Addition of DMSO was used in all cell culture experiments as control condition for β -NMN application. However, β -NMN is DMSO insoluble. One wonders whether the strong effects on OPC proliferation and differentiation as well as protein abundance by proteomics observed in cultures treated with β -NMN is hampered by usage of the organic solvent DMSO more in the control experiment.

(3) Can a gradient in the substrate incubation be excluded for the immunoblot in Fig 1j?

(4) Authors should show single channel images for Fig 1l to prove nuclear localization of SIRT2 in OPCs because spherical SIRT2⁺ debris on the section could be misleading.

(5) Is remyelination after LPC in SIRT2^{-/-} mice permanently reduced? As OPC numbers are normalized 10dpi (Fig 2j), could this indicate a transient effect that is normalized later?

(6) In the normal corpus callosum of mice about half of the axons remain non-myelinated. In Fig 2r the authors observe >55% remyelinated axons. What is this number is normalized to?

(7) Measurement of myelin pathology by staging in categories 1 and 2 refer to non-adhered single myelin layers observed in TEM. The authors must show this effect is not due to fixation or processing artifacts and provide a comprehensible analysis flow in the methods section.

(8) Fig 2q shows some mitochondrial enlargements and possible organelle accumulations in the axons. Is there axonal pathology in SIRT2^{-/-} and old WT mice?

(9) NAD⁺ was one out of many metabolites altered in TERC^{-/-} OPCs (Fig 3b). Could a more general metabolic effect of body wide telomere shortening explain these global changes? Were similar alterations observed in other cells or organs of these mice? As the precursors of NAD⁺ (nicotinamide and nicotinate) were all detected at normal levels, what is the cause for NAD⁺ reduction?

(10) What is the reason for using primary rat OPCs for proteomics in Fig 3d when all other primary cell culture experiments were performed with mouse OPCs?

(11) The improved differentiation to oligodendrocytes observed in Fig 4c, d is not reflected by the proteomics data in Fig 3e, f (where no increase in myelin proteins e.g. MBP was detectable). Do the authors have an explanation?

(12) Experiments searching for a molecular mechanism detected changes in ID4 mRNA abundance. Are these changes also detectable at the protein level? Do they result in changes of MBP expression?

Reviewer #2:

Remarks to the Author:

Ma et al. have carried out a substantial amount of work using various technical tools to address the role of SIRT2 in improving myelination and remyelination in ageing. They show that loss of SIRT2 and NAD⁺ in ageing and a mouse model of premature ageing correlates well with poor myelination and remyelination outcomes post a demyelination insult. Additionally, increasing SIRT2 expression, especially the nuclear component in OPCs via administration of β NMN (NAD⁺ precursor) ameliorates the poor myelination and remyelination seen in ageing and the premature ageing mouse model. This was observed via improved compound action potential propagation, increased frequency of remyelinated axons, lowered g-ratios, reduced appearance of pathological myelin and improved 'compaction' of myelination as shown by the myelin dense lines measurements. They further show that β NMN administration in mice lacking SIRT2 shows absolutely no change in OPC, OL or myelin outcomes post demyelination, indicating β NMN mediates its effects through SIRT2. Finally using invitro experiments, the authors show β NMN increases SIRT2 nuclear entry, decreased ID4 expression whereas knockdown of SIRT2 increases acetylation of H3K18 and tubulin and increases ID4. They go on to show enrichment of SIRT2 and acetylated H3K18 on ID4 promoter. The authors propose that SIRT2 deacetylates H3K18, inhibiting ID4 and this allows MBP transcription to go ahead leading to OPC differentiation into OLs and hence improving myelin outcomes.

Overall, I am in agreement with the experiments and results, however, there are a few matters that need to be addressed to elevate the manuscript to a better level.

In figure 1 relative mRNA level of Sirt2 in mature OLs in panels b and c are very different 250 vs 750 while the levels of sirt2 are relatively similar in the OPCs. Do the authors have an explanation for this. Additionally, these experiments carried out using OPCs from the rat cerebral cortex. This is a curious choice since the authors show other data in the manuscript using cultured mouse OPCs. Graphs in Figure 1e, 1m & n show that a large proportion but not all the OPCs show SIRT2 nuclear localisation during development and demyelination/remyelination phases. Do the authors have a reason as to why that might be the case? Since most OPCs in a demyelinated lesion would either differentiate or die it is likely that only OPCs meant to be differentiating would have nuclear SIRT2. Is it a case of timing? i.e., because it is transient expression all of them do express SIRT2 but only a fraction may be recorded at any snapshot?

In figure 2h & i, a significant drop in the proliferative ability of OPCs is seen in the SIRT2^{-/-} mice 5dpl. However, in WT or G3Terc^{-/-} mice supplemented with β NMN there is no change in OPC's proliferative ability. However, the authors make no attempt to explain or pursue this finding.

In the results and discussion authors state that SIRT2 is required & necessary for remyelination however this is not strictly true since a 25-30% remyelination frequency is seen in SIRT2^{-/-} LPC lesion compared to ~55% in age matched WT mice (figure 2r) and 20% of remyelinated axons in SIRT2 group show no (grade 0) myelin pathology (figure 2s). Also, it is not simply the loss of SIRT2 (line 356) but also the inability to efficiently upregulate and recruit SIRT2 to the nucleus that causes impaired remyelination as seen in figure 1e and 1m and 1n where, by P21 there is almost no expression of SIRT2 in OPC cytoplasm or nucleus however mice that young can

remyelinate very efficiently. Clarity of words is important in the take home message.

Myelin ultrastructure of G3Terc -/- mice T 6mo is significantly worse than that of 18mo WT mice (extended data Figure 3g) so there is certainly more than simply ageing going on in G3 Terc -/- mice.

In several places in the manuscript authors state that β NMN or nuclear SIRT2 in OPCs rejuvenates myelin. This I believe is mediated through the myelin made by the newly differentiating OLs (ones coming from rejuvenated OPCs) that make a new 'superior quality' myelin. Often, one gets the impression from the language used in the paper that the old poor-quality myelin is mended or fixed by administering β NMN. Do the authors mean to imply that NAD⁺ supplementation can actually repair grade 1 and possibly grade 2 myelin back to grade 0? And if so how? The authors do not show the in vivo effects of β NMN addition on OPC proliferation and differentiation in G3 Terc-/- mice or Old WT mice in un-lesioned animals. Only improved myelin outcomes are shown (figure 4e-t). If SIRT2 facilitates MBP expression, and MBP protein is produced throughout the lifetime of OL, it is possible that older pre-existing OLs may be able to compact myelin better with increased MBP expression through greater SIRT2 availability when β NMN is added. This could happen even in absence of any change in proliferation or differentiation.

Long term treatment with β NMN has very similar outcomes to immediate as well as delayed β NMN administration. This is an interesting finding showing that requirement of upregulated nuclear SIRT2 may be only necessary only immediately prior or at the point of differentiation. The total number of Olig2⁺ cells in G3 Terc -/- mice treated with β NMN (long term and immediate) is similar to PBS treated animals 5dpl and 10dpl, as are numbers of proliferating OPCs and proportion of proliferating OPCs. However, at 10dpl there are more differentiated OLs (in immediate) and proportionally more OLs compared to OL lineage cells. What population is giving rise to these differentiating cells? Non-proliferating OPCs? In absence of total OPC density data in the lesion (at 5dpl and 10dpl) it is difficult to imagine how the proliferating OPCs numbers are similar, total OL (CC1⁺) density is greater but the total Olig2⁺ lineage is balanced.

In the material and methods under immunofluorescence it says that primary antibodies were incubated for 48-72 hours. Since the tissue sections are 12 μ m thick, 3 days of incubation seems excessive and may produce labelling artefacts. Post-mortem human tissues for antigen detection often require long exposures, but in this study the human tissue was incubated only for 12 hours according to the methods.

I am amazed at almost 16 weeks of daily i.p injection in a mouse not producing significant peritonitis or damage. The authors need to provide the standard definition of g-ratio under assessment of remyelination and then they can explain how they calculated it using by tracing inner and outer circles. As it stands right now it looks as if myelin thickness is being measured. The measurement of axonal diameter is mentioned later as an aside and it may be confusing to the uninitiated as to how this figures into the g-ratio calculation. It would be helpful, if a mention was made that only axons displaying grade 0 myelin pathology were assessed for g-ratios, (I assume that this was the case). This is important because there are notable number of axons displaying grade 1 and 2 pathology levels even in WT young mice. I think the flow of information in the model shown in figure 8 could be improved, it not easy to follow in the 'on' panel.

The manuscript is simply and clearly written which makes for a good reading, however, there are a fair number of grammatical errors that when fixed will make for a much smoother reading.

Reviewer #3:

Remarks to the Author:

Ma et al. report that SIRT2 is highly expressed in OPCs mainly with a nuclear localization during myelin development. Demyelination, induced by LPC, allows for SIRT2 re-expression in the nucleus of young mice. SIRT2 is necessary for remyelination following chemical demyelination. They show that supplementation of NAD⁺ through β -NMN enhances SIRT2 nuclear entry in the aged OPC upon demyelination in G3 Terc-/- mice. They also demonstrate that NAD⁺ supplementation delays myelin aging and enhances remyelination in the aged mice, which required SIRT2. Finally, they describe the molecular mechanism by which the addition of NMN on OPC induces SIRT2 nuclear entry, SIRT2 deacetylates H3K18 leading to the inhibition of the transcription factor ID4 which will in turn promote the differentiation of OPC. This paper highlights a new role for SIRT2 in OPC aging. This is a timely finding given that the sirtuin field is focused on the role of sirtuins in stem/progenitor cell aging and tissue degeneration.

Major comments:

- The nuclear localization of SIRT2 is not convincing. e.g. in Figure 1d, FDGFR, a cellular surface protein, and in Figure 1g, NG2, also a plasma membrane protein, also appear to be in the nucleus in P0 or P3. Since this study places strong emphasis on nuclear translocation of SIRT2 as a mechanism, more convincing imaging data is necessary. Also, quantification "cell number/mm²" can be complicated by overall cellular density in the images. It is better to quantify "percent of cells with nuclear SIRT2". In addition, typical cellular localization studies include cellular fractionation and Western blotting.
- 1m and 1n: SIRT2+PDGFRa+ cells were quantified. This could be complicated by changes in PDGFRa+ cells while the authors were trying to argue for an increase in SIRT2. SIRT2+PDGFRa+/PDGFRa+ should be quantified.
- Figure 3: The argument that NMN increases SIRT2 expression and nuclear localization is puzzling. NMN increases NAD levels, which is a cofactor of SIRT2. How would NAD increase SIRT2 expression or nuclear localization?
- Figure 6: 3-month-old SIRT2 KO mice were tested for the effect of NMN, without age-matched WT control. In Figure 4 and 5, old WT mice responded to NMN. However, it is unclear if young WT mice respond to NMN.
- Figure 7f shows increased ID4 in SIRT2 KO brain but this could be complicated by changes in cell number etc. What about in OPC cells specifically?
- Figure 7h. SIRT2 KO cells should be used as a control to ensure specific binding of SIRT2. As discussed above, it is puzzling how NMN increases SIRT2 binding to chromosome.
- Figure 7i. SIRT2 has deacetylase activity. If NMN increases SIRT2 binding to the promoter, it would reduce H3K18Ac. This is opposite to what's shown in 7i.

Minor comments:

- Redundant figures 1a, 2b, 3d and 6a. No need to show "cultured cells or OPC" as a figure or just once.
- Line 140: "in contrast"

Reviewer #1 (Remarks to the Author):

Restoring nuclear entry of SIRT2 rejuvenates the aged oligodendrocyte progenitor cell and
enhances remyelination (Ma et al.)

5 Ma et al. investigated the function of the NAD⁺ dependent deacetylase SIRT2 in oligodendrocyte
precursor cells (OPCs). This is timely and interesting but focusses only on one function of SIRT2
in the oligodendrocyte lineage. While most studies in the past have analyzed SIRT2 in microglia
and neurons, Ma provide further evidence that oligodendrocytes and OPCs are the main SIRT2
expressing cells in the brain. This is in accordance with the original papers by Werner et al. 2007,
Cahoy et al. 2008 and Zhang et al. 2014.

Reply: Thank Reviewer #1 for the positive comments on our work. Reviewer #1 mentioned 3
relevant original papers, and among them, one was cited in our submission (Werner et al. 2007).
Werner et al. and Li et al. showed that SIRT2 is expressed throughout the oligodendrocyte lineage
cells, and exclusively in mature oligodendrocytes in the adult CNS (Werner et al. 2007; Li et al.,
2007, both were cited in our submission). This current study further reveals that nuclear entry of
SIRT2 is seen in postnatal developmental OPCs but is depleted in the adult and aged OPCs.
Furthermore, NAD⁺ restores nuclear entry of SIRT2 in the aged OPCs and delays myelin aging
and enhances remyelination through a previously unknown pathway,
NAD⁺-SIRT2-H3K18Ac-ID4. During myelin repair, SIRT2 plays a critical role. Moreover, the
effects of NAD⁺ on OPCs requires SIRT2. Therefore, this work identifies SIRT2 as a novel
molecular target in OPCs, discovering a new strategy to rejuvenate myelin aging and enhance
myelin repair in the aged CNS. Following the advice of Reviewer #1, we have now added the
other 2 papers in the Reference list (Cahoy et al. 2008; Zhang et al. 2014).

However, there are some major concern and minor points that should be addressed:

(1) There is transient SIRT2 expression in OPCs during myelination and sustained SIRT2
expression in oligodendrocytes, yet the authors concentrate only on the function of SIRT2 in
OPCs. However, with the genetic tools used, they cannot distinguish between SIRT2
function in OPCs and oligodendrocytes. Moreover, the effects of SIRT2 deficiency were
most visible for myelin, a compartment made by mature oligodendrocytes. Thus, it may be
necessary to compare conditional SIRT2 knockout mice that can separate the different
functions of SIRT2 in OPCs (nuclear localization) and oligodendrocytes (cytoplasmic
localization).

Reply: Thanks to Reviewer #1 for making a very important point, which is to distinguish the
function of SIRT2 between that in the OPCs and the mature oligodendrocytes. As Reviewer #1
pointed out in the above comments, Werner et al. 2007 reported that SIRT2 is expressed
throughout the oligodendrocyte lineage¹, and transcriptome database provided by Cahoy et al.
2008 and Zhang et al. 2014 also showed that the mRNA of SIRT2 is enriched in oligodendrocytes
lineage cells^{2,3}. These data are in accordance with our results and we have cited these
literatures in the “Discussion” section (Lines 404-408).

Werner et al.¹ and Li et al.⁴ reported in 2007 that SIRT2 is expressed throughout the
oligodendrocyte lineage cells, and exclusively in mature oligodendrocytes in the adult CNS. Since
then the function of SIRT2 in mature oligodendrocytes has been explored in several studies. Ji et
al. showed that SIRT2 enhances oligodendroglia differentiation⁵ while Li et al. reported that
SIRT2 decelerates oligodendroglia differentiation by deacetylating α -tubulin⁴. The reasons for the

controversial results on the function of SIRT2 in mature oligodendrocytes are still not clear,
however, partially because the roles of SIRT2 has not been clearly separated between OPCs and
the mature oligodendrocytes.

In 2011, Beirowski et al. showed that in peripheral nerves system, conditional knockout of
SIRT2 in Schwann cells using MPZ-Cre mice crossed with $Sirt2^{flox/flox}$ mice, delays peripheral
myelin formation⁶. A recent study reported in 2021 by Chamberlain et al. showed that SIRT2
expressed in mature oligodendrocytes plays an important role in enhancing axonal energy
metabolism⁷. They proposed that the oligodendrocyte-to-axon delivery of SIRT2-filled exosomes
enhances axonal ATP production by deacetylating mitochondrial proteins in axons.

However, so far, the function of SIRT2 in OPCs remains unknown. In response to
Reviewer #1, to clearly investigate the role of SIRT2 specifically in OPCs in vivo, we purchased
$Sirt2^{flox/flox}$ mice from Shanghai Model Organisms (#NM-CKO-190038) and in the $Sirt2^{flox}$ allele
of $Sirt2^{flox/flox}$ mice, two loxP sites flank three critical exons (Reviewer #1-Figure 1a). Mice that
are homozygous for the $Sirt2^{flox}$ allele appeared normal and were fertile. To specifically delete
SIRT2 in OPCs, the $Sirt2^{flox/flox}$ mice were crossed with NG2-Cre^{ERT} mice, and the heterozygous
offspring were crossed again to obtain NG2-Cre^{ERT}; $Sirt2^{flox/flox}$ mice. It took us about 5 months to
obtain NG2-Cre^{ERT}; $Sirt2^{flox/flox}$ mice. The mice were intraperitoneally injected with tamoxifen
(Sigma, #10540-29-1, 100mg/kg) for 5 days⁸⁻¹⁰ so that Cre recombinase-mediated excision creates
a nonfunctional allele exclusively in OPCs in temporally controlled manner, and the tamoxifen
was dissolved in a 9:1 ratio of corn oil: ethanol. Four days post the last administration of
tamoxifen^{9,10}, we induced demyelination by focal injection of lysolecithin (LPC) into the corpus
callosum and detected remyelination at 21 dpl with transmission electron microscope (TEM,
Reviewer #1-Figure 1b).

We first confirmed the genetic deletion of SIRT2 in OPCs by immunofluorescence (Reviewer
#1-Figure 1c). Our results showed that in the demyelinated lesion, SIRT2 was not detected in the
OPCs of NG2-Cre^{ERT}; $Sirt2^{flox/flox}$ mice that received tamoxifen injection but SIRT2 was indeed
detected in the nuclei of OPCs of $Sirt2^{flox/flox}$ mice (Reviewer #1-Figure 1c). Then we tested the
remyelination efficiency by TEM. Our new results showed that conditional knockout of SIRT2 in
OPCs significantly impaired myelin repair (Reviewer #1-Figure 1d), reduced the frequency of
remyelinated axon (Reviewer #1-Figure 1e), and decreased the thickness of newly formed myelin
(Reviewer #1-Figure 1f). The pathological analysis of myelin showed that the frequency of
normal-looking newly formed myelin reduced while the grade 1 and grade 2 myelin increased by
conditional knockout SIRT2 in OPCs (Reviewer #1-Figure 1g). These new results provide
direct evidence that SIRT2 in OPCs plays a critical role in remyelination. These new results
**have been put as Revised Extended Data Fig.5** and we have added it in the “Results” section
(Lines 191-202) accordingly. The relevant information on mice, genotyping, and tamoxifen
treatment have also been added to the “Methods” section (Lines 497-506 and lines 511-516).

In this research, we revealed that NAD⁺ restores nuclear entry of SIRT2 in the aged OPCs
and delays myelin aging and enhances remyelination, and the SIRT2 in OPCs is critical for
remyelination. Since it is technically challenging to manipulate nuclear entry of SIRT2 in the
OPCs, we think conditionally deletion of SIRT2 in OPCs is the best method we can do at present.

Reviewer #1-Figure 1 (Revised Extended Data Fig.5)

Conditional SIRT2 knockout specifically in OPCs of mouse verified that SIRT2 plays a

critical role in remyelination. a, Schematic illustration of the conditional *Sirt2* allele with loxP

sequences (red) flanking exons 5–7 (black). The Flp recombinase recognition sequences (FRT site)

are showed in blue. b, The flow chart of experimental design. Tamoxifen was given (100 mg/kg,

intraperitoneal daily) for 5 days started at age of 2 months, and 4 days later, LPC was focally

injected into the corpus callosum to induce demyelination. The mice were sacrificed at 21 days

post LPC injection. c, Immunofluorescence of SIRT2 and PDGFR α ⁺ OPCs within the demyelinated lesion in the corpus callosum of the *Sirt2*^{fl/fl} mice and the *NG2-Cre^{ERT}; Sirt2*^{fl/fl} mice at 21 dpl. Scale bar, 1 μm. d, TEM micrographs within the lesions at 21 dpl. Scale bar, 1 μm. e, Quantification of the proportion of remyelinated axons. f, Quantification of individual G-Ratio distribution (linear regression). g, Quantification of myelin pathology level. n=3 for the *Sirt2*^{fl/fl} mice group, n=4 for the *NG2-Cre^{ERT}; Sirt2*^{fl/fl} mice group. All data are presented as mean \pm SEM. * p<0.05 by two-tailed t test (e).

As Reviewer #1 pointed out, the expression and the role of SIRT2 in mature oligodendrocytes have been reported^{1,4}. In response to Reviewer #1, we have started breeding mice in which SIRT2 is intended to be specifically deleted in mature oligodendrocytes. This line of mouse, *Plp-Cre^{ERT}; Sirt2*^{fl/fl}, is still in breeding, and it will take much longer time. Considering the focus of the current work is the role of SIRT2 in OPCs and the deadline for revision has been extended twice already, we feel that the results on the specific role of SIRT2 in mature oligodendrocytes are more

suitable to be presented later in another independent story. We greatly appreciate understanding
from Reviewer #1 on this point, and hope that our responses are adequate to this question.

(2) Similarly, reexpression of SIRT2 specifically in OPCs (e.g. using viral vectors) would
strengthen the idea that it is the nuclear entry in OPCs (but not in oligodendrocytes) that is
required for efficient remyelination.

Reply: Thanks to Reviewer #1 for the advice to specifically infect OPCs or mature
oligodendrocytes using viruses. Following Reviewer #1's suggestion, we have tried
AAV1/2-MBP-Cre, which was reported to infect mature oligodendrocyte only¹¹, and this virus
infected both neurons and oligodendrocyte lineage cells. We also tried two viruses (Retro virus
and olig001-AAV1/2) that were reported to infect oligodendrocyte lineages cells (OPCs and
mature oligodendrocytes)^{12,13}. Unfortunately, we found that both Retro virus and olig001-AAV1/2
infected more neurons than oligodendrocyte lineage cells. The non-specificity of the viruses which
have been reported to specifically target OPCs or mature oligodendrocytes makes the application
of viruses to differentially target OPCs or mature oligodendrocytes in our revision impractical. We
have checked literature, together with our trial data, it seems that lacking virus tool to specifically
target OPCs or mature oligodendrocyte is a key technical bottleneck in the field.

(3) Ma et al. use an impressive broad range of techniques, including genetically modified
mice, cells acutely isolated cells (by immunopanning), primary cell culture and cell lines,
metabolomics, proteomics, qPCR and immunoblot. However, this broad technical spectrum,
which is a strength of the paper, also yields some inconclusive data (see below).

Reply: Thanks to Reviewer #1 for recognizing the strength of this manuscript in which we used a
broad spectrum of techniques. To minimize the possible inconclusiveness, in accordance with the
recommendations of Reviewer #1, we have modified the following:

- a. We explained the reason why T3 was omitted in the medium used for culturing OPC.
- b. We added protocols for culturing neurons, microglia and astrocytes in "Methods".
- c. Taking the possible defects of stripping in Western blot into accounts, the stripping procedure
were carefully reconsolidated and we have now added the stripping procedure in "Methods".
In our new experiments, the primary antibodies we used for the modified and total protein
were reexamined and consolidated. The primary antibodies obtained from different species
were used and previous results of WB were consolidated.
- 140 d. We did new experiments using PBS as control in vitro to detect the effect of β -NMN on the
141 proliferation and/or differentiation of OPC, and consolidated the conclusion of our submission
in which DMSO was used as control.
- e. We elaborated the statistical methods of remyelinated axons frequency and explained why we
observed >55% remyelinated axons.
- f. We elaborated on the special measures that were taken to minimize the possibility that the
loose myelin was caused by processing artifacts.

The details of these modifications can be seen throughout the reply and we hope our reply and
modification have minimized the inconclusiveness of this work.

(4) The discussion of data is very brief. Important references strengthening the observations
by Ma et al. are missing (see above). In contrast, the authors draw too strong conclusions on

causality from SIRT2 loss of function and impaired remyelination and relate this to
inflammatory demyelinating events in MS and EAE. As SIRT2 is an abundant myelin
protein, SIRT2 reduction could also be a consequence of impaired remyelination, i.e. a
measure of myelin abundance.

Reply: Thanks to Reviewer #1 for reminding us of the references that we have missed. In
response, we have carefully rechecked the literature and have cited the 2 papers mentioned by
Reviewer #1. Accordingly, **we have cited 10 more references to strengthen our results.**

Concerning the role of SIRT2 during remyelination, we have carefully reexamined the relevant
data, together with our new results on conditional SIRT2 knockout in OPCs (Reviewer #1-Figure
1). Accordingly, we have modified the related result title into “SIRT2 is **critical** for remyelination”
in **line 152**, and “Our in vitro and in vivo data indicate that SIRT2 is **critical** for remyelination” in
the content in **line 177**. Together with our new data on conditional knockout of SIRT2 specifically
in OPC, we have added a paragraph in “Results” (**lines 191-202**) and a paragraph in “Discussion”
(**lines 420-429**)

Minor points:

(1) The methods section is missing important details. The culture conditions described do
not support OPC differentiation. Culture conditions used for neurons and microglia are
missing. Immunoblots appear to be stripped for detection of modified and total protein
species. That procedure is missing and it is critical that primary antibodies are removed
(note that by conventional stripping at low pH and elevated temperatures only secondary
antibodies are stripped). Alternatively, primary antibodies obtained from different species
should be used.

**“The culture conditions described do not support OPC differentiation.”**

Reply: Thanks to Reviewer #1 for carefully reviewing our “Methods” section. It has been
reported that omission of T3 in differentiation medium was used in previous work to screen
compounds that enhance the differentiation of primary cultured young OPCs¹⁴. Therefore, we
followed this protocol and did not add T3 in our medium. Accordingly, we have added this paper
into the Reference list. Our culture medium is Neurobasal medium containing 2% B27
Supplement and 1% N2 Supplement. To demonstrate why we use the medium omitted T3, we did
new experiment of OPCs culture. In this medium, less than half of the OPCs automatically
differentiate into mature oligodendrocytes. In comparison, traditional supplementation of T3 in
medium strongly promotes the differentiation of OPCs and raised the differentiation proportion of
OPCs to about 80% (Reviewer #1-Figure 2), and this made additional enhancement of the
differentiation of OPCs by other compounds very difficult, and adding T3 masked the effect of
β -NMN on OPCs (Reviewer #1-Figure 2).

Reviewer #1-Figure 2

**Adding T3 in differentiation medium masked the effect of β-NMN on OPCs.** Quantification of
differentiated oligodendrocytes (MBP⁺Olig2⁺) cultured for 48 hours from P0 WT mice (n=3). All
data are presented as mean ± SEM. *p<0.05 by one-way ANOVA followed by Tukey's post hoc
test. n.s., no significance.

**“Culture conditions used for neurons and microglia are missing.”**

Reply: Thanks to Reviewer #1 for spotting this out, and we have now added them in “Methods”
(lines 528-538).

In paragraph “Primary culture of OPC, microglia and astrocyte”, we have added the
following:

At day 8, the flask was shaken on a shaker at 200 rpm for 2 hours at 37°C and the medium was
centrifuged at 300 g for 5 min to collect microglia. Then fresh medium was added into the flask.
For OPC purification, the flask was shaken at 250 rpm for 16-18 hours and the medium was
centrifuged at 1400 rpm for 5 min and then OPCs were collected. The firmly attached cells at the
bottom of the flask are astrocytes. Astrocytes were collected by 2 min digestion in 0.25% trypsin
at 37°C and 5 min centrifugation at 300 g. After purification, microglia and astrocytes were
seeded onto dishes and cultured in DMEM/F12 containing 10% FBS. OPCs were seeded onto
coverslips or dishes coated with PDL and cultured in DMEM/F12 containing 10% FBS. After 3-4
209 hours culture, the medium of OPCs was changed to Neurobasal™ (#21103049, Gibco) medium
containing B27™ Supplement (#17504044, Gibco) and N2 Supplement (#17502001, Gibco).

We have also added a new paragraph (lines 546-554):

Primary culture of neuron. For rat neuron culture, hippocamps from embryonic day 18 rats
were dissected out in ice-cold HBSS and digested for 12 min at 37°C in 0.125% trypsin. The
digestion was terminated with DMEM (#11965092, Gibco) containing 10% F12 (#11765062,
Gibco) and 10% FBS (#10099141, Gibco). Then the tissue was gently pipetted into a single-cell
suspension with a pipette (1 mL) and left still for 2 min. The single-cell suspension was then
transferred to a new 1.5 mL tube and centrifuged at 1200 rpm for 5min at 4°C. The cells were
resuspended in Neurobasal™ medium containing 2% B27™ Supplement and 1% Glutamax
(#35050061, Gibco) and seeded onto dishes that were pre-coated with PDL. 1.25 μM cytarabine
was added to inhibit the growth of glial cells.

“Immunoblots appear to be stripped for detection of modified and total protein species.
That procedure is missing and it is critical that primary antibodies are removed (note that
by conventional stripping at low pH and elevated temperatures only secondary antibodies

are stripped). Alternatively, primary antibodies obtained from different species should be
used.”

Reply: Thanks to Reviewer #1 for the questions on the stripping procedure. Following advice of
Reviewer #1, we have added the stripping procedure in “Methods” (lines 790-793):

The stripping buffer was: 25 mM Glycine and 1% SDS in ddH₂O, pH 2.0. The blots were
incubated in stripping buffer on a shaker for 20 min at room temperature (RT) to remove
antibodies and washed in TBST for 10 min for 3 times. The blots were blocked at RT for 2 hours
in 5% skim milk blocking buffer in TBST and then incubated with another primary antibody.

To resolve the concerns of Reviewer #1 on stripping, we have done new blots using
stripping procedure in different sequence of total tubulin and acetylated tubulin, and two
species (rabbit and mouse) antibodies of both tubulin and H3 were used. Our new results
confirmed that in our experimental setup, blots of total tubulin, acetylated tubulin,
H3K18Ac and H3 were specific and was not caused by incomplete stripping. Our new
results also consolidated the conclusion that SIRT2 overexpression decreased the acetylation
levels of both tubulin and H3K18 but did not affect the level of total tubulin and H3.

We used OLN93 cells with SIRT2 overexpression (Reviewer #1-Figure 3a) to test the
acetylation level of tubulin and H3K18. In the first set, we did blots on the same membrane in
sequence of acetylated Tubulin-secondary antibody-Mouse anti-Tubulin-Rabbit
anti-Tubulin. The acetylation level of tubulin at Lysine 40 significantly reduced after SIRT2
overexpression (Reviewer #1-Figure 3b, exposure time: 12 s). To test if the primary antibody has
been removed, the bolt was incubated in stripping buffer as described above and washed with
TBST. Then the blot was incubated with the same second antibody (Peroxidase affinipure donkey
anti-mouse IgG (H+L), Jackson ImmunoResearch) and no obvious bands were detected even
with long exposure time (Reviewer #1-Figure 3c, exposure time: 2 min), confirming that the
primary antibody Ac-TubulinLys⁴⁰-Mouse was removed. The anti-tubulin antibodies obtained
from both mouse (after the second stripping, Reviewer #1-Figure 3d, exposure time: 8 s) and
rabbit (after the third stripping, Reviewer #1-Figure 3e, exposure time: 10 s) showed that the
tubulin level did not change. The same blot was used in Reviewer #1-Figure 3b-e.

In the second set, we did blots on the same membrane in sequence of Mouse anti-Tubulin-
secondary antibody-acetylated Tubulin-Rabbit anti-Tubulin. The blot was first incubated with
tubulin antibody (Reviewer #1-Figure 3f, exposure time: 6 s) and SIRT2 overexpression did not
affect the level of total tubulin. After the first stripping, the blot was incubated with the same
second antibody (Peroxidase affinipure donkey anti-mouse IgG (H+L)), we found that weak
bands were still detected after long exposure (Reviewer #1-Figure 3g, exposure time: 2 min),
indicating this stripping did not completely remove the primary antibody-Mouse
anti-Tubulin. The blot was incubated with Ac-tubulin^{Lys40} and showed that SIRT2 overexpression
significantly decreased the acetylation level of tubulin (after the second stripping, mouse,
Reviewer #1-Figure 3h, exposure time: 12 s). Blot using Rabbit anti-tubulin (after the third
stripping, rabbit, Reviewer #1-Figure 3i, exposure time: 10 s) showed that SIRT2 overexpression
did not change the total tubulin level, identical to the blot that before stripping mouse anti-Tubulin
was used. The same blot was used in Reviewer #1-Figure 3f-i.

In the third set, we did blots on the same membrane in sequence of Rabbit anti-H3K18Ac-
Mouse anti-H3-Rabbit anti-H3. Results showed that the acetylation level of H3K18 reduced
after SIRT2 overexpression (Reviewer #1-Figure 3j, exposure time: 30 s), whereas the H3 level

did not change using antibodies of both mouse (after the first stripping, Reviewer #1-Figure 3k,
 exposure time: 8 s) and rabbit (after the second stripping, Reviewer #1-Figure 3l, exposure time:
 8 s). The same blot was used in Reviewer #1-Figure 3j-l.

In summary, technically, our new data indicated that despite whether stripping can completely
 remove the afore-labelled primary antibodies is case-dependent, in our experimental setup, we
 only use very short exposure time (6-30 seconds), and within this time window, we did not
 detect the remained afore-labelled primary antibody signal. Scientifically, our new data
 consistently supported our conclusion.

Reviewer #1-Figure 3

Various sequences of stripping and primary antibodies from different species validated WB
 results. a, Immunoblot of SIRT2 in OLN93 cell line. ctrl-flag:
 PCDH-EF1 α -MCS-flag-P2A-copGFP (control); SIRT2-flag:
 PCDH-EF1 α -SIRT2-flag-P2A-copGFP (SIRT2 overexpression). b, immunoblot of
 Ac-Tubulin^{Lys40} in OLN93 cell line (primary antibody obtained from mouse). c, blot incubated
 with secondary antibody (donkey anti-mouse) after stripping. d, immunoblot of Tubulin in
 OLN93 cell line (primary antibody obtained from mouse). e, immunoblot of Tubulin in OLN93
 cell line (primary antibody obtained from rabbit). The same blot was used in b-e. f, immunoblot of
 Tubulin in OLN93 cell line (primary antibody obtained from mouse). g, blot incubated with
 secondary antibody (donkey anti-mouse) after stripping. h, immunoblot of Ac-Tubulin^{Lys40} in
 OLN93 cell line (primary antibody obtained from mouse). i, immunoblot of Tubulin in OLN93
 cell line (primary antibody obtained from rabbit). The same blot was used in f-i. j, immunoblot of

H3K18Ac in OLN93 cell line (primary antibody obtained from rabbit). k, immunoblot of H3 in
OLN93 cell line (primary antibody obtained from mouse). l, immunoblot of H3 in OLN93 cell
line (primary antibody obtained from rabbit). The same blot was used in j-l.

(2) Addition of DMSO was used in all cell culture experiments as control condition for
β -NMN application. However, β -NMN is DMSO insoluble. One wonders whether the strong
effects on OPC proliferation and differentiation as well as protein abundance by proteomics
observed in cultures treated with β -NMN is hampered by usage of the organic solvent
DMSO more in the control experiment.

Reply: Thanks to Reviewer #1 for pointing out this. β -NMN was dissolved in PBS in our in vitro
experiment. During our experiment, because many compounds were screened at the same time,
most of the compounds were soluble in DMSO, so we chose DMSO as the control. In response to
the concern of Reviewer #1, we have done new in vitro experiments in which PBS was used as the
control. Now the “previous Fig. 4b, d” have been replaced by “Reviewer #1-Figures 4a & b”,
and “previous Fig. 6c, e” have been replaced by “Reviewer #1-Figures 4c & d”, and the
Reviewer #1-Figure 4 are presented below. Our new results showed that, in comparison with
PBS, β -NMN significantly improved the proliferation of G3 Terc^{-/-} OPCs (Reviewer #1-Figure
4a) but had no effect on WT OPCs (Reviewer #1-Figure 4a) and SIRT2^{-/-} OPCs (Reviewer
#1-Figure 4c). Besides, β -NMN improved the differentiation of G3 Terc^{-/-} and WT OPCs
(Reviewer #1-Figure 4b), but had no effect on SIRT2^{-/-} OPCs (Reviewer #1-Figure 4d). In brief,
using PBS as control does not change our conclusion when DMSO was used as control.

Reviewer #1-Figure 4 (Revised Fig. 4b, d & Revised Fig. 6c, e)

Similar results were yielded between using PBS as control and using DMSO as control. a-b, Quantification of proliferating OPCs (a, Ki67⁺Olig2⁺) cultured for 36 hours and differentiated oligodendrocytes (b, MBP⁺Olig2⁺) cultured for 48 hours from P0 WT mice and P0 G3 Terc^{-/-} mice (n=3). c-d, Quantification of proliferating OPCs (c, Ki67⁺Olig2⁺) cultured for 36 hours and differentiated oligodendrocytes (d, MBP⁺Olig2⁺) cultured for 48 hours from P0 SIRT2^{-/-} mice (n=3). β -NMN was dissolved in PBS. *p<0.05, **p<0.01, ***p<0.001 by one-way ANOVA followed by Tukey's post hoc test (a, b) or two-tailed t test (c, d). n.s.: no significance.

(3) Can a gradient in the substrate incubation be excluded for the immunoblot in Fig 1j?

Reply: Thanks to Reviewer #1 for the question. Fig. 1j is a representative of three independent WB images, all of which gave similar results, and is the part of the whole image. The unprocessed whole membrane blots of both SIRT2 and β -actin are shown in Reviewer #1-Figure 5. We had

327 excluded the possible artifacts caused by improper conditions such as uneven incubation
of the antibody during doing the experiment.

Reviewer #1-Figure 5

A representative set of unprocessed whole membrane blots excludes the gradient effect.

(4) Authors should show single channel images for Fig 1l to prove nuclear localization of
SIRT2 in OPCs because spherical SIRT2⁺ debris on the section could be misleading.

Reply: Thanks to Reviewer #1 for the advice. The single channel images for Fig 1l were actually
presented in Supplementary Figure 1d in our submission. Following advice of Reviewer #1, now
we have added the single channel images for Fig 1l and the previous Fig 1l has now been
replaced by Reviewer #1-Figure 6 which is shown below. Accordingly, we have modified the
text in the "Figure legends".

Reviewer #1-Figure 6 (Revised Fig. 1l)

The single channel images for immunofluorescence of SIRT2⁺ OPCs in the corpus callosum
of WT young (at age of 3 months) or old mice (at age of 21 months). N-L, non-lesion; 5 dpl, 5
344 days postlesion, and the demyelinated lesion was induced by focal injection of LPC. Scale bar, 10
μ m.

(5) Is remyelination after LPC in SIRT2^{-/-} mice permanently reduced? As OPC numbers are
normalized 10dpi (Fig 2j), could this indicate a transient effect that is normalized later?

Reply: Thanks to Reviewer #1 for raising this very interesting question. In Fig. 2j, we showed that
at 10 dpl, within the demyelinated lesion, in comparison with WT mice, the oligodendrocyte
lineage cell number (not OPCs) did not decrease in SIRT2^{-/-} mice, while mature oligodendrocytes
decreased in both density and proportion. In response to Reviewer #1, to test whether loss of
SIRT2 causes a transient effect on remyelination, we did new experiments and tested the effect on
remyelination in SIRT2^{-/-} mice at 30, 60, and 90 dpl in addition 21 dpl which has already been
shown in the submission version, and we have prepared Reviewer #1-Figure 7. We found that in

comparison with the WT mice at 21 dpl when remyelination has finished, the frequency of
 remyelinated axon (Reviewer #1-Figures 7a & b) and the thickness of newly formed myelin
 (Reviewer #1-Figures 7c & d) decreased at 21, 30, and 60 dpl in SIRT2^{-/-} mice, and reached
 similar level of the 21 dpl in WT mice at 90 dpl despite the frequency of normal-looking newly
 formed myelin is still lower in SIRT2^{-/-} mice group (Reviewer #1-Figure 7e). These data indicate
 that for SIRT2^{-/-} mice even when it took 4 folds of time, the remyelination still cannot fully reach
 the level of WT mice, suggesting that the reduction of remyelination in SIRT2^{-/-} mice is NOT a
 transient effect. Accordingly, we have these results to “Results” (lines 185-190).

Reviewer #1-Figure 7 (Revised Extended Data Fig.4)

The reduction of remyelination in SIRT2^{-/-} mice is NOT a transient effect. a, TEM micrographs within the lesions. Scale bar, 1 µm. b, Quantification of the proportion of remyelinated axons. c, Quantification G-Ratio. d, Quantification of individual G-Ratio distribution (linear regression). e, Quantification of myelin pathology level. n=6 for the WT 21 dpl group, n=5 for the SIRT2^{-/-} 21 dpl group, n=3 for the SIRT2^{-/-} 30 dpl group, n=3 for the SIRT2^{-/-} 60 dpl group, n=3 for the SIRT2^{-/-} 90 dpl group. All data are presented as mean ± SEM. * p<0.05, ** p<0.01, *** p<0.001 by one-way ANOVA followed by Tukey’s post hoc test (b, c). n.s., no significance.

(6) In the normal corpus callosum of mice about half of the axons remain non-myelinated. In Fig 2r the authors observe >55% remyelinated axons. What is this number is normalized to?
 Reply: Thanks to Reviewer #1 for pointing out this detail on method. Sturrock reported that in the mouse corpus callosum, the mean diameter of unmyelinated axons is more or less constant with an

379 overall mean diameter of $0.25 \pm 0.01 \mu\text{m}^{15}$. When we counted unmyelinated axons within the
380 demyelination lesion, we **only counted axons with a diameter greater than $0.25 \mu\text{m}$** . These
381 axons are more likely to be axons without remyelination, rather than axons that were
382 initially unmyelinated. We have made this clear in “**Methods**” (lines 666-670).

(7) Measurement of myelin pathology by staging in categories 1 and 2 refer to non-adhered
single myelin layers observed in TEM. The authors must show this effect is not due to
fixation or processing artifacts and provide a comprehensible analysis flow in the methods
section.

Reply: Thanks to Reviewer #1 for the questions about our TEM sample preparation process. The
advantage of fixing samples with glutaraldehyde is that it is fast. Samples which are relatively
easy to fix, such as the cerebral cortex, are fixed by 2.5% glutaraldehyde perfusion during sample
preparation. Myelin is not easy to fix due to its unique layered myelin structure which is
composed of 70% lipids and 30% protein, so we used 4% glutaraldehyde to perfuse. To minimize
the possibility that the loose myelin was likely caused by processing artifacts, the corpus callosum
samples were post fixed in 4% glutaraldehyde for 7 days, and the samples of both the
experimental group and the control group were strictly carried out at the same time undergoing
identical protocols. Myelin TEM is a standard procedure in our lab, and the processing artifacts
were excluded by increasing the sample size. Following the advice of Reviewer #1, we have
added myelin analysis flow in the “**Methods**” (lines 655-670).

(8) Fig 2q shows some mitochondrial enlargements and possible organelle accumulations in
the axons. Is there axonal pathology in $\text{SIRT2}^{-/-}$ and old WT mice?

Reply: Thanks to Reviewer #1 for raising a very interesting question about the impact of SIRT2
knockout on axonal pathology. To answer this question, we have quantified “normal-looking
axons” and “defective axons” in WT young, WT old, and $\text{SIRT2}^{-/-}$ mice. In this work, “defective
axons” was evaluated by: shrunk axons, mitochondria accumulation or swelling, myelin sphere
inclusion in axon or in the space between myelin and axon, and axon break. Only the axons with
diameter greater than $0.25 \mu\text{m}$ and were wrapped with myelin were analyzed. Results showed that
the proportion of normal-looking axons significantly decreased in $\text{SIRT2}^{-/-}$ and WT old mice
(Reviewer #1-Figure 8), indicating the axonal pathology in $\text{SIRT2}^{-/-}$ and WT old mice. These new
data confirmed that the speculation of Reviewer #1 was right. Accordingly, we have put Reviewer
#1-Figure 8 as Extended Data Fig.3 and added a sentence in “**Results**” (lines 179-184). The
evaluation of axonal pathology has been added in “**Methods**” (lines 671-674).

Reviewer #1-Figure 8 (Revised Extended Data Fig.3)

The percentage of normal-looking axons significantly decreased in SIRT2^{-/-} and WT old mice. Quantification of proportion of normal myelin in demyelinated lesion of WT young, WT old, and SIRT2^{-/-} mice at 21 dpl. n=5 in WT young mice group, n=5 in SIRT2^{-/-} mice group, n=7 in WT old mice group. All data are presented as mean ± SEM. *p<0.05, **p<0.01, ***p<0.001 by one-way ANOVA followed by Tukey's post hoc test. n.s.: no significance.

(9) NAD⁺ was one out of many metabolites altered in TERC^{-/-} OPCs (Fig 3b). Could a more general metabolic effect of body wide telomere shortening explain these global changes? Were similar alterations observed in other cells or organs of these mice? As the precursors of NAD⁺ (nicotinamide and nicotinate) were all detected at normal levels, what is the cause for NAD⁺ reduction?

Reply: Thanks to Reviewer #1 for asking this important question. A lot of evidence showed that NAD⁺ concentration decreases with age in worm, fly, mouse and human^{16,17}. In G3 Terc^{-/-} mice, in addition to OPCs, Zhu et. al. reported that the NAD⁺ level in cardiomyocytes of G3 Terc^{-/-} mice was also decreased compared to WT mice¹⁸. Recently study showed that modest tissue NAD⁺ depletion (median decrease ~30%) was observed in aged mice, while circulating NAD⁺ precursors were not significantly changed¹⁹. These data are consistent with our results, suggesting that the decline of NAD⁺ in the aged mice is not due to the decline in NAD⁺ synthesis. Instead, consumers of NAD⁺ are overactivated in aged mice, such as CD38²⁰ (activated by inflammatory) and PARP1²¹ (activated by DNA damage), which may reduce the concentration of NAD⁺ by increasing NAD⁺ consumption. In addition, in the nervous system, Sarm1 can degrade NAD⁺ and cause axonal degeneration, and activation of Sarm1 may also be an important reason for the reduction of NAD⁺^{22,23}.

(10) What is the reason for using primary rat OPCs for proteomics in Fig 3d when all other primary cell culture experiments were performed with mouse OPCs?

Reply: Thanks to Reviewer #1 for asking the question why we use primary rat OPCs for proteomics. In comparison with mouse OPCs, the amount of primary cultured rat OPCs is larger, which makes it easier to obtain enough protein and is more suitable for proteomics. That is why we used primary rat OPCs for proteomics. To resolve the concern of Reviewer #1, we have tested the effect of β-NMN treatment on differentiation of primary OPCs obtained from brain of both mouse and rat, and a similar effect was found (Reviewer #1-Figure 9). We have put Reviewer #1-Figure 9 as Extended Data Fig.9.

Reviewer #1-Figure 9 (Revised Extended Data Fig.9)

β -NMN promotes differentiation of primary OPCs obtained from brain of both rat and mouse in vitro. a, Quantification of differentiated oligodendrocytes (MBP⁺Olig2⁺) cultured for 48 hours from P0 SD rats. b, Quantification of differentiated oligodendrocytes (MBP⁺Olig2⁺) cultured for 48 hours from P0 mice. All data are presented as mean \pm SEM. * p <0.05, ** p <0.01, *** p <0.001 by two-tailed t test.

(11) The improved differentiation to oligodendrocytes observed in Fig 4c, d is not reflected by the proteomics data in Fig 3e, f (where no increase in myelin proteins e.g. MBP was detectable). Do the authors have an explanation?

Reply: Thanks to Reviewer #1 for pointing out this inconsistency. We extracted and analyzed the MBP data from our proteomics data and found there was a trend of increase in MBP protein level in the β -NMN group with the P value of 0.09 (Reviewer #1-Figure 10a). In response to Reviewer #1, we did new experiment to reevaluate the protein level of MBP using Western blot, and the results showed that in primary cultured rat OPCs, the protein level of MBP increased by supplementing β -NMN (Reviewer #1-Figure 10b). In addition, the immunofluorescence of MBP also showed that the expression of MBP was significantly improved by β -NMN treatment (Reviewer #1-Figure 10d). Reviewer #1-Figure 10b & 10c have been put as Revised Fig. 4e & 4f, respectively.

We think that the difference between the results of proteomics and Western blot is due to the different lysis buffer used. To avoid the possible blocking caused by too much detergents in proteomics, the lysis buffer used for proteomics is RIPA (Beyotime, #P0013B), which contains less detergents than the lysis buffer used for Western blot. Less detergents could cause less efficient protein extraction.

The components of the two lysis buffers are as follows:

RIPA, lysis buffer used for proteomics: 50 mM Tris (pH 7.4), 150 mM NaCl, 1% Triton X-100, 1% sodium deoxycholate, 0.1% SDS, inhibitors such as sodium orthovanadate, sodium fluoride, EDTA, leupeptin.

Lysis buffer used for Western blot: 50 mM Tris-HCl (pH 7.5), 5 mM EDTA, 1% SDS, 1% NP-40, 0.5% sodium deoxycholate, 3.69% CHAPS, 1% Triton X-100, 50 mM NaCl, 1 mM DTT (#3483-12-3, Biosharp), protease inhibitor (#18518900, Roche), phosphatase inhibitor A (#B15001-A, Biotool) and B (#B15001-B, Biotool), MG132.

Reviewer #1-Figure 10 (Revised Fig. 4e, f)

The protein level of MBP is **upregulated by supplementing β -NMN** to primary cultured rat
 OPCs. a, Proteomics showed the protein levels of MBP in primary cultures rat OPCs treated with
 DMSO or β -NMN for 48 hours (n=3). b-c, Western blots showed the protein levels of MBP in
 primary cultures rat OPCs treated with PBS or β -NMN for 48 hours (n=3). d,
 Immunofluorescence of differentiated oligodendrocytes (MBP⁺Olig2⁺) treated with DMSO or
 β -NMN for 48 hours. Scale bar, 50 μ m. All data are presented as mean \pm SEM. *p<0.05,
 **p<0.01, ***p<0.001 by two-tailed t test.

(12) Experiments searching for a molecular mechanism detected changes in ID4 mRNA
 abundance. Are these changes also detectable at the protein level? Do they result in changes
 of MBP expression?

Reply: Thanks to Reviewer #1 for this question. We have tested again the protein level of ID4 and
 MBP. Our new results showed that, in primary cultured rat OPCs, the protein level of ID4
 decreased by supplementing β -NMN (Reviewer #1-Figures 11a & b), which is consistent with
 the change of its mRNA level. In contrast, the protein level of MBP increased by supplementing
 β -NMN (Reviewer #1-Figures 11c & d). These data confirm that β -NMN downregulates ID4
 but upregulates MBP at protein level. Reviewer #1-Figure 11a & 10b have been put as
 Revised Fig. 7g & 7h, respectively.

Reviewer #1-Figure 11 (Revised Fig. 7g, h)

The protein level of ID4 is downregulated but MBP is upregulated by supplementing
 β -NMN to primary cultured rat OPCs. a-b, Western blots showed the protein levels of ID4 in
 primary cultures rat OPCs treated with DMSO or β -NMN for 48 hours (n=3). c-d, Western blots
 showed the protein levels of MBP in primary cultures rat OPCs treated with DMSO or β -NMN for
 48 hours (n=3).

References

- 1. Werner, H.B., *et al.* Proteolipid protein is required for transport of sirtuin 2 into CNS myelin. *J*
 *Neurosci* **27**, 7717-7730 (2007).
- 2. Cahoy, J.D., *et al.* A transcriptome database for astrocytes, neurons, and oligodendrocytes: a
 new resource for understanding brain development and function. *J Neurosci* **28**, 264-278

- (2008).
- 3. Zhang, Y., *et al.* An RNA-sequencing transcriptome and splicing database of glia, neurons,
and vascular cells of the cerebral cortex. *J Neurosci* **34**, 11929-11947 (2014).
- 4. Li, W., *et al.* Sirtuin 2, a mammalian homolog of yeast silent information regulator-2 longevity
regulator, is an oligodendroglial protein that decelerates cell differentiation through
deacetylating alpha-tubulin. *J Neurosci* **27**, 2606-2616 (2007).
- 5. Ji, S., Doucette, J.R. & Nazarali, A.J. Sirt2 is a novel in vivo downstream target of Nkx2.2 and
enhances oligodendroglial cell differentiation. *J Mol Cell Biol* **3**, 351-359 (2011).
- 6. Beirowski, B., *et al.* Sir-two-homolog 2 (Sirt2) modulates peripheral myelination through
polarity protein Par-3/atypical protein kinase C (aPKC) signaling. *Proc Natl Acad Sci U S A*
**108**, E952-961 (2011).
- 7. Chamberlain, K.A., *et al.* Oligodendrocytes enhance axonal energy metabolism by
deacetylation of mitochondrial proteins through transcellular delivery of SIRT2. *Neuron* **109**,
3456-3472 e3458 (2021).
- 8. Zhu, X., *et al.* Age-dependent fate and lineage restriction of single NG2 cells. *Development*
**138**, 745-753 (2011).
- 9. McLane, L.E., *et al.* Loss of Tuberous Sclerosis Complex1 in Adult Oligodendrocyte
Progenitor Cells Enhances Axon Remyelination and Increases Myelin Thickness after a Focal
Demyelination. *J Neurosci* **37**, 7534-7546 (2017).
- 10. Kosaraju, J., *et al.* Metformin promotes CNS remyelination and improves social interaction
following focal demyelination through CBP Ser436 phosphorylation. *Exp Neurol* **334**, 113454
(2020).
- 11. von Jonquieres, G., *et al.* Glial promoter selectivity following AAV-delivery to the immature
brain. *PLoS One* **8**, e65646 (2013).
- 12. Weinberg, M.S., Criswell, H.E., Powell, S.K., Bhatt, A.P. & McCown, T.J. Viral Vector
Reprogramming of Adult Resident Striatal Oligodendrocytes into Functional Neurons. *Mol*
*Ther* **25**, 928-934 (2017).
- 13. Chen, T.J., *et al.* In Vivo Regulation of Oligodendrocyte Precursor Cell Proliferation and
Differentiation by the AMPA-Receptor Subunit GluA2. *Cell Rep* **25**, 852-861 e857 (2018).
- 14. Neumann, B., *et al.* Metformin Restores CNS Remyelination Capacity by Rejuvenating Aged
Stem Cells. *Cell Stem Cell* **25**, 473-485 e478 (2019).
- 15. Sturrock, R.R. Myelination of the mouse corpus callosum. *Neuropathol Appl Neurobiol* **6**,

- 415-420 (1980).
- 16. Mouchiroud, L., *et al.* The NAD(+)/Sirtuin Pathway Modulates Longevity through Activation
of Mitochondrial UPR and FOXO Signaling. *Cell* **154**, 430-441 (2013).
- 17. Yoshino, J., Baur, J.A. & Imai, S.I. NAD(+) Intermediates: The Biology and Therapeutic
Potential of NMN and NR. *Cell Metab* **27**, 513-528 (2018).
- 18. Zhu, X., *et al.* Fine-Tuning of PGC1alpha Expression Regulates Cardiac Function and
Longevity. *Circ Res* **125**, 707-719 (2019).
- 19. McReynolds, M.R., *et al.* NAD(+) flux is maintained in aged mice despite lower tissue
concentrations. *Cell Syst* (2021).
- 20. Camacho-Pereira, J., *et al.* CD38 Dictates Age-Related NAD Decline and Mitochondrial
Dysfunction through an SIRT3-Dependent Mechanism. *Cell Metab* **23**, 1127-1139 (2016).
- 21. Massudi, H., *et al.* Age-associated changes in oxidative stress and NAD+ metabolism in
human tissue. *PLoS One* **7**, e42357 (2012).
- 22. Gerdts, J., Brace, E.J., Sasaki, Y., DiAntonio, A. & Milbrandt, J. SARM1 activation triggers
axon degeneration locally via NAD(+) destruction. *Science* **348**, 453-457 (2015).
- 23. Jiang, Y., *et al.* The NAD(+)-mediated self-inhibition mechanism of pro-neurodegenerative
SARM1. *Nature* **588**, 658-663 (2020).

Reviewer #2 (Remarks to the Author):

568 Ma et al. have carried out a substantial amount of work using various technical tools to address the
569 role of SIRT2 in improving myelination and remyelination in ageing. They show that loss of
570 SIRT2 and NAD⁺ in ageing and a mouse of model of premature ageing correlates well with poor
myelination and remyelination outcomes post a demyelination insult. Additionally, increasing
SIRT2 expression, especially the nuclear component in OPCs via administration of β NMN (NAD⁺
precursor) ameliorates the poor myelination and remyelination seen in ageing and the premature
ageing mouse model. This was observed via improved compound action potential propagation,
increased frequency of remyelinated axons, lowered g-ratios, reduced appearance of pathological
myelin and improved ‘compaction’ of myelination as shown by the myelin dense lines
measurements. They further show that β NMN administration in mice lacking SIRT2 shows
absolutely no change in OPC, OL or myelin outcomes post demyelination, indicating β NMN
mediates its effects through SIRT2. Finally using invitro experiments, the authors show β NMN
increases SIRT2 nuclear entry, decreased ID4 expression whereas knockdown of SIRT2 increases
acetylation of H3K18 and tubulin and increases ID4. They go on to show enrichment of SIRT2
and acetylated H3K18 on ID4 promoter. The authors propose that SIRT2 deacetylates H3K18,
inhibiting ID4 and this allows MBP transcription to go ahead leading to OPC differentiation into
OLs and hence improving myelin outcomes.

Overall, I am in agreement with the experiments and results, however, there are a few matters that
need to be addressed to elevate the manuscript to a better level.

(1) In figure1 relative mRNA level of Sirt2 in mature OLs in panels b and C are very
different 250 vs 750 while the levels of sirt2 are relatively similar in the OPCs. Do the
authors have an explanation for this?

Reply: We appreciate Reviewer #2 for agreeing with our experiments and results. Thanks to
Reviewer #2 for pointing out this inconsistency. In “Figure 1b”, the mRNA levels of sirtuins in
rat OPC group were normalized to “OPC-sirt1” while the mRNA levels of sirtuins in OL group
were normalized to “OL-sirt1”. To avoid the inconsistency caused by using different controls in
the same figure, now the mRNA levels of sirtuins in both rat OPC group and rat OL group have
been normalized to “OPC-sirt1” consistently, and we have replaced “Fig. 1b” with a new figure
(Reviewer #2-Figure 1). In this new figure, the level of sirt2 mRNA in rat mature OL is nearly 5
598 times of rat OPC, similar to the fold change shown Figure 1c.

Reviewer #2-Figure 1 (Revised Fig. 1b)

qRT-PCR of 7 members of sirtuins in primary cultured rat OPC and mature
oligodendrocyte (OL) from the cortex of P0 rat (n=3). All groups were normalized to
“OPC-sirt1” group.

(2) Additionally, these experiments carried out using OPCs from the rat cerebral cortex.
This is a curious choice since the authors show other data in the manuscript using cultured
mouse OPCs.

Reply: Thanks to Reviewer #2 for the question why we use primary rat OPC for qRT-PCR. In
comparison with mouse OPC, the amount of primary cultured rat OPC is larger, which makes it
easier to obtain enough mRNA and is more suitable for qRT-PCR. That’s why we used primary
rat OPC.

To resolve the concern of Reviewer #2, we have done new experiment and tested the mRNA
level of 7 members of sirtuins in primary mouse OPC and mouse mature oligodendrocyte (OL)
which were cultured from the brain of P0 mouse. Our new results showed that among the 7
members of sirtuins, the abundance of sirt2 mRNA dominates in both mouse OPC and
mouse mature OL, with a much higher level in the mature OL (Reviewer #2-Figure 2). These
results obtained from mouse OPC and OL mirror with high fidelity the results obtained from rat
OPC and OL, respectively (compare Reviewer #2-Figure 1 Vs Reviewer #2-Figure 2/ Fig. 1b),
suggesting the high similarity between mouse and rat on both OPC and mature OL. We have
put this figure as Revised Extended Data Fig.1a and added it in “Results” (lines 108-109).

Reviewer #2-Figure 2 (Revised Extended Data Fig.1a)

qRT-PCR of 7 members of sirtuins in primary cultured mouse OPC and mouse mature
oligodendrocyte (OL) from the cortex of P0 mouse (n=3). All groups were normalized to
“OPC-sirt1” group.

(3) Graphs in Figure 1e, 1m & n show that a large proportion but not all the OPCs show
SIRT2 nuclear localization during development and demyelination/remyelination phases. Do
the authors have a reason as to why that might be the case? Since most OPCs in a
demyelinated lesion would either differentiate or die it is likely that only OPCs meant to be
differentiating would have nuclear SIRT2. Is it a case of timing? i.e., because it is transient
expression all of them do express SIRT2 but only a fraction may be recorded at any
snapshot?

Reply: Reviewer #2 raised a very interesting question-why only a proportion of OPCs express
SIRT2 in nuclear during postnatal development and within the demyelination lesion? This is still a
puzzle and certainly deserves further study, and we plan to explore this question which will be an
independent interesting story itself. Reviewer #2 suggested a very interesting explanation with
which we agree: only a fraction of OPCs were recorded SIRT2 nuclear positive as a snapshot.

Considering that myelination during postnatal development or remyelination post demyelination
can effectively take place depending on continuous OPCs activation, and it is not a one-go process.
It is logical to think that at a certain point of time there is a fraction of OPCs are activated.
Therefore, it is not surprising that only a proportion of OPCs show nuclear localization of SIRT2.
(4) In figure 2h & I, a significant drop in the proliferative ability of OPCs is seen in the
SIRT2^{-/-} mice 5dpl. However, in WT or G3Terc^{-/-} mice supplemented with βNMN there is no
change in OPC's proliferative ability. However, the authors make no attempt to explain or
pursue this finding.

Reply: Thanks to Reviewer #2 for rightly pointing out that a significant drop in the proliferative
ability of OPCs is seen in the SIRT2^{-/-} mice at 5 dpl (Reviewer #2-Figures 3a-c). In comparison,
we can see that a more significant drop in the differentiation of OPCs is seen in the SIRT2^{-/-} mice
at 10 dpl (Reviewer #2-Figures 3d-f). These results suggest that for SIRT2^{-/-} mice impaired
differentiation of OPCs is more serious than impaired proliferation of OPCs. Therefore, in
this work we focus more on how SIRT2 affects differentiation of OPCs. Accordingly, we have
added "and the differentiation of OPCs is more severely impaired." in "Results" (Line 159).

In response to the question raised by Reviewer #2, to explore whether supplementing β-NMN
affect proliferation and/or differentiation of OPCs in wild type (WT) young mice in vivo, we did
new experiment. Our new results showed that supplementing β-NMN to WT young mice in vivo
did not affect OPC's proliferation (Reviewer #2-Figures 3g&h) but enhanced OPC's
differentiation (Reviewer #2-Figures 3i&j). Together with these new data, our results indicate
that β-NMN enhances differentiation of OPCs but does not affect proliferation of OPCs in
both WT and G3 Terc^{-/-} mice in vivo. Accordingly, we have added "and wild type" In "Results"
(line356-367) and Reviewer #2-Figures 3g&h have been put as Revised Fig. 6e&h, Reviewer
#2-Figures 3i&j have been put as Revised Fig. 6j&m. Since the bottleneck problem for failed
remyelination, such as in the aged CNS, is insufficient differentiation of OPCs¹, this highlights the
importance of β-NMN on enhancing differentiation of OPCs.

We are also aware that loss of SIRT2 impairs proliferation of OPCs in vivo and this deserves
further research, and plan to explore this further in future study.

Reviewer #2-Figure 3 (Revised Fig. 6)

 β -NMN enhances differentiation of OPCs but does not affect proliferation of OPCs in both
 WT and $G3\ Terc^{-/-}$ mice in vivo. a, Schematic diagram of the experiment for testing OPC
 proliferation in vivo at 5 dpl. b-c, Quantification of proliferating OPCs (Ki67⁺Olig2⁺) within the
 demyelination lesions at 5 dpl (n=5 for WT group, n=4 for the SIRT2^{-/-} group). d, Schematic diagram
 of the experiment for testing OPC differentiation in vivo at 10 dpl. e-f, Quantification of differentiated
 oligodendrocytes (CC1⁺Olig2⁺) within the demyelination lesions at 10 dpl (n=5 for WT group, n=4 for
 the SIRT2^{-/-} group). g, Schematic diagram of the experiment for testing OPC proliferation in vivo at 5
 dpl. h, Quantification of proliferating OPCs (Ki67⁺Olig2⁺) within the demyelination lesions at 5 dpl
 (n=4 for the WT+PBS group and the WT+ β -NMN group, n=5 for the SIRT2^{-/-}+PBS group and the
 SIRT2^{-/-}+ β -NMN group). i, Schematic diagram of the experiment for testing OPC differentiation in
 vivo at 10 dpl. j, Quantification of differentiated oligodendrocytes (CC1⁺Olig2⁺) within the
 demyelination lesions at 10 dpl (n=4 for the WT+PBS group, n=5 for the WT+ β -NMN group, the
 SIRT2^{-/-}+PBS group and the SIRT2^{-/-}+ β -NMN group). All data are presented as mean \pm SEM.
 *p<0.05, **p<0.01, ***p<0.001 by two-tailed t test (b, c, e, f) or one-way ANOVA followed by
 Tukey's post hoc test (h, j).

 (5) In the results and discussion authors state that SIRT2 is required & necessary for
 remyelination however this is not strictly true since a 25-30% remyelination frequency is
 seen in SIRT2^{-/-} LPC lesion compared to ~55% in age matched WT mice (figure 2r) and 20%
 of remyelinated axons in SIRT2 group show no (grade 0) myelin pathology (figure 2s). Also,
 it is not simply the loss of SIRT2 (line 356) but also the inability to efficiently upregulate and
 recruit SIRT2 to the nucleus that causes impaired remyelination as seen in figure 1e and 1m
 and 1n where, by P21 there is almost no expression of SIRT2 in OPC cytoplasm or nucleus
 however mice that young can remyelinate very efficiently. Clarity of words is important in
 the take home message.

Reply: Thanks to Reviewer #2 for pointing out the words we used in “Results” and “Discussion”
concerning the role of SIRT2 in remyelination. In response to Reviewer #2, to clearly investigate
the role of SIRT2 specifically in OPCs in vivo, we used $Sirt2^{flox/flox}$ mice in which two loxP sites
flank three critical exons in the $Sirt2^{flox}$ allele of $Sirt2^{flox/flox}$ mice (Reviewer #2-Figure 4a). Mice
that are homozygous for the $Sirt2^{flox}$ allele appeared normal and were fertile. To specifically delete
SIRT2 in OPCs, the $Sirt2^{flox/flox}$ mice were crossed with $NG2-Cre^{ERT}$ mice, and the heterozygous
offspring were crossed again to obtain $NG2-Cre^{ERT}; Sirt2^{flox/flox}$ mice. It took us about 5 months to
obtain $NG2-Cre^{ERT}; Sirt2^{flox/flox}$ mice. The mice were intraperitoneally injected with tamoxifen
(Sigma, #10540-29-1, 100mg/kg) for 5 days²⁻⁴ so that Cre recombinase-mediated excision creates
a nonfunctional allele exclusively in OPCs in temporally controlled manner, and the tamoxifen
was dissolved in a 9:1 ratio of corn oil: ethanol. Four days post the last administration of
tamoxifen^{3,4}, we induced demyelination by focal injection of lysolecithin (LPC) into the corpus
callosum and detected remyelination at 21 dpl with transmission electron microscope (TEM,
Reviewer #2-Figure 4b).

We first confirmed the genetic deletion of SIRT2 in OPCs by immunofluorescence (Reviewer
#2-Figure 4c). Our results showed that in the demyelinated lesion, SIRT2 was not detected in the
OPCs of $NG2-Cre^{ERT}; Sirt2^{flox/flox}$ mice that received tamoxifen injection but SIRT2 was indeed
detected in the nuclei of OPCs of $Sirt2^{flox/flox}$ mice (Reviewer #2-Figure 4c). Then we tested the
remyelination efficiency by TEM. Our new results showed that conditional knockout of SIRT2 in
OPCs significantly impaired myelin repair (Reviewer #2-Figure 4d), reduced the frequency of
remyelinated axon (Reviewer #2-Figure 4e), and decreased the thickness of newly formed myelin
(Reviewer #2-Figure 4f). The pathological analysis of myelin showed that the frequency of
normal-looking newly formed myelin reduced while the grade 1 and grade 2 myelin increased by
conditional knockout SIRT2 in OPCs (Reviewer #2-Figure 4g).

These new results confirmed that the SIRT2 expressed in OPCs plays a critical role in
myelin repair, providing direct evidence that SIRT2 in OPCs plays a critical role in
remyelination. These new results **have been put as Revised Extended Data Fig. 5** Together with
our new data, we also have carefully re-evaluated relevant data concerning the role of SIRT2 in
remyelination, we agree with the comments of Reviewer #2. Accordingly, we have modified the
related result title into “SIRT2 is **critical** for remyelination” in **line 152**, and “Our in vitro and in
vivo data indicate that SIRT2 is **critical** for remyelination” in the content in **line 177**. Together
with our new data on conditional knockout of SIRT2 specifically in OPC, we have added a
paragraph in “Results” (**lines 191-202**) and a paragraph in “Discussion” (**lines 420-429**)

Reviewer #2-Figure 4 (Revised Extended Data Fig. 5)

Conditional SIRT2 knockout specifically in OPCs of mouse verified that SIRT2 plays a

critical role in remyelination. a, Schematic illustration of the conditional *Sirt2* allele with loxP

sequences (red) flanking exons 5–7 (black). The Flp recombinase recognition sequences (FRT site)

are showed in blue. b, The flow chart of experimental design. Tamoxifen was given (100 mg/kg,

intraperitoneal daily) for 5 days started at age of 2 months, and 4 days later, LPC was focally

injected into the corpus callosum to induce demyelination. The mice were sacrificed at 21 days

post LPC injection. c, Immunofluorescence of SIRT2 and PDGFRα⁺ OPCs within the

demyelinated lesion in the corpus callosum of the *Sirt2*^{fl/fl} mice and the *NG2-Cre^{ERT2};*

Sirt2^{fl/fl} mice at 21 dpl. Scale bar, 1 μm. d, TEM micrographs within the lesions at 21 dpl.

Scale bar, 1 μm. e, Quantification of the proportion of remyelinated axons. f, Quantification of

individual G-Ratio distribution (linear regression). g, Quantification of myelin pathology level.

n=3 for the *Sirt2*^{fl/fl} mice group, *n*=4 for the *NG2-Cre^{ERT2};* *Sirt2*^{fl/fl} mice group. All data are

presented as mean ± SEM. * *p*<0.05 by two-tailed *t* test (e).

(6) Myelin ultrastructure of *G3Terc*^{-/-} mice T 6mo is significantly worse than that of 18mo

WT mice (extended data Figure 3g) so there is certainly more than simply ageing going on in

G3Terc^{-/-} mice.

Reply: Thanks to Reviewer #2 for rightly pointing out that there is more than simply ageing going

on in *G3Terc*^{-/-} mice. *G3Terc*^{-/-} mice exhibit systemic shortening of telomere length therefore is a

747 systemic premature aging model which not only involves CNS but also involves almost all
748 systems^{5,6}. In addition, ageing of WT mice is slower and milder, and 18-month-old mice are
749 generally regarded as an early stage of ageing while mice over 24 months old are regarded as aged
mice.

**(7) In several places in the manuscript authors state that β NMN or nuclear SIRT2 in OPCs**
**rejuvenates myelin. This I believe is mediated through the myelin made by the newly**
**differentiating OLs (ones coming from rejuvenated OPCs) that make a new ‘superior**
**quality’ myelin. Often, one gets the impression from the language used in the paper that the**
**old poor-quality myelin is mended or fixed by administering β NMN. Do the authors mean to**
**imply that NAD⁺ supplementation can actually repair grade 1 and possibly grade 2 myelin**
**back to grade 0? And if so, how?**

Reply: Thanks to Reviewer #2 for pointing out the misleading wording. What we want to say is
that β -NMN targets OPCs and promotes the differentiation of OPCs into new
oligodendrocytes, and the mature oligodendrocytes make new “superior quality” myelin. To
clarify the meaning, we have modified the relevant wording by adding more words in lines 95-96
in “Introduction”, lines 399-400 in “Results”, and lines 468-469 in “Discussion”: “by promoting it
differentiating into mature oligodendrocyte and eventually enhances new myelin generation in the
aged CNS”.

**(8) The authors do not show the in vivo effects of β NMN addition on OPC proliferation and**
**differentiation in G3 *Terc*^{-/-} mice or Old WT mice in un-lesioned animals. Only improved**
**myelin outcomes are shown (figure 4e-t).**

Reply: Thanks to Reviewer #2 for the question. To address this question, we have done new
experiments and detected the density of OPCs and differentiated oligodendrocytes in the corpus
callosum of non-lesioned normal WT young (6 M) and G3 *Terc*^{-/-} (6 M) mice. We found that in
non-lesion condition, in comparison with WT young mice, the density of OPCs (Reviewer
#2-Figure 5a), the differentiated oligodendrocytes (Reviewer #2-Figure 5b) and the
oligodendrocyte lineage cells (Reviewer #2-Figure 5c) decreased significantly in G3 *Terc*^{-/-} mice.
In non-lesioned condition, β -NMN increased the density of OPCs, the differentiated
oligodendrocytes, and the oligodendrocyte lineage cells (Reviewer #2-Figures 5a-c). The
proportion of OPCs (Reviewer #2-Figure 5d) and the differentiated oligodendrocytes (Reviewer
#2-Figure 5e) in the oligodendrocyte lineage cells did not change in G3 *Terc*^{-/-} mice while
β -NMN improved the proportion of the differentiated oligodendrocytes (Reviewer #2-Figure 5e).
For oligodendrocyte lineage cells, their proportion in all cells of corpus callosum decreased in G3
*Terc*^{-/-} mice and β -NMN restored their proportion (Reviewer #2-Figure 5f).

In non-lesioned WT old (21 M) mice, the effects of β -NMN mirror with high fidelity all the
above indexes in G3 *Terc*^{-/-} (6 M) mice (Reviewer #2-Figures 5g-l). Furthermore, all the indexes
impaired in both G3 *Terc*^{-/-} (6 M) mice and WT old (21 M) mice were fully restored to the levels
of WT young mice (6 M), showing the rejuvenating effects of β -NMN on oligodendrocyte lineage
cells. Together, these data indicate that:

1) In non-lesion condition, β -NMN promotes the differentiation of OPCs in both G3 *Terc*^{-/-}
mice and WT old mice;

2) In non-lesion condition, β -NMN restores the age-related decline indexes in both G3
 $Terc^{-/-}$ mice and WT old mice, including the density of OPCs, oligodendrocytes and the
 oligodendrocyte lineage cells and the proportion of the oligodendrocyte lineage cells among all
 cells, to the level of young mice. These results confirm that β -NMN rejuvenates OPCs in both
 G3 $Terc^{-/-}$ mice and WT old mice.

In non-lesioned normal corpus callosum of mice, we found that few $Ki67^+$ cells can be detected
 (data not shown), indicating that the proliferation efficiency of OPCs is very low in physiological
 condition. This made quantification of the proliferating OPCs ($Ki67^+PDGFR\alpha^+$ cells) very
 difficult.

We have added lines 267-278 in “Results” and Reviewer #2-Figure 5 has been put as Revised
 Extended Data Fig.10.

 Reviewer #2-Figure 5 (Revised Extended Data Fig.10)

In non-lesion condition, β -NMN promotes the differentiation of OPCs and rejuvenates the
 aged OPCs in both G3 $Terc^{-/-}$ mice and WT old mice. a-f, Quantification of the density and
 proportion of OPCs (a, d, PDGFR α^+ Olig2 $^+$), mature oligodendrocytes (b, e, CC1 $^+$ Olig2 $^+$) and
 oligodendrocyte lineage cells (c, f, Olig2 $^+$) in corpus callosum of the WT young mice and G3
 $Terc^{-/-}$ mice. n=5 per group. g-l, Quantification of the density and proportion of OPCs (g, j,
 PDGFR α^+ Olig2 $^+$), mature oligodendrocytes (h, k, CC1 $^+$ Olig2 $^+$) and oligodendrocyte lineage cells
 (i, l, Olig2 $^+$) in corpus callosum of the WT young and WT old mice. n=5 per group. All data are
 presented as mean \pm SEM. * p < 0.05, ** p < 0.01, *** p < 0.001 by one-way ANOVA followed by
 Tukey's post hoc test.

(9) If SIRT2 facilitates MBP expression, and MBP protein is produced throughout the
lifetime of OL, it is possible that older pre-existing OLs may be able to compact myelin
better with increased MBP expression through **greater SIRT2 availability when β NMN in**
**added.** This could happen even in absence of any change in proliferation or differentiation.

Reply: Thanks to Reviewer #2 for raising a very interesting question which deserves further study,
and the essence of this question is what the role SIRT2 plays in mature oligodendrocytes. More
specifically, whether β -NMN facilitates MBP expression in pre-existing oligodendrocytes and
produce compact myelin. To address this question, we did new experiment and showed that
β -NMN significantly promoted MBP protein level in the mature oligodendrocytes in the
differentiation condition in vitro using primary OPCs culture from both rat and mouse
(Reviewer #2-Figure 6), and this occurs in parallel to β -NMN increased nuclear entry of SIRT2
in OPCs at an earlier time. Our new data indicate that β -NMN facilitates MBP expression by
mature oligodendrocytes at least in vitro. We have put Reviewer #2-Figure 6 as Revised
Extended Data Fig.9.

Reviewer #2-Figure 6 (Revised Extended Data Fig.9)

β -NMN facilitates MBP expression by mature oligodendrocytes of both rat and mouse in
vitro. a, Quantification of differentiated oligodendrocytes (MBP+Olig2+) cultured for 48 hours
from P0 SD rats. b, Quantification of differentiated oligodendrocytes (MBP+Olig2+) cultured for
48 hours from P0 mice. All data are presented as mean \pm SEM. * p <0.05, ** p <0.01, *** p <0.001
by two-tailed t test.

However, to address whether β -NMN facilitates MBP expression and produce compact myelin
by pre-existing mature oligodendrocytes, it has to be done in vivo using conditional deletion of
SIRT2 specifically in mature oligodendrocytes. For this purpose, we have started breeding mice in
which SIRT2 is intended to be specifically deleted in mature oligodendrocytes. This line of mouse,
Plp-Cre^{ERT}; Sirt2^{flox/flox}, is still in breeding, and it will take much longer time. Considering the
focus of the current work is the role of SIRT2 in OPCs and the deadline for revision has been
extended twice already, we feel that the results on the specific role of SIRT2 in mature
oligodendrocytes are more suitable to be presented later in another independent story. We greatly
appreciate understanding of Reviewer #2 on this point.

(10) Long term treatment with β NMN has very similar outcomes to immediate as well as
delayed β NMN administration. This is an interesting finding showing that requirement of
upregulated nuclear SIRT2 may be only necessary only immediately prior or at the point of
differentiation. The total number of Olig2+ cells in G3 Terc^{-/-} mice treated with β NMN (long

term and immediate) is similar to PBS treated animals 5dpl and 10dpl, as are numbers of
proliferating OPCs and proportion of proliferating OPCs. However, at 10dpl there are more
differentiated OLs (in immediate) and proportionally more OLs compared to OL lineage
cells. What population is giving rise to these differentiating cells? Non-proliferating OPCs?
In absence of total OPC density data in the lesion (at 5dpl and 10dpl) it is difficult to imagine
how the proliferating OPCs numbers are similar, total OL (CC1⁺) density is greater but the
total Olig2⁺ lineage is balanced.

Reply: Thanks to Reviewer #2 for asking a very important question. To address this question, we
did new experiment and quantified the density and proportion of OPCs at 5 dpl and 10 dpl. We
showed that at 5 dpl when the proliferation of OPCs peaks in the demyelination model, the density
of oligodendrocyte lineage cells, the density and proportion of both the total OPCs and the
proliferating OPCs did not change in the lesion of G3 *Terc*^{-/-} mice by β -NMN supplementation
(Reviewer #2-Figures 7a-e). At 10 dpl when the differentiation of OPCs peaks in the
demyelination model, the density of oligodendrocyte lineage cells did not change (Reviewer
#2-Figures 7f&g), while the proportion of the differentiated oligodendrocytes increased
(Reviewer #2-Figure 7h). Interestingly, in comparison with PBS, at 10 dpl, our new data showed
that β -NMN supplementation caused a reduction in both the density and the proportion of OPCs
within the lesion of G3 *Terc*^{-/-} mice, inversely correlated with an increase in the proportion of
differentiated mature OLs (Reviewer #2-Figures 7h-j), providing evidence that at 10 dpl, the
differentiated mature OLs originated from the differentiation of OPCs. Our results indicate that in
G3 *Terc*^{-/-} mice, β -NMN enhances OPCs differentiate into mature oligodendrocytes while does not
affect proliferation of OPCs.

In the non-lesioned condition, in WT young mice, the density of OPCs in the normal corpus
callosum is about 120 cells/mm². When demyelination occurs, OPCs residing the surrounding
brain regions migrate into the lesion region and proliferate to expand their number to ~320
cells/mm² at 5 dpl (Reviewer #2-Figure 7d). At 10 dpl when differentiation of OPCs peaks and
remyelination starts, a large number of OPCs differentiated into mature oligodendrocytes and the
density of OPCs within the demyelinated lesion decreased from ~320 cells/mm² at 5 dpl to 185
cells/mm² at 10 dpl (Reviewer #2-Figure 7i), accompanying an increase in mature OLs
(Reviewer #2-Figure 7: b Vs g; c Vs h). Interestingly, within the lesion of G3 *Terc*^{-/-} mice, at 5
dpl when proliferation of OPCs peaks, both the density and the proportion of OPCs were the same
to the WT mice (Reviewer #2-Figures 7d&e), indicating that the number and proportion of OPCs
are not affected in G3 *Terc*^{-/-} mice. However, from 5 dpl to 10 dpl, both the density and the
proportion of OPCs remained unchanged in G3 *Terc*^{-/-} mice, and there were more OPCs in the
lesion in G3 *Terc*^{-/-} mice than WT young mice (Reviewer #2-Figure 7: d Vs i; e Vs j),
accompanying a reduction of both the density and the proportion of mature OLs within the lesion
(Reviewer #2-Figure 7: b Vs g; c Vs h), indicating that impaired differentiation of OPCs is the
key pathology for G3 *Terc*^{-/-} mice. Supplementation of β -NMN did not affect the density and the
proportion of both proliferating OPCs and the total OPCs at 5 dpl (Reviewer #2-Figures 7b-e)
while significantly increased the density and the proportion of mature OLs and decreased the
density and the proportion of OPCs at 10 dpl (Reviewer #2-Figures 7f-j), indicating that β -NMN
promoted OPCs to differentiate into mature OLs in G3 *Terc*^{-/-} mice. Taken together, our data
showed that the increased number of mature oligodendrocytes comes from the differentiation
of OPCs. Whether these OPCs are proliferating or non-proliferating, it needs further study using

techniques such as cell tracking, which is more suitable to be addressed in an independent study.
 We have added lines 292-296 to “Results” and Reviewer #2-Figure 7 has been put as Revised
 Extended Data Fig.11.

Reviewer #2-Figure 7 (Revised Extended Data Fig.11)

The differentiated mature oligodendrocytes come from the differentiation of OPCs. a-e,
 Quantification of oligodendrocyte lineage cells (a), proliferating OPCs (b, c) and total OPCs (d, e)
 within the demyelination lesions at 5 dpl (n=5 for the WT young group and the G3 Terc^{-/-}+PBS
 group, n=4 for the G3 Terc^{-/-}+β-NMN group). f-j, Quantification of oligodendrocyte lineage cells
 (f), differentiated oligodendrocytes (g, h) and total OPCs (i, j) within the demyelination lesions at
 10 dpl (n=4 for the WT young group and the G3 Terc^{-/-}+PBS group, n=5 for the G3
 Terc^{-/-}+β-NMN group). All data are presented as mean ± SEM. *p<0.05, **p<0.01, ***p<0.001
 by one-way ANOVA followed by Tukey’s post hoc test.

(11) In the material and methods under immunofluorescence it says that primary antibodies
 were incubated for 48-72 hours. Since the tissue sections are 12 μm thick, 3 days of
 incubation seems excessive and may produce labelling artefacts. Post-mortem human tissues
 for antigen detection often require long exposures, but in this study the human tissue was
 incubated only for 12 hours according to the methods.

Reply: Thanks to Reviewer #2 for the question concerning details on the immunofluorescence
 procedure. Before the formal experiment, we did preliminary experiment to determine the best
 time for incubation of antibodies. It is true that the 12-hour incubation time had a signal, but
 48-hour incubation yielded more complete morphology of SIRT2 positive structure without
 nonspecific labelling. To resolve the concern of Reviewer #2, we redid immunofluorescence in
 consecutive brain slices of LPC-injected WT mice brain at 10dpl, and the primary antibodies
 (Olig2, green; SIRT2, red; DAPI, blue) were incubated for 12 hrs, 48 hrs, and 72 hrs, respectively.
 The results showed that in comparison with no primary antibody control, the non-specific
 labelling was fully excluded. The expression pattern of SIRT2 of different incubation times was
 basically the same, and Olig2 was also the same (Reviewer #2-Figure 8). Under the same
 exposure conditions, the longer the incubation time of the SIRT2 antibody, the more complete
 morphology of SIRT2 positive structure were seen. These data indicate that as long as the
 labelling condition remains identical among groups for comparison, 72 hrs primary antibody
 incubation does not yield nonspecific labelling.

Reviewer #2-Figure 8

Primary antibody incubation for 72 hrs does not yield nonspecific labelling. Immunofluorescence of Olig2 and SIRT2 in consecutive brain slices of LPC-induced demyelination lesion in WT mice brain at 10dpl, and the primary antibodies (Olig2, green; SIRT2, red; DAPI, blue) were incubated for 12 hrs, 48 hrs, and 72 hrs, respectively.

We also analyzed the proliferation and differentiation of OPCs in demyelinated region of corpus callosum in G3 *Terc*^{-/-} mice at 5 dpl and 10dpl. We found that the results of antibody incubation for 12 hours are consistent with the results of incubation for 48 hours. Supplementation of β -NMN did not change the proliferation of OPCs (Reviewer #2-Figures 9a-f), while β -NMN improved differentiation of OPCs (Reviewer #2-Figures 9g-l) in demyelinated lesion in the corpus callosum of G3 *Terc*^{-/-} mice, consistent with results in our original submission.

In most cases, samples from post-mortem human brain do not give strong signals largely due to over-fixation during pathological process. In order to amplify signal, biotin conjugated second antibody was used in immunocytochemistry of post-mortem human brain (avidin-biotin complex immunofluorescence method, ABC method). This ABC methods not only amplify signal but also enhances background signal. To reduce the background signal, we chose a 12-hour incubation time for antibodies staining in human brain samples.

Reviewer #2-Figure 9

Primary antibody incubation for 12 hours yielded results consistent with antibody incubation for 48 hours. a-c, The proliferation of OPCs in demyelinated lesion in the corpus callosum of G3 Terc^{-/-} mice at 5 dpi. The primary antibodies were incubated for 12 hrs. d-e, The proliferation of OPCs in demyelinated region of corpus callosum in G3 Terc^{-/-} mice at 5 dpi. The primary antibodies were incubated for 48 hrs. g-i, The differentiation of OPCs in demyelinated lesion in the corpus callosum of G3 Terc^{-/-} mice at 10dpi. The primary antibodies were incubated for 12 hrs. j-l, The differentiation of OPCs in demyelinated region of corpus callosum in G3 Terc^{-/-} mice at 10dpi. The primary antibodies were incubated for 48 hrs. All data are presented as mean ± SEM. *p<0.05, **p<0.01, ***p<0.001 by two-tailed t test.

(12) I am amazed at almost 16 weeks of daily i.p injection in a mouse not producing significant peritonitis or damage.

Reply: Thanks to the comments of Reviewer #2. 16 weeks of daily i.p. injection is indeed a long-term process. Luckily, we did not observe significant peritonitis or damage after β-NMN injection. Long-term intraperitoneal (i.p.) injection of β-NMN in mice has also been reported in other studies⁷⁻¹⁰ and β-NMN does not cause peritonitis. This did not happen without efforts, in fact, in order to reduce the chance which could lead to peritonitis or damage caused by daily i.p injection in mice, researchers were first trained by the experienced researchers with details on the experimental protocol. Important details include abdominal massage before injection, disinfection, hold the skin to avoid possible wrong injection into any organs and post injection gentle massage. As routine, the control group were i.p. injected with PBS.

(13) The authors need to provide the standard definition of g-ratio under assessment of remyelination and then they can explain how they calculated it using by tracing inner and outer circles. As it stands right now it looks as if myelin thickness is being measured. The

972 measurement of axonal diameter is mentioned later as an aside and it may be confusing to
973 the uninitiated as to how this figure into the g-ratio calculation. It would be helpful, if a
974 mention was made that only axons displaying grade 0 myelin pathology were assessed for
g-ratios, (I assume that this was the case). This is important because there are notable
number of axons displaying grade 1 and 2 pathology levels even in WT young mice.

Reply: Thank Reviewer #2 for this suggestion. In response, we have added a detailed method on
G-Ratio measurement and calculation in lines 655-670 in the “Methods” and the schematic
diagram was added to “Revised Extended Data Fig.14”.

The details are as follows:

Assessment of remyelination. G-Ratio was measured on transmission electron micrographs
using ImageJ software¹¹. 70-100 normal myelinated (grade 0) axons were analyzed per mouse.
The myelin sheath was regarded as a torus, and the areas of inner and outer circles of myelin
sheath were measured with freehand tool on ImageJ by tracing the outer surface of each structure,
then converted the areas into hypothetical radius. The G-Ratio was calculated as the ratio of the
radius of inner circle over that of outer circle on the same myelin (Reviewer #2-Figure 10), which
is inversely correlated with myelin thickness. The G-Ratio of unmyelinated and demyelinated
axons is 1, which is the maximum value for G-Ratio. The diameter of axons was calculated by
area, which was also measured by tracking the outer surface of axons. Axon diameters were used
in our scatter plot of axon diameter and G-Ratio, which represent the linear relationship between
G-Ratio and axon diameter. The distance between DL was calculated by dividing the distance from
the outermost DL to the innermost DL by the number of DL.

Reviewer #2-Figure 10 (Revised Extended Data Fig.14).

The schematic diagram of G-Ratio. R1: radius of inner circle (blue); area1: area of inner circle
(blue); R2: radius of outer circle (green); area2: area of outer circle (green); D-axon: diameter of
axon (red); area3: area of axon (red).

$R1 = \sqrt{\text{area}1/\pi}$; $R2 = \sqrt{\text{area}2/\pi}$; $G - Ratio = R1/R2$; $D - axon = 2\sqrt{\text{area}3/\pi}$

(14) I think the flow of information in the model shown in figure 8 could be improved, it not
**easy to follow in the ‘on’ panel.**

Reply: Thanks to Reviewer #2 for the helpful advice. In response, we have made substantial
modifications to Figure 8 in order to make it easy to follow. The revised figure is shown below
(Reviewer #2-Figure 11), and we hope now the revised Figure 8 is clear and easy to understand.

NAD⁺ enhances SIRT2 nuclear entry and rejuvenates aged OPC

Reviewer #2-Figure 11 (Revised Fig. 8)

A working model for NAD⁺ on rejuvenating the aged OPC.

(15) The manuscript is simply and clearly written which makes for a good reading, however,
 there are a fair number of grammatical errors that when fixed will make for a much
 smoother reading.

Reply: Thanks to Reviewer #2 for reminding on grammatical errors. In response, we have
 carefully checked the full manuscript and have revised in quite many places which are labeled in
 yellow colour in the main text.

**References**

1. Franklin, R.J.M. & Ffrench-Constant, C. Regenerating CNS myelin - from mechanisms to
 experimental medicines. *Nat Rev Neurosci* **18**, 753-769 (2017).

2. Zhu, X., *et al.* Age-dependent fate and lineage restriction of single NG2 cells. *Development*
 **138**, 745-753 (2011).

3. McLane, L.E., *et al.* Loss of Tuberous Sclerosis Complex1 in Adult Oligodendrocyte
 Progenitor Cells Enhances Axon Remyelination and Increases Myelin Thickness after a Focal
 Demyelination. *J Neurosci* **37**, 7534-7546 (2017).

4. Kosaraju, J., *et al.* Metformin promotes CNS remyelination and improves social interaction
 following focal demyelination through CBP Ser436 phosphorylation. *Exp Neurol* **334**, 113454

- (2020).
- 5. Martinez, P. & Blasco, M.A. Telomere-driven diseases and telomere-targeting therapies. *J Cell*
*Biol* **216**, 875-887 (2017).
- 6. Blasco, M.A., *et al.* Telomere shortening and tumor formation by mouse cells lacking
telomerase RNA. *Cell* **91**, 25-34 (1997).
- 7. Assiri, M.A., *et al.* Investigating RNA expression profiles altered by nicotinamide
mononucleotide therapy in a chronic model of alcoholic liver disease. *Hum Genomics* **13**, 65
(2019).
- 8. Liang, H., *et al.* Nicotinamide mononucleotide alleviates Aluminum induced bone loss by
inhibiting the TXNIP-NLRP3 inflammasome. *Toxicol Appl Pharmacol* **362**, 20-27 (2019).
- 9. Chandrasekaran, K., *et al.* Nicotinamide Mononucleotide Administration Prevents
Experimental Diabetes-Induced Cognitive Impairment and Loss of Hippocampal Neurons. *Int*
*J Mol Sci* **21**(2020).
- 10. Zhou, X., *et al.* beta-Nicotinamide Mononucleotide (NMN) Administrated by Intraperitoneal
Injection Mediates Protection Against UVB-Induced Skin Damage in Mice. *J Inflamm Res* **14**,
5165-5182 (2021).
- 11. Goebbels, S., *et al.* Elevated phosphatidylinositol 3,4,5-trisphosphate in glia triggers
cell-autonomous membrane wrapping and myelination. *J Neurosci* **30**, 8953-8964 (2010).

Reviewer #3 (Remarks to the Author):

1047 Ma et al. report that SIRT2 is highly expressed in OPCs mainly with a nuclear localization during
myelin development. Demyelination, induced by LPC, allows for SIRT2 re-expression in the
nucleus of young mice. SIRT2 is necessary for remyelination following chemical demyelination.
They show that supplementation of NAD⁺ through β -NMN enhances SIRT2 nuclear entry in the
aged OPC upon demyelination in G3 Terc^{-/-} mice. They also demonstrate that NAD⁺
supplementation delays myelin aging and enhances remyelination in the aged mice, which
required SIRT2. Finally, they describe the molecular mechanism by which the addition of NMN
on OPC induces SIRT2 nuclear entry, SIRT2 deacetylates H3K18 leading to the inhibition of the
transcription factor ID4 which will in turn promote the differentiation of OPC. This paper
highlights a new role for SIRT2 in OPC aging. This is a timely finding given that the sirtuin field
is focused on the role of sirtuins in stem/progenitor cell aging and tissue degeneration.

Major comments:

(1) The nuclear localization of SIRT2 is not convincing. e.g. in Figure 1d, FDGFR, a cellular
surface protein, and in Figure 1g, NG2, also a plasma membrane protein, also appear to be
in the nucleus in P0 or P3. Since this study places strong emphasis on nuclear translocation
of SIRT2 as a mechanism, more convincing imaging data is necessary.

Reply: Thanks to Reviewer #3 for pointing out the imperfect images we used in submission. In
response, we have provided new images in “Figure 1d” and “Figure 1g”, as shown in below as
Reviewer #3-Figures 1a-c. To validate the nuclear localization of SIRT2 in an alternative way,
we showed the gray value of SIRT2, PDGFR α , DAPI (Reviewer #3-Figure 1b) and the gray
value of SIRT2, NG2, DAPI (Reviewer #3-Figure 1d). The results verified that SIRT2 is
partially colocalized with DAPI and partially colocalized with PDGFR α or NG2, supporting
our results on nuclear localization of SIRT2 in OPCs in this work. Parts of Reviewer#
3-Figure 1 have been used to replace parts of Revised Fig. 1d & 1g, and the whole figure has
been put as Revised Extended Data Fig.1b-e.

Reviewer #3-Figure 1 (Revised Extended Data Fig.1b-e)

Nuclear localization of SIRT2 in OPCs in the brains of P0 mouse and P3 marmoset. a,
Immunofluorescence of SIRT2⁺ cells in P0 mouse. b, Gray value of SIRT2, PDGFR α and DAPI.
c, Immunofluorescence of SIRT2⁺ cells in P3 marmoset. d, Gray value of SIRT2, NG2 and DAPI.

Also, quantification “cell number/mm²” can be complicated by overall cellular density in
the images. It is better to quantify “percent of cells with nuclear SIRT2”.

Reply: Thank Reviewer #3 for rightly pointing out that the quantification of “percent of cells with
nuclear SIRT2” is better than the quantification of cell density. Following this advice, we have
added the quantification of “percent of cells with nuclear SIRT2” to “Figure 1”, as shown below
in Reviewer #3-Figure 2. The percentage of OPCs with nuclear SIRT2 in cortex significantly
decreased at P14 and almost disappear at P21 and P90, well mirror the results shown in
Figure 1e.

Reviewer #3-Figure 2

The percentage of OPCs with nuclear SIRT2 in cortex of mouse during postnatal
development. Quantification of percentage of OPCs with nuclear SIRT2 in the cortex of mouse at
different ages. Data are presented as mean \pm SEM. * $p < 0.05$, ** $p < 0.01$, *** $p < 0.001$ by one-way
ANOVA followed by Tukey’s post hoc test.

“In addition, typical cellular localization studies include cellular fractionation and Western
blotting.”

Reply: We greatly appreciate Reviewer #3 for suggesting an effective alternative way to confirm
the nuclear localization of SIRT2. Using “PARIS™ kit” (ThermoFisher, #AM1921), we did new
experiment and separated the cytoplasm and the nucleus of OLN93 cells. Our new results showed
that SIRT2 localized in the cytoplasm of OLN93 cells, and β -NMN treatment caused part of the
SIRT2 translocated to the nucleus of OLN93 cells (Reviewer #3-Figure 3). These data provide
new evidence and support our immunofluorescence results that β -NMN promotes SIRT2 nuclear
entry. Reviewer #3-Figure 3 has been put as “Revised Fig. 7d”. Accordingly, we have added
lines 371-372 in “Results”.

Reviewer #3-Figure 3 (Revised Fig. 7d)

Cellular fractionation validates that β -NMN promotes SIRT2 nuclear entry. Western blot of
SIRT2 in nuclear protein and cytoplasmic protein of OLN93 cells treated with β -NMN or PBS.

(2) 1m and 1n: SIRT2⁺PDGFR α ⁺ cells were quantified. This could be complicated by
changes in PDGFR α ⁺ cells while the authors were trying to argue for an increase in SIRT2.
SIRT2⁺PDGFR α ⁺/PDGFR α ⁺ should be quantified.

Reply: Thanks to Reviewer #3 for a good advice. In response, we have quantified
SIRT2⁺PDGFR α ⁺/PDGFR α ⁺ in Figure 1m and 1n and produced Reviewer #3-Figure 4 as shown
below. Results showed that in comparison with non-lesion WT young mice, the proportion of
SIRT2⁺ OPCs in total OPCs significantly increased within the lesion in WT young mice after
demyelination, whereas it is still very low in the WT old mice after demyelination (Reviewer
#3-Figure 4). Reviewer #3-Figure 4 has been added as “Revised Fig.1m”.

Reviewer #3-Figure 4 (Revised Fig.1m)

Comparison of the proportion of SIRT2⁺ OPCs in total OPCs. Quantification of the
percentage of SIRT2⁺ OPCs in total OPCs in the corpus callosum of the non-lesion WT young
mice, the demyelinated WT young mice and the demyelinated WT old mice. NL-Y, non-lesioned
young; L-Y, lesioned young; L-O, lesioned old. Data are presented as mean \pm SEM. * p <0.05,
** p <0.01, *** p <0.001 by one-way ANOVA followed by Tukey's post hoc test.

(3) Figure 3: The argument that NMN increases SIRT2 expression and nuclear localization
is puzzling. NMN increases NAD levels, which is a cofactor of SIRT2. How would NAD
increase SIRT2 expression or nuclear localization?

Reply: Thank Reviewer #3 for raising a very interesting question. Several studies have reported
that the translocation of SIRT2 from cytoplasm to nuclear is mediated by dephosphorylation of
SIRT2 at serine 25^{1,2} or phosphorylation of SIRT2 at serine 368 and 372³, suggesting that
phosphorylation of SIRT2 is important for the location of SIRT2. In addition, several other studies
reported that phosphorylation of protein enhances the protein stability and thus regulates protein
level^{4,5}. NAD⁺ or β -NMN may directly or indirectly regulate the phosphorylation of SIRT2 to
regulate the protein stability and nuclear localization of SIRT2. This deserves further study and
will be an interesting independent story.

(4) Figure 6: 3-month-old SIRT2 KO mice were tested for the effect of NMN, without
age-matched WT control. In Figure 4 and 5, old WT mice responded to NMN. However, it is
unclear if young WT mice respond to NMN.

Reply: Thanks to Reviewer #3 for raising this important question. In response, new experiment of
 age-matched WT mice was performed and the results are shown below in Reviewer #3-Figure 5,
 and this new figure has been added as “Revised Fig. 6”.

To answer the question whether young WT mice respond to β -NMN, we daily i.p. injected
 β -NMN or PBS for 3 months in WT young mice (3-month-old). Then demyelination was induced
 by focal injection of LPC in the corpus callosum. We found at 5 dpl, long-term β -NMN
 supplementation did not change the proliferation of OPCs both in WT and SIRT2^{-/-} mice
 (Reviewer #3-Figures 5a-e). At 10 dpl, long-term β -NMN supplementation improved the
 differentiation of WT OPCs but had no effect on OPCs in SIRT2^{-/-} mice (Reviewer #3-Figures
 5f-j).

Then we tested remyelination at 21 dpl when remyelination completes (Reviewer #3-Figures
 5k-l), and our results showed that in WT young mice, the frequency of grade 0 myelin (normal
 myelin) was increased while grade 2 myelin was decreased after β -NMN supplementation
 (Reviewer #3-Figure 5m). The frequency of remyelinated axons and myelin thickness did not
 change (Reviewer #3-Figures 5n-r). These new data showed that young WT mice respond to
 β -NMN as well: β -NMN promotes differentiation of OPCs and improves the myelin quality
 in young WT mice.

In SIRT2^{-/-} mice, β -NMN supplementation did not change any of these indexes (Reviewer
 #3-Figures 5k-r), indicating that SIRT2 is required or indispensable for the effect of β -NMN on
 remyelination, validating the conclusion in our submission.

Reviewer #3-Figure 5 (Revised Fig. 6)

In young WT mice, β -NMN promotes differentiation of OPCs and improves the myelin
 quality. a, Experiment design for testing the impact of β -NMN on OPC proliferation in vivo. b-e,
 Images and quantification of oligodendrocyte lineage cells and proliferating OPCs within the
 demyelination lesions at 5 dpl (n=5). Scale bar, 50 μ m. f, Experiment design for testing the impact
 of β -NMN on OPC differentiation in vivo. g-j, Images and quantification of oligodendrocyte
 lineage cells and differentiated oligodendrocytes within the demyelination lesions at 10 dpl (n=5).
 Scale bar, 50 μ m. k, Experiment design for testing the impact of β -NMN on remyelination
 efficiency in vivo at 21 dpl. l, TEM micrographs within the lesions at 21 dpl. Scale bar, 1 μ m. m-r,
 Quantification of the myelin pathology level (m), proportion of remyelinated axons (n), G-Ratio
 average (o), individual G-Ratio distribution (p, linear regression) and distance between DL (q and
 r) within the lesions at 21 dpl (n=3 in the WT group and n=5 in the SIRT2^{-/-} group). All data are
 presented as mean \pm SEM. The center, upper and lower line represent the median, upper and lower
 quartiles, respectively (r). *p<0.05, **p<0.01, ***p<0.001 by one-way ANOVA followed by
 Tukey's post hoc test (c-e, h-j, n, o, q, r) or two-way repeated ANOVA followed by Sidak's post
 hoc test (m).

(5) Figure 7f shows increased ID4 in SIRT2 KO brain but this could be complicated by
 changes in cell number etc. What about in OPC cells specifically?

Reply: Thanks to Reviewer #3 for this question. In response to the concern of Reviewer #3, we
 have tested the mRNA level of ID4 in OPCs cultured from SIRT2^{-/-} mice and found that in
 comparison with OPCs of WT mice, the mRNA level of ID4 significantly increased in SIRT2^{-/-}
 OPCs (Reviewer #3-Figure 6). The Reviewer #3-Figure 6 has been put as "Revised Fig. 7i".

Reviewer #3-Figure 6 (Revised Fig. 7i)

The mRNA level of ID4 increases in SIRT2^{-/-} OPCs. Relative mRNA level of ID4 in primary
 cultured OPCs of SIRT2^{-/-} mice and the WT mice (n=3). Data are presented as mean \pm SEM.
 **p<0.01 by two-tailed t test.

(6) Figure 7h. SIRT2 KO cells should be used as a control to ensure specific binding of
 SIRT2. As discussed above, it is puzzling how NMN increases SIRT2 binding to
 chromosome.

Reply: Thanks to Reviewer #3 for raising this important question. Following the advice of
 Reviewer #3, we have used SIRT2 knockdown OLN93 cells as a control, our results show that

β -NMN increases SIRT2 binding to promoter of ID4 (Reviewer #3-Figure 7). The Reviewer
#3-Figure 7 has been put as “Revised Fig. 7i”.

How does NMN increase SIRT2 binding to chromosome? This is an interesting question which
certainly deserves further study. Pereira et al. reported that dephosphorylation of SIRT2 at serine
25 during infection is an essential event for SIRT2 chromatin association¹. We speculate NAD⁺ or
β -NMN may directly or indirectly regulate the phosphorylation of SIRT2 to regulate the SIRT2
chromatin association. Due to time limit for the revision and the focus of this paper is the effect of
β -NMN on myelin aging and remyelination, we feel further study to explore how SIRT2 binds to
chromosome and how SIRT2 location is regulated will be a very interesting independent story.

Reviewer #3-Figure 7 (Revised Fig. 7i)

β -NMN increases SIRT2 binding to promoter of ID4. ChIP-qPCR assessment of the
enrichment of SIRT2 at the promoter region of ID4 in OLN93 cell line knocking down or
overexpressing SIRT2 (n=3). Data are presented as mean \pm SEM. ***p<0.001 by two-way
repeated ANOVA followed by Sidak's post hoc test.

(7) Figure 7i. SIRT2 has deacetylase activity. If NMN increases SIRT2 binding to the
promoter, it would reduce H3K18Ac. This is opposite to what's shown in 7i.

Reply: We sincerely thank Reviewer #3 for pointing out this serious mistake. After we carefully
rechecked the original data and records, we realized that we made a mistake by confusing data
between two groups when we prepared figures. We sincerely apologize for this primitive but
serious mistake! The reminding of the Reviewer #3 made us realize that WT OLN93 cells is better
than SIRT2 over-expression OLN93 cells, because the acetylation level of H3K 18 is higher in WT
OLN93 cells (Reviewer #3-Figure 8a).

To resolve any concerns and reinvestigate this question, we did new experiment using WT OLN93
cells, and the new results showed that the enrichment of H3K 18Ac significantly decreased
after β -NMN treatment (Reviewer #3-Figure 8b), consisting with the results in the SIRT2
over-expression OLN93 cells.

The Reviewer #3-Figure 8 has been put as “Revised Fig. 7m”. Accordingly, the results have
been modified (lines 392-393).

Reviewer #3-Figure 8 (Revised Fig. 7m)

β-NMN decreases H3K 18Ac binding to promoter of ID4. a, Immunoblot of H3K 18Ac (primary antibody obtained from rabbit) and H3 (primary antibody obtained from mouse) in OLN93 cell line. b, ChIP-qPCR assessment of the enrichment of H3K18Ac at the promoter region of ID4 in WT OLN93 cell line (n=3). Data are presented as mean ± SEM. ***p<0.001 by two-way repeated ANOVA followed by Sidak's post hoc test.

Minor comments:

(8) Redundant figures 1a, 2b, 3d and 6a. No need to show “cultured cells or OPC” as a figure or just once.

Reply: Thanks to Reviewer #3 for the suggestion. Following the advice, we have deleted the

redundant figures 2b, 3d and 6a.

(9) Line 140: “in contrast”

Reply: Thanks to Reviewer #3 for careful spotting the spelling errors. We have changed “in

contract” to “in contrast”.

References

1. Pereira, J.M., *et al.* Infection Reveals a Modification of SIRT2 Critical for Chromatin Association. *Cell Rep* **23**, 1124-1137 (2018).

2. Eskandarian, H.A., *et al.* A role for SIRT2-dependent histone H3K18 deacetylation in bacterial infection. *Science* **341**, 1238858 (2013).

3. Zhang, Z., *et al.* CDK5-mediated phosphorylation of Sirt2 contributes to depressive-like behavior induced by social defeat stress. *Biochim Biophys Acta Mol Basis Dis* **1864**, 533-541 (2018).

4. Vazquez, F., Ramaswamy, S., Nakamura, N. & Sellers, W.R. Phosphorylation of the PTEN tail regulates protein stability and function. *Mol Cell Biol* **20**, 5010-5018 (2000).

5. Li, Y., Dowbenko, D. & Lasky, L.A. AKT/PKB phosphorylation of p21Cip/WAF1 enhances protein stability of p21Cip/WAF1 and promotes cell survival. *J Biol Chem* **277**, 11352-11361

(2002).

Reviewers' Comments:

Reviewer #1:

Remarks to the Author:

The authors have addressed all points raised and greatly improved the manuscript with additional experiments and by extensively revising text and figures. I think the paper is acceptable as is.

Reviewer #2:

Remarks to the Author:

In the revised manuscript Ma et al have done several new experiments to their credit. The two of which particularly stand out are a) the use of NG2CreERT:Sirt2flox/flox mice to show that the main reason for impaired remyelination and poor grade myelin seen in Sirt2 deficient mice is due to its absence mainly from OPCs and b) the work of β NMN supplementation in non-lesioned G3 Terc-/- and old WT mice. I like the revised figure for the working model that is now straight forward. There are still some things that need further work like why in vitro and invivo un-lesioned G3Terc-/- and older animals bNMN administration increases OPC density and OPC differentiation but in lesioned animals it does not and only improves differentiation. However, having said that it does not detract from the authors' main message of the paper, and I am satisfied with their changes and as the manuscript stand.

Few minor suggestions, please simplify the language in lines 144-47, it makes for a difficult reading. Also, addition of the words "untreated" towards the end of line 270 would eliminate confusion as to which group they are referring to since they start out by talking about NAD+ supplementation. Other than that, a quick proofread will fix minor spelling/grammar issues.

Reviewer #3:

Remarks to the Author:

The authors have addressed the comments. I support its publication.

Reviewer #1 (Remarks to the Author):

The authors have addressed all points raised and greatly improved the manuscript with additional experiments and by extensively revising text and figures. I think the paper is acceptable as is.

Reply: We greatly appreciate the positive comments made by Reviewer #1 on our revised manuscript.

Reviewer #2 (Remarks to the Author):

In the revised manuscript Ma et al have done several new experiments to their credit. The two of which particularly stand out are a) the use of NG2CreERT:Sirt2flox/flox mice to show that the main reason for impaired remyelination and poor grade myelin seen in Sirt2 deficient mice is due to its absence mainly from OPCs and b) the work of bNMN supplementation in non-lesioned G3 Terc^{-/-} and old WT mice. I like the revised figure for the working model that is now straight forward.

Reply: We really appreciate Reviewer #2's positive comments on our new experiments and the revised figure for the working model.

There are still some things that need further work like why in vitro and in vivo un-lesioned G3Terc^{-/-} and older animals bNMN administration increases OPC density and OPC differentiation but in lesioned animals it does not and only improves differentiation. However, having said that it does not detract from the authors' main message of the paper, and I am satisfied with their changes and as the manuscript stand.

Reply: Thanks to Reviewer #2 for pointing out an interesting point. We agree with Reviewer #2 and will address it in another independent story.

Few minor suggestions, please simplify the language in lines 144-47, it makes for a difficult reading. Also, addition of the words "untreated" towards the end of line 270 would eliminate confusion as to which group they are referring to since they start out by talking about NAD⁺ supplementation. Other than that, a quick proofread will fix minor spelling/grammar issues.

Reply: We appreciate Reviewer #2 for pointing out these. Following advice of Reviewer #2, we have simplified the language to "In contrast, in the old mice aged 21M, demyelination upregulated the level of SIRT2 in OPCs to a much lower level than the young mice (Fig. 1I, Extended Data Fig. 2d), equivalent to 1/3 in the proportion (Fig. 1m) and about half in the density of SIRT2⁺ OPCs of the young mice (Fig. 1n). Among SIRT2⁺ OPCs, nuclear SIRT2⁺ OPCs in the aged was only 1/3 of the young (Fig. 1I-o, Extended Data Fig. 2d)." in lines 144-148, and we have added "untreated" to the end of lines 269 and 271. We also have carefully checked the whole manuscript and corrected the spelling/grammar issues we have spotted to our credit. All the changes are highlighted in yellow.

Reviewer #3 (Remarks to the Author):

The authors have addressed the comments. I support its publication.

Reviewed by Danica Chen

Reply: We really appreciate the expert comments made by Dr. Danica Chen, the Reviewer #3.